# Implicit Bias of MSE Gradient Optimization in Underparameterized Neural Networks

**Benjamin Bowman**
UCLA Departments of Mathematics
`benbowman314@math.ucla.edu`

**Guido Montúfar**
UCLA Departments of Mathematics and Statistics
and MPI MIS
`montufar@math.ucla.edu`

## Abstract

We study the dynamics of a neural network in function space when optimizing the mean squared error via gradient flow. We show that in the underparameterized regime the network learns eigenfunctions of an integral operator $T_{K^\infty}$ determined by the Neural Tangent Kernel (NTK) at rates corresponding to their eigenvalues. For example, for uniformly distributed data on the sphere $S^{d-1}$ and rotation invariant weight distributions, the eigenfunctions of $T_{K^\infty}$ are the spherical harmonics. Our results can be understood as describing a spectral bias in the underparameterized regime. The proofs use the concept of "Damped Deviations", where deviations of the NTK matter less for eigendirections with large eigenvalues due to the occurence of a damping factor. Aside from the underparameterized regime, the damped deviations point-of-view can be used to track the dynamics of the empirical risk in the overparameterized setting, allowing us to extend certain results in the literature. We conclude that damped deviations offers a simple and unifying perspective of the dynamics when optimizing the squared error.

## 1 Introduction

A surprising but well established empirical fact is that neural networks optimized by gradient descent can find solutions to the empirical risk minimization (ERM) problem that generalize. This is surprising from an optimization point-of-view because the ERM problem induced by neural networks is nonconvex (Sontag & Sussmann, 1989; 1991) and can even be NP-Complete in certain cases (Blum & Rivest, 1993). Perhaps even more surprising is that the discovered solution can generalize even when the network is able to fit arbitrary labels (Zhang et al., 2017), rendering traditional complexity measures such as Rademacher complexity inadequate. How does deep learning succeed in the face of pathological behavior by the standards of classical optimization and statistical learning theory?

Towards addressing generalization, a modern line of thought that has emerged is that gradient descent performs implicit regularization, limiting the solutions one encounters in practice to a favorable subset of the model's full capacity (see, e.g., Neyshabur et al., 2015; 2017; Gunasekar et al., 2017; Wu et al., 2017). An empirical observation is that neural networks optimized by gradient descent tend to fit the low frequencies of the target function first, and only pick up the higher frequencies later in training (Rahaman et al., 2019; Ronen et al., 2019; Basri et al., 2020; Xu et al., 2019). A closely related theme is gradient descent's bias towards smoothness for regression problems (Williams et al., 2019; Jin & Montúfar, 2021). For classification problems, in suitable settings gradient descent provably selects max-margin solutions (Soudry et al., 2018; Ji & Telgarsky, 2019). Gradient descent is not impartial, thus understanding its bias is an important program in modern deep learning.

Generalization concerns aside, the fact that gradient descent can succeed in a nonconvex optimization landscape warrants attention on its own. A brilliant insight made by Jacot et al. (2018) is that in function space the neural network follows a kernel gradient descent with respect to the "Neural Tangent Kernel" (NTK). This kernel captures how the parameterization biases the trajectory in function space, an abstraction that allows one to largely ignore parameter space and its complications. This is a profitable point-of-view, but there is a caveat. The NTK still depends on the evolution of the network parameters throughout time, and thus is in general time-dependent and complicated to analyze. However, under appropriate scaling of the parameters in the infinite-width limit it remains

constant (Jacot et al., 2018). Once the NTK matrix has small enough deviations to remain strictly positive definite throughout training, the optimization dynamics start to become comparable to that of a linear model (Lee et al., 2019). For wide networks (quadratic or higher polynomial dependence on the number of training data samples $n$ and other parameters) this property holds and this has been used by a variety of works to prove global convergence guarantees for the optimization (Du et al., 2019b; Oymak & Soltanolkotabi, 2020; Du et al., 2019a; Allen-Zhu et al., 2019a;b; Zou et al., 2020; Zou & Gu, 2019; Song & Yang, 2020; Dukler et al., 2020)[1] and to characterize the solution throughout time (Arora et al., 2019; Basri et al., 2020). The NTK has been so heavily exploited in this setting that it has become synonymous with polynomially wide networks where the NTK is strictly positive definite throughout training. This begs the question, to what extent is the NTK informative outside this regime?

While the NTK has hitherto been associated with the heavily overparameterized regime, we demonstrate that refined analysis is possible in the underparameterized setting. Our theorems primarily concern a one-hidden layer network, however unlike many NTK results appearing in the literature our network has *biases* and *both* layers are trained. In fact, the machinery we build is strong enough to extend some existing results in the overparameterized regime appearing in the literature to the case of training both layers.

## 1.1 RELATED WORK

There has been a deluge of works on the Neural Tangent Kernel since it was introduced by Jacot et al. (2018), and thus we do our best to provide a partial list. Global convergence guarantees for the optimization, and to a lesser extent generalization, for networks polynomially wide in the number of training samples $n$ and other parameters has been addressed in several works (Du et al., 2019b; Oymak & Soltanolkotabi, 2020; Du et al., 2019a; Allen-Zhu et al., 2019a;b; Zou et al., 2020; Zou & Gu, 2019; Song & Yang, 2020; Arora et al., 2019). To our knowledge, for the regression problem with arbitrary labels, quadratic overparameterization $m \gtrsim n^2$ is state-of-the art (Oymak & Soltanolkotabi, 2020; Song & Yang, 2020; Nguyen & Mondelli, 2020). E et al. (2020) gave a fairly comprehensive study of optimization and generalization of shallow networks trained under the standard parameterization. Under the standard parameterization, changes in the outer layer weights are more significant, whereas under the NTK parameterization both layers have roughly equal effect. Since we study the NTK parameterization in this work, we view the analysis as complementary.

Our work is perhaps most closely connected with Arora et al. (2019). In Theorem 4.1 in that work they showed that for a shallow network in the polynomially overparameterized regime $m \gtrsim n^7$, the training error along eigendirections of the NTK matrix decay linearly at rates that correspond to their eigenvalues. Our main Theorem 3.5 can be viewed as an analogous statement for the actual risk (not the empirical risk) in the underparameterized regime: eigenfunctions of the NTK integral operator $T_{K^\infty}$ are approximately learned linearly at rates that correspond to their eigenvalues. In contrast with Arora et al. (2019), we have that the requirements on width $m$ and number of samples $n$ required to learn eigenfunctions with large eigenvalues are smaller compared to those with small eigenvalues. Surprisingly the machinery we build is also strong enough to prove in our setting the direct analog of Theorem 4.1. Note that Arora et al. (2019) train the hidden layer of a ReLU network via gradient descent, whereas we are training both layers with biases for a network with smooth activations via gradient flow. Due to the different settings, the results are not directly comparable. This important detail notwithstanding, our overparameterization requirement ignoring logarithmic factors is smaller by a factor of $\frac{n^2}{d\delta^4}$ where $n$ is the number of input samples, $d$ is the input dimension, and $\delta$ is the failure probability. Basri et al. (2020) extended Theorem 4.1 in Arora et al. (2019) to deep ReLU networks without bias where the first and last layer are fixed, with a higher overparameterization requirement than the original (Arora et al., 2019). Since the first and last layers are fixed this cannot be specialized to get a guarantee for training both layers of a shallow network even with ReLU activations.

Although it was not our focus, the tools to prove Theorem 3.5 are enough to prove analogs of Theorem 4 and Corollary 2 in the work of Su & Yang (2019). Theorem 4 and Corollary 2 of Su & Yang (2019) are empirical risk guarantees that show that for target functions that participate mostly

---

[1] Not all these works explicitly use that the NTK is positive definite. However, they all operate in the regime where the weights do not vary much and thus are typically associated with the NTK regime.

in the top eigendirections of the NTK integral operator $T_{K^\infty}$, moderate overparameterization is possible. Again in this work they train the hidden layer of a ReLU network via gradient descent, whereas we are training both layers with biases for a network with smooth activations via gradient flow. Again due to the different settings, we emphasize the results are not directly comparable. In our results the bounds and requirements are comparable to Su & Yang (2019), with neither appearing better. Nevertheless we think it is important to demonstrate that these results hold for training both layers with biases, and we hope our "Damped Deviations" approach will simplify the interpretation of the aforementioned works.

Cao et al. (2020, Theorem 4.2) provide an analogous statement to our Theorem 3.5 if you replace our quantities with their empirical counterparts. While our statement concerns the projections of the test residual onto the eigenfunctions of an operator associated with the Neural Tangent Kernel, their statement concerns the inner products of the empirical residual with those eigenfunctions. Their work was a crucial step towards explaining the spectral bias from gradient descent, however we view the difference between tracking the empirical quantities versus the actual quantities to be highly nontrivial. Another difference is they consider a ReLU network whereas we consider smooth activations; also they consider gradient descent versus we consider gradient flow. Due to the different settings we would like to emphasize that the scalings of the different parameters are not directly comparable, nevertheless the networks they consider are significantly wider. They require at least $m \geq \tilde{O}(\max\{\sigma_k^{-14}, \epsilon^{-6}\})$, where $\sigma_k$ is a cutoff eigenvalue and $\epsilon$ is the error tolerance. By contrast in our work, to have the projection onto the top $k$ eigenvectors be bounded by epsilon in L2 norm requires $m = \tilde{\Omega}(\sigma_k^{-4}\epsilon^{-2})$. Another detail is their network has no bias whereas ours does.

## 1.2 OUR CONTRIBUTIONS

The key idea for our work is the concept of "Damped Deviatons", the fact that for the squared error deviations of the NTK are softened by a damping factor, with large eigendirections being damped the most. This enables the following results.

- In Theorem 3.5 we characterize the bias of the neural network to learn the eigenfunctions of the integral operator $T_{K^\infty}$ associated with the Neural Tangent Kernel (NTK) at rates proportional to the corresponding eigenvalues.

- In Theorem 3.7 we show that in the overparameterized setting the training error along different directions can be sharply characterized, showing that Theorem 4.1 in Arora et al. (2019) holds for smooth activations when training both layers with a smaller overparameterization requirement.

- In Theorem 3.8 and Corollary 3.9 we show that moderate overparameterization is sufficient for solving the ERM problem when the target function has a compact representation in terms of eigenfunctions of $T_{K^\infty}$. This extends the results in Su & Yang (2019) to the setting of training both layers with smooth activations.

## 2 GRADIENT DYNAMICS AND DAMPED DEVIATIONS

### 2.1 NOTATIONS

We will use $\|\bullet\|_2$ and $\langle\bullet,\bullet\rangle_2$ to denote the $L^2$ norm and inner product respectively (for vectors or for functions depending on context). For a symmetric matrix $A \in \mathbb{R}^{k \times k}$, $\lambda_i(A)$ denotes its $i$th largest eigenvalue, i.e. $\lambda_1(A) \geq \lambda_2(A) \geq \cdots \geq \lambda_k(A)$. For a matrix $A$, $\|A\|_{op} := \sup_{\|x\|_2 \leq 1} \|Ax\|_2$ is the operator norm induced by the Euclidean norm. We will let $\langle\bullet,\bullet\rangle_{\mathbb{R}^n}$ denote the standard inner product on $\mathbb{R}^n$ normalized by $\frac{1}{n}$, namely $\langle x,y\rangle_{\mathbb{R}^n} = \frac{1}{n}\langle x,y\rangle_2 = \frac{1}{n}\sum_{i=1}^{n} x_i y_i$. We will let $\|x\|_{\mathbb{R}^n} = \sqrt{\langle x,x\rangle_{\mathbb{R}^n}}$ be the associated norm. This normalized inner product has the convenient property that if $v \in \mathbb{R}^n$ such that $v_i = O(1)$ for each $i$ then $\|v\|_{\mathbb{R}^n} = O(1)$, where by contrast $\|v\|_2 = O(\sqrt{n})$. This is convenient as we will often consider what happens when $n \to \infty$. $\|\bullet\|_\infty$ will denote the supremum norm with associated space $L^\infty$. We will use the standard big $O$ and $\tilde{\Omega}$ notation with $\tilde{O}$ and $\tilde{\Omega}$ hiding logarithmic terms.

## 2.2 GRADIENT DYNAMICS AND THE NTK INTEGRAL OPERATOR

We will let $f(x; \theta)$ denote our neural network taking input $x \in \mathbb{R}^d$ and parameterized by $\theta \in \mathbb{R}^p$. The specific architecture of the network does not matter for the purposes of this section. Our training data consists of $n$ input-label pairs $\{(x_1, y_1), \ldots, (x_n, y_n)\}$ where $x_i \in \mathbb{R}^d$ and $y_i \in \mathbb{R}$. We focus on the setting where the labels are generated from a fixed target function $f^*$, i.e. $y_i = f^*(x_i)$. We will concatenate the labels into a label vector $y \in \mathbb{R}^n$, i.e. $y_i = f^*(x_i)$. We will let $\hat{r}(\theta) \in \mathbb{R}^n$ be the vector whose $i$th entry is equal to $f(x_i; \theta) - f^*(x_i)$. Hence $\hat{r}(\theta)$ is the residual vector that measures the difference between our neural networks predictions and the labels. We will be concerned with optimizing the squared loss

$$\Phi(\theta) = \frac{1}{2n} \|\hat{r}(\theta)\|_2^2 = \frac{1}{2} \|\hat{r}(\theta)\|_{\mathbb{R}^n}^2 .$$

Optimization will be done by gradient flow

$$\partial_t \theta_t = -\partial_\theta \Phi(\theta),$$

which is the continuous time analog of gradient descent. We will denote the residual at time $t$, $\hat{r}(\theta_t)$, as $\hat{r}_t$ for the sake of brevity and similarly we will let $f_t(x) = f(x; \theta_t)$. We will let $r_t(x) := f_t(x) - f^*(x)$ denote the residual off of the training set for an arbitrary input $x$.

We quickly recall some facts about the Neural Tangent Kernel and its connection to the gradient dynamics. For a comprehensive tutorial we suggest Jacot et al. (2018). The analytical NTK is the kernel given by

$$K^\infty(x, x') := \mathbb{E}\left[\left\langle \frac{\partial f(x; \theta)}{\partial \theta}, \frac{\partial f(x'; \theta)}{\partial \theta} \right\rangle_2\right],$$

where the expectation is taken with respect to the parameter initialization for $\theta$. We associate $K^\infty$ with the integral operator $T_{K^\infty} : L_\rho^2(X) \to L_\rho^2(X)$ defined by

$$T_{K^\infty} f(x) := \int_X K^\infty(x, s) f(s) d\rho(s),$$

where $X$ is our input space with probability measure $\rho$. Our training data $x_i \in X$ are distributed according to this measure $x_i \sim \rho$. By Mercer's theorem we can decompose

$$K^\infty(x, x') = \sum_{i=1}^\infty \sigma_i \phi_i(x) \phi_i(x'),$$

where $\{\phi_i\}_{i=1}^n$ is an orthonormal basis of $L^2$, $\{\sigma_i\}_{i=1}^\infty$ is a nonincreasing sequence of positive values, and each $\phi_i$ is an eigenfunction of $T_{K^\infty}$ with eigenvalue $\sigma_i > 0$. When $X = S^{d-1}$ is the unit sphere, $\rho$ is the uniform distribution, and the weights of the network are from a rotation invariant distribution (e.g. standard Gaussian), $\{\phi_i\}_{i=1}^\infty$ are the spherical harmonics (which in $d = 2$ is the Fourier basis) due to $K^\infty$ being rotation-invariant (see Bullins et al., 2018, Theorem 2.2). We will let $\kappa := \max_{x \in X} K^\infty(x, x)$ which will be a relevant quantity in our later theorems. In our setting $\kappa$ will always be finite as $K^\infty$ will be continuous and $X$ will be bounded. The training data inputs $\{x_1, \ldots, x_n\}$ induce a discretization of the integral operator $T_{K^\infty}$, namely

$$T_n f(x) := \frac{1}{n} \sum_{i=1}^n K^\infty(x, x_i) f(x_i) = \int_X K^\infty(x, s) f(s) d\rho_n(s),$$

where $\rho_n = \frac{1}{n} \sum_{i=1}^n \delta_{x_i}$ is the empirical measure. We recall the definition of the time-dependent $NTK^2$,

$$K_t(x, x') := \left\langle \frac{\partial f(x; \theta_t)}{\partial \theta}, \frac{\partial f(x'; \theta_t)}{\partial \theta} \right\rangle_2.$$

We can look at the version of $T_n$ corresponding to $K_t$, namely

$$T_n^t f(x) := \frac{1}{n} \sum_{i=1}^n K_t(x, x_i) f(x_i) = \int_X K_t(x, s) f(s) d\rho_n(s).$$

---

[2] The NTK is an overloaded term. To help ameliorate the confusion, we will use NTK to describe $K^\infty$ and $NTK$ (italic font) to describe the time-dependent version $K_t$.

We recall that the residual $r_t(x) := f(x; \theta) - f^*(x)$ follows the update rule

$$\partial_t r_t(x) = -\frac{1}{n}\sum_{i=1}^{n} K_t(x, x_i) r_t(x_i) = -T_n^t r_t.$$

We will let $(H_t)_{i,j} := K_t(x_i, x_j)$ and $H_{i,j}^{\infty} := K^{\infty}(x_i, x_j)$ denote the Gram matrices induced by these kernels and we will let $G_t := \frac{1}{n}H_t$ and $G^{\infty} := \frac{1}{n}H^{\infty}$ be their normalized versions[3]. Throughout we will let $u_1, \ldots, u_n$ denote the eigenvectors of $G^{\infty}$ with corresponding eigenvalues $\lambda_1, \ldots, \lambda_n$. The $u_1, \ldots, u_n$ are chosen to be orthonormal with respect to the inner product $\langle \bullet, \bullet \rangle_{\mathbb{R}^n}$. When restricted to the training set we have the update rule

$$\partial_t \hat{r}_t = -\frac{1}{n}H_t \hat{r}_t = -G_t \hat{r}_t.$$

## 2.3 Damped Deviations

The concept of damped deviations comes from the very simple lemma that follows (the proof is provided in Appendix D). The lemma compares the dynamics of the residual $\hat{r}(t)$ on the training set to the dynamics of an arbitrary kernel regression $\exp(-Gt)\hat{r}(0)$:

**Lemma 2.1.** *Let $G \in \mathbb{R}^{n\times n}$ be an arbitrary positive semidefinite matrix and let $G_s$ be the time dependent NTK matrix at time $s$. Then*

$$\hat{r}_t = \exp(-Gt)\hat{r}_0 + \int_0^t \exp(-G(t-s))(G - G_s)\hat{r}_s ds.$$

Let's specialize the lemma to the case where $G = G^{\infty}$. In this case the first term is $\exp(-G^{\infty}t)\hat{r}_0$, which is exactly the dynamics of the residual in the exact NTK regime when $G_t = G^{\infty}$ for all $t$. The second term is a correction term that weights the NTK deviations $(G^{\infty} - G_s)$ by the damping factor $\exp(-G^{\infty}(t-s))$. We see that damping is largest along the large eigendirections of $G^{\infty}$. The equation becomes most interpretable when projected along a specific eigenvector. Fix an eigenvector $u_i$ of $G^{\infty}$ corresponding to eigenvalue $\lambda_i$. Then the equation along this component becomes

$$\langle \hat{r}_t, u_i \rangle_{\mathbb{R}^n} = \exp(-\lambda_i t)\langle \hat{r}_0, u_i \rangle_{\mathbb{R}^n} + \int_0^t \langle \exp(-\lambda_i(t-s))(G^{\infty} - G_s)\hat{r}_s, u_i \rangle_{\mathbb{R}^n} ds.$$

The first term above converges to zero at rate $\lambda_i$. The second term is a correction term that weights the deviatiations of the $NTK$ matrix $G_s$ from $G^{\infty}$ by the damping factor $\exp(-\lambda_i(t-s))$. The second term can be upper bounded by

$$\left| \int_0^t \langle \exp(-\lambda_i(t-s))(G^{\infty} - G_s)\hat{r}_s, u_i \rangle_{\mathbb{R}^n} ds \right| \leq \int_0^t \exp(-\lambda_i(t-s)) \|G^{\infty} - G_s\|_{op} \|\hat{r}_s\|_{\mathbb{R}^n} ds$$

$$\leq \frac{[1 - \exp(-\lambda_i t)]}{\lambda_i} \sup_{s\in[0,t]} \|G^{\infty} - G_s\|_{op} \|\hat{r}_0\|_{\mathbb{R}^n},$$

where we have used the property $\|\hat{r}_s\|_{\mathbb{R}^n} \leq \|\hat{r}_0\|_{\mathbb{R}^n}$ from gradient flow. When $f^* = O(1)$ we have that $\|\hat{r}_0\|_{\mathbb{R}^n} = O(1)$, thus whenever $\|G^{\infty} - G_s\|_{op}$ is small relative to $\lambda_i$ this term is negligible. It has been identified that the NTK matrices tend to have a small number of outlier large eigenvalues and exhibit a low rank structure (Oymak et al., 2020; Arora et al., 2019). In light of this, the dependence of the above bound on the magnitude of $\lambda_i$ is particularly interesting. We reach following important conclusion.

**Observation 2.2.** *The dynamics in function space will be similar to the NTK regime dynamics along eigendirections whose eigenvalues are large relative to the deviations of the time-dependent NTK matrix from the analytical NTK matrix.*

The equation in Lemma 2.1 concerns the residual restricted to the training set, but we will be interested in the residual for arbitrary inputs. Recall that $r_t(x) = f(x; \theta_t) - f^*(x)$ denotes the residual at time $t$ for an arbitrary input. Then more generally we have the following damped deviations lemma for the whole residual (proved in Appendix C.3).

---

[3] $G_t$ and $G^{\infty}$ are the natural matrices to work with when working with the mean squared error as opposed to the unnormalized squared error. Also $G^{\infty}$'s spectra concentrates around the spectrum of the associated integral operator $T_{K^{\infty}}$ and is thus a more convenient choice in our setting.

**Lemma 2.3.** *Let $K(x, x')$ be an arbitrary continuous, symmetric, positive-definite kernel. Let $[T_K h](\bullet) = \int_X K(\bullet, s)h(s)d\rho(s)$ be the integral operator associated with $K$ and let $[T_n^s h](\bullet) = \frac{1}{n}\sum_{i=1}^n K_s(\bullet, x_i)h(x_i)$ denote the operator associated with the time-dependent $NTK$ $K_s$. Then*

$$r_t = \exp(-T_K t)r_0 + \int_0^t \exp(-T_K(t-s))(T_K - T_n^s)r_s ds,$$

*where the equality is in the $L^2$ sense.*

For our main results we will specialize the above lemma to the case where $K = K^\infty$. However there are other natural kernels to compare against, say $K_0$ or the kernel corresponding to some subset of parameters. We will elaborate further on this point after we introduce the main theorem. When specializing Lemma 2.3 to the case $K = K^\infty$, we have that $T_{K^\infty}$ and $T_n^s$ are the operator analogs of $G^\infty$ and $G_s$ respectively. From this statement the same concepts holds as before, the dynamics of $r_t$ will be similar to that of $\exp(-T_{K^\infty}t)r_0$ along eigendirections whose eigenvalues are large relative to the deviations $(T_{K^\infty} - T_n^s)$. In the underparameterized regime we can bound the second term and make it negligible (Theorem 3.5) and thus demonstrate that the eigenfunctions $\phi_i$ of $T_{K^\infty}$ with eigenvalues $\sigma_i$ will be learned at rate $\sigma_i$. When the input data are distributed uniformly on the sphere $S^{d-1}$ and the network weights are from a rotation-invariant distribution, the eigenfunctions of $T_{K^\infty}$ are the spherical harmonics (which is the Fourier basis when $d = 2$). In this case the network is biased towards learning the spherical harmonics that correspond to large eigenvalues of $T_{K^\infty}$. It is in this vein that we will demonstrate a spectral bias.

## 3    MAIN RESULTS

Our theorems will concern the shallow neural network

$$f(x; \theta) = \frac{1}{\sqrt{m}}\sum_{\ell=1}^m a_\ell \sigma(\langle w_\ell, x\rangle_2 + b_\ell) + b_0 = \frac{1}{\sqrt{m}}a^T \sigma(Wx + b) + b_0,$$

where $W \in \mathbb{R}^{m \times d}$, $a, b \in \mathbb{R}^m$ and $b_0 \in \mathbb{R}$ and $w_\ell = W_{\ell,:}$ denotes the $\ell$th row of $W$ and $\sigma : \mathbb{R} \to \mathbb{R}$ is applied entry-wise. $\theta = (a^T, vec(W)^T, b^T, b_0)^T \in \mathbb{R}^p$ where $p = md + 2m + 1$ is the total number of parameters. Here we are utilizing the NTK parameterization (Jacot et al., 2018). For a thorough analysis using the standard parameterization we suggest E et al. (2020). We will consider two parameter initialization schemes. The first initializes $W_{i,j}(0) \sim \mathcal{W}$, $b_\ell(0) \sim \mathcal{B}$, $a_\ell(0) \sim \mathcal{A}$, $b_0 \sim \mathcal{B}'$ i.i.d., where $\mathcal{W}, \mathcal{B}, \mathcal{A}, \mathcal{B}'$ represent zero-mean unit variance subgaussian distributions. In the second initialization scheme we initialize the parameters according to the first scheme and then perform the following swaps $W(0) \to \begin{bmatrix} W(0) \\ W(0) \end{bmatrix}$, $b(0) \to \begin{bmatrix} b(0) \\ b(0) \end{bmatrix}$, $a(0) \to \begin{bmatrix} a(0) \\ -a(0) \end{bmatrix}$, $b_0 \to 0$ and replace the $\frac{1}{\sqrt{m}}$ factor in the parameterization with $\frac{1}{\sqrt{2m}}$. This is called the "doubling trick" (Chizat et al., 2019; Zhang et al., 2020) and ensures that the network is identically zero $f(x; \theta_0) \equiv 0$ at initialization. We will explicitly state where we use the second scheme and otherwise will be using the first scheme.

The following assumptions will persist throughout the rest of the paper:

**Assumption 3.1.** *$\sigma$ is a $C^2$ function satisfying $\|\sigma'\|_\infty, \|\sigma''\|_\infty < \infty$.*

**Assumption 3.2.** *The inputs satisfy $\|x\|_2 \leq M$.*

The following assumptions will be used in most, but not all theorems. We will explicitly state when they apply.

**Assumption 3.3.** *The input domain $X$ is compact with strictly positive Borel measure $\rho$.*

**Assumption 3.4.** *$T_{K^\infty}$ is strictly positive, i.e., $\langle f, T_{K^\infty} f\rangle_2 > 0$ for $f \neq 0$.*

Most activation functions other than ReLU satisfy Assumption 3.1, such as Softplus $\sigma(x) = \ln(1 + e^x)$, Sigmoid $\sigma(x) = \frac{1}{1+e^{-x}}$, and Tanh $\sigma(x) = \frac{e^x - e^{-x}}{e^x + e^{-x}}$. Assumption 3.2 is a mild assumption which is satisfied for instance for RGB images and has been commonly used (Du et al., 2019b;a; Oymak & Soltanolkotabi, 2020). Assumption 3.3 is so that Mercer's decomposition holds, which

is often assumed implicitly. Assumption 3.4 is again a mild assumption that is satisfied for a broad family of parameter initializations (e.g. Gaussian) anytime $\sigma$ is not a polynomial function, as we will show in Appendix G. Assumption 3.4 is not strictly necessary but it simplifies the presentation by ensuring $T_{K\infty}$ has no zero eigenvalues.

We will track most constants that depend on parameters of our theorems such as $M$, the activation function $\sigma$, and the target function $f^*$. However, constants appearing in concentration inequalities such as Hoeffding's or Bernstein's inequality or constants arising from $\delta/2$ or $\delta/3$ arguments will not be tracked. We will reserve $c, C > 0$ for untracked constants whose precise meaning can vary from statement to statement. In the proofs in the appendix it will be explicit which constants are involved.

## 3.1 Underparameterized Regime

Our main result compares the dynamics of the residual $r_t(x) = f(x; \theta_t) - f^*(x)$ to that of $\exp(-T_{K\infty}t)r_0$ in the underparameterized setting. Note that $\langle \exp(-T_{K\infty}t)r_0, \phi_i \rangle_2 = \exp(-\sigma_i t)\langle r_0, \phi_i \rangle_2$, thus $\exp(-T_{K\infty}t)r_0$ learns the eigenfunctions $\phi_i$ of $T_{K\infty}$ at rate $\sigma_i$. Therefore $\exp(-T_{K\infty}t)r_0$ exhibits a bias to learn the eigenfunctions of $T_{K\infty}$ corresponding to large eigenvalues more quickly. To our knowledge no one has been able to rigorously relate the dynamics in function space of the residual $r_t$ to $\exp(-T_{K\infty}t)r_0$, although that seems to be what is suggested by Ronen et al. (2019); Basri et al. (2020). The existing works we are aware of (Arora et al., 2019; Basri et al., 2020; Cao et al., 2020) characterize the bias of the empirical residual primarily in the heavily overparameterized regime (Cao et al. (2020) stands out as requiring wide but not necessarily overparameterized networks). By contrast, we characterize the bias of the whole residual in the underparameterized regime.

**Theorem 3.5.** *Assume that Assumptions 3.3 and 3.4 hold. Let $P_k$ be the orthogonal projection in $L^2$ onto $span\{\phi_1, \ldots, \phi_k\}$ and let $D := 3 \max\{|\sigma(0)|, M \|\sigma'\|_\infty, \|\sigma'\|_\infty, 1\}$. If we are doing the doubling trick set $S' = 0$ and otherwise set $S' = O\left(\sqrt{\tilde{O}(d) + \log(c/\delta)}\right)$, $\quad S = \|f^*\|_\infty + S'$.*

*Also let $T > 0$. Assume $m \geq D^2 \|y\|^2_{\mathbb{R}^n} T^2$, and $m \geq O(\log(c/\delta) + \tilde{O}(d)) \max\{T^2, 1\}$. Then with probability at least $1 - \delta$ we have that for all $t \leq T$ and $k \in \mathbb{N}$*

$$\|P_k(r_t - \exp(-T_{K\infty}t)r_0)\|_2 \leq \frac{1 - \exp(-\sigma_k t)}{\sigma_k} \tilde{O}\left(S[1 + tS]\frac{\sqrt{d}}{\sqrt{m}} + S(1 + T)\frac{\sqrt{p}}{\sqrt{n}}\right)$$

*and*

$$\|r_t - \exp(-T_{K\infty}t)r_0\|_2 \leq t\tilde{O}\left(S[1 + tS]\frac{\sqrt{d}}{\sqrt{m}} + S(1 + T)\frac{\sqrt{p}}{\sqrt{n}}\right).$$

Theorem 3.5 will be proved in Appendix C. The proof uses the uniform deviation bounds for the $NTK$ to bound $T_n - T_n^s$ and tools from empirical process theory to show convergence of $T_n$ to $T_{K\infty}$ uniformly over a class of functions corresponding to networks with bounded parameter norms.

To interpret the results, we observe that to track the dynamics for eigenfunctions corresponding to eigenvalue $\sigma_k$ and above, the expression under the $\tilde{O}$ needs to be small relative to $\frac{1}{\sigma_k}$. Thus the bias towards learning the eigenfunctions corresponding to large eigenvalues appears more pronounced. When $t = \log(\|r_0\|_2/\epsilon)/\sigma_k$, we have that $\|P_k \exp(-T_{K\infty}t)r_0\|_2 \leq \epsilon$. Thus by applying this stopping time we get that to learn the eigenfunctions corresponding to eigenvalue $\sigma_k$ and above up to $\epsilon$ accuracy we need $\frac{t^2}{\sqrt{m}} \lesssim \epsilon$ and $\frac{t^2\sqrt{p}}{\sqrt{n}} \lesssim \epsilon$ which translates to $m \gtrsim \sigma_k^{-4}\epsilon^{-2}$ and $n \gtrsim p\sigma_k^{-4}\epsilon^{-2}$. In typical NTK works the width $m$ needs to be polynomially large relative to the number of samples $n$, where by contrast here the width depends on the inverse of the eigenvalues for the relevant components of the target function. From an approximation point-of-view this makes sense; the more complicated the target function the more expressive the model must be. We believe future works can adopt more precise requirements on the width $m$ that do not require growth relative to the number of samples $n$. To further illustrate the scaling of the parameters required by Theorem 3.5, we can apply Theorem 3.5 for an appropriate stopping time to get a bound on the test error.

**Corollary 3.6.** *Assume Assumptions 3.3 and 3.4 hold. Suppose that $f^* = O(1)$ and assume we are performing the doubling trick where $f_0 \equiv 0$ so that $r_0 = -f^*$. Let $k \in \mathbb{N}$ and let $P_k$ be*

*the orthogonal projection onto* $span\{\phi_1, \ldots, \phi_k\}$. *Set* $t = \frac{\log(\sqrt{2}\|P_k f^*\|_2/\epsilon^{1/2})}{\sigma_k}$ *Then we have that* $m = \tilde{\Omega}(\frac{d}{\epsilon\sigma_k^4})$ *and* $n = \tilde{\Omega}\left(\frac{p}{\sigma_k^4\epsilon}\right)$ *suffices to ensure with probability at least* $1 - \delta$

$$\frac{1}{2}\|r_t\|_2^2 \le 2\epsilon + 2\|(I - P_k)f^*\|_2^2.$$

If one specialized to the case where $f^*$ is a finite sum of eigenfunctions of $T_{K^\infty}$ (when the data is uniformly distributed on the sphere $S^{d-1}$ and the network weights are from a rotation invariant distribution this corresponds to a finite sum of spherical harmonics, which in $d = 2$ is equivalently a bandlimited function) one can choose $k$ such that $\|(I - P_k)f^*\|_2^2 = 0$. It is interesting to note that in this special case gradient flow with early stopping achieves essentially the same rates with respect to $m$ and $n$ (up to constants and logarithms) as the estimated network in the classical approximation theory paper by Barron (1994). It is also interesting to note that the approximation results by Barron (1994) depend on the decay in frequency domain of the target function $f^*$ via their constant $C_{f^*}$, and similarly for us the constant $1/\sigma_k^4$ grows with the bandwidth of the target function in the case of uniform distribution on the sphere $S^1$ which we mentioned parenthetically above.

While in Theorem 3.5 we compared the dynamics of $r_t$ against that of $\exp(-T_{K^\infty}t)r_0$, the damped deviations equation given by Lemma 2.3 enables you to compare against $\exp(-T_K t)r_0$ for an arbitrary kernel $K$. There are other natural choices for $K$ besides $K = K^\infty$, the most obvious being $K = K_0$. In Appendix C.8 we prove a version of Theorem 3.5 where $K = K_0$ and $\theta_0$ is an arbitrary deterministic parameter initialization. This could be interesting in scenarios where the parameters are initialized from a pretrained network or one has a priori knowledge that informs the selection of $\theta_0$. One could let $K$ be the kernel corresponding to some subset of parameters, such as the random feature kernel (Rahimi & Recht, 2008b) corresponding to the outer layer. This would compare the dynamics of training all layers to that of training a subset of the parameters. If one wanted to account for adaptations of the kernel $K_t$ one could try to set $K = K_{t_0}$ for some $t_0 > 0$. However since $\theta_{t_0}$ depends on the training data it is not obvious how one could produce a bound for $T_n^s - K_{t_0}$. Nevertheless we leave the suggestion open as a possibility for future work.

## 3.2 OVERPARAMETERIZED REGIME

Once one has deviation bounds for the $NTK$ so that the quantity $\|G^\infty - G_s\|_{op}$ is controlled, the damped deviations equation (Lemma 2.1) allows one to control the dynamics of the empirical risk. In this section we will demonstrate three such results that follow from this approach. The following is our analog of Theorem 4.1 from Arora et al. (2019) in our setting, proved in Appendix E. The result demonstrates that when the network is heavily overparameterized, the dynamics of the residual $\hat{r}_t$ follow the NTK regime dynamics $\exp(-G^\infty t)\hat{r}_0$.

**Theorem 3.7.** *Assume* $m = \tilde{\Omega}(dn^5\epsilon^{-2}\lambda_n(H^\infty)^{-4})$ *and* $m \ge O(\log(c/\delta) + \tilde{O}(d))$ *and* $f^* = O(1)$. *Assume we are performing the doubling trick so that* $\hat{r}_0 = -y$. *Let* $v_1, \ldots, v_n$ *denote the eigenvectors of* $G^\infty$ *normalized to have unit L2 norm* $\|v_i\|_2 = 1$. *Then with probability at least* $1 - \delta$

$$\hat{r}_t = \exp(-G^\infty t)(-y) + \delta(t),$$

*where* $\sup_{t\ge 0}\|\delta(t)\|_2 \le \epsilon$. *In particular*

$$\|\hat{r}_t\|_2 = \sqrt{\sum_{i=1}^n \exp(-2\lambda_i t)|\langle y, v_i\rangle_2|^2} \pm \epsilon.$$

In the work of Arora et al. (2019) the requirement is $m = \Omega(\frac{n^7}{\lambda_n(H^\infty)^4\kappa^2\delta^4\epsilon^2})$ and $\kappa = O(\frac{\epsilon\delta}{\sqrt{n}})$ where $w_\ell \sim N(0, \kappa^2 I)$ (not to be confused with our definition of $\kappa := \max_x K^\infty(x, x)$). By contrast our weights have unit variance, which for Gaussian initialization corresponds to $w_\ell \sim N(0, I)$. They require $\kappa$ to be small to ensure the neural network is small in magnitude at initializeation. To achieve the same effect we can perform antisymmetric initializeation to ensure the network is equivalently 0 at initializeation. Our overparameterization requirement ignoring logarithmic factors is smaller by a factor of $\frac{n^2}{d\delta^4}$. Again due to the different settings we do not claim superiority over this work.

The following is our analog of Theorem 4 by Su & Yang (2019) proved in Appendix F. This shows that when the target function has a compact representation in terms of eigenfunctions of $T_{K^\infty}$, a more moderate overparametrization is sufficient to approximately solve the ERM problem.

**Theorem 3.8.** *Assume Assumptions 3.3 and 3.4 hold. Furthermore assume $m = \tilde{\Omega}\left(\epsilon^{-2}dT^2 \|f^*\|_\infty^2 (1 + T \|f^*\|_\infty)^2\right)$ where $T > 0$ is a time parameter and $m \geq O(\log(c/\delta) + \tilde{O}(d))$ and $n \geq \frac{128\kappa^2 \log(2/\delta)}{(\sigma_k - \sigma_{k+1})^2}$. Also assume $f^* \in L^\infty(X) \subset L^2(X)$ and let $P^{T_{K^\infty}}$ be the orthogonal projection onto the eigenspaces of $T_{K^\infty}$ corresponding to the eigenvalue $\alpha \in \sigma(T_{K^\infty})$ and higher. Assume that $\left\|(I - P^{T_{K^\infty}})f^*\right\|_\infty \leq \epsilon'$ for some $\epsilon' \geq 0$. Pick $k$ so that $\sigma_k = \alpha$ and $\sigma_{k+1} < \alpha$, i.e. $k$ is the index of the last repeated eigenvalue corresponding to $\alpha$ in the ordered sequence $\{\sigma_i\}_i$. Also assume we are performing the doubling trick so that $\hat{r}(0) = -y$. Then we have with probability at least $1 - 3\delta$ over the sampling of $x_1, \ldots, x_n$ and $\theta_0$ that for $t \leq T$*

$$\|\hat{r}_t\|_{\mathbb{R}^n} \leq \exp(-\lambda_k t) \|y\|_{\mathbb{R}^n} + \frac{4\kappa \|f^*\|_2 \sqrt{10 \log(2/\delta)}}{(\sigma_k - \sigma_{k+1})\sqrt{n}} + 2\epsilon' + \epsilon.$$

Su & Yang (2019) have $\|f^*\|_2 \leq \|f^*\|_\infty \leq 1$, $\kappa \leq \frac{1}{2}$ and they treat $d$ as a constant. Taking these into account we do not see the overparameterization requirements or bounds of either work being superior to the other. From Theorem 3.8, setting $\epsilon = \frac{4\kappa\|f^*\|_2\sqrt{10\log(2/\delta)}}{(\sigma_k - \sigma_{k+1})\sqrt{n}}$ and $\epsilon' = 0$ we immediately get the analog of Corollary 2 in the work of Su & Yang (2019). This explains how in the special case that the target function is a finite sum of eigenfunctions of $T_{K^\infty}$, the width $m$ and the number of samples $n$ can grow at the same rate, up to logarithms, and still solve the ERM problem. This is an ERM guarantee for $m = \tilde{\Omega}(n)$ and thus attains moderate overparameterization.

**Corollary 3.9.** *Assume Assumptions 3.3 and 3.4 hold. Furhtermore assume $m = \tilde{\Omega}\left(\frac{n(\sigma_k - \sigma_{k+1})^2 d \|f^*\|_\infty^2 (1 + \lambda_k^{-1}\|f^*\|_\infty)^2}{\kappa^2 \|f^*\|_2^2 \lambda_k^2}\right)$ $m \geq O(\log(c/\delta) + \tilde{O}(d)$ $n \geq \frac{128\kappa^2 \log(2/\delta)}{(\sigma_k - \sigma_{k+1})^2}$. Let $f^*$, $P^{T_{K^\infty}}$, and $k$ be the same as in the hypothesis of Theorem 3.8. Furthermore assume that $\left\|(I - P^{T_{K^\infty}})f^*\right\|_\infty = 0$. Also assume we are performing the doubling trick so that $\hat{r}(0) = -y$. Set $T = \log(\sqrt{n} \|\hat{r}(0)\|_{\mathbb{R}^n})/\lambda_k$. Then we have with probability at least $1 - 3\delta$ over the sampling of $x_1, \ldots, x_n$ and $\theta_0$ that for $t \leq T$*

$$\|\hat{r}_t\|_{\mathbb{R}^n} \leq \exp(-\lambda_k t) \|y\|_{\mathbb{R}^n} + \frac{8\kappa \|f^*\|_2 \sqrt{10 \log(2/\delta)}}{(\sigma_k - \sigma_{k+1})\sqrt{n}}.$$

Note Su & Yang (2019) are training only the hidden layer of a ReLU network by gradient descent, by contrast we are training both layers with biases of a network with smooth activations by gradient flow. For Corollary 2 by Su & Yang (2019) they have the overparameterization requirement $m \gtrsim n \log n \left(\frac{1}{\lambda_k^4} + \frac{\log^4 n \log^2(1/\delta)}{(\lambda_k - \lambda_{k+1})^2 n^2 \lambda_k^4}\right)$. Thus both bounds scale like $\frac{n}{\lambda_k^4}$. Our bound has the extra factor $(\sigma_k - \sigma_{k+1})^2$ in front which could make it appear smaller at first glance but their Theorem 4 is strong enough to include this factor in the corollary they just chose not to. Thus we view both overparameterization requirements as comparable with neither superior to the other.

# 4 CONCLUSION AND FUTURE DIRECTIONS

The damped deviations equation allows one to compare the dynamics when optimizing the squared error to that of an arbitrary kernel regression. We showed how this simple equation can be used to track the dynamics of the test residual in the underparameterized regime and extend existing results in the overparameterized setting. In the underparameterized setting the neural network learns eigenfunctions of the integral operator $T_{K^\infty}$ determined by the Neural Tangent Kernel at rates corresponding to their eigenvalues. In the overparameterized setting the damped deviations equation combined with NTK deviation bounds allows one to track the dynamics of the empirical risk. In this fashion we extended existing work to the setting of a network with smooth activations where all parameters are trained as in practice. We hope damped deviations offers a simple interpretation of the MSE dynamics and encourages others to compare against other kernels in future work.

ACKNOWLEDGMENTS

Benjamin Bowman was at the Max Planck Institute for Mathematics in the Sciences while working on parts of this project. This project has received funding from the European Research Council (ERC) under the European Union's Horizon 2020 research and innovation programme (grant agreement no 757983).

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

## APPENDIX

## A ADDITIONAL NOTATIONS

We let $[k] := \{1, 2, 3, \ldots, k\}$. For a set $A$ we let $|A|$ denote its cardinality. $\|\bullet\|_F$ denotes the Frobenius norm for matrices, and for two matrices $A, B \in \mathbb{R}^{n \times m}$ we will let $\langle A, B \rangle = Tr(A^T B) = \sum_{i=1}^{n} \sum_{j=1}^{m} A_{i,j} B_{i,j}$ denote the Frobenius or entry-wise inner product. We will let $B_R := \{x : \|x\|_2 \leq R\}$ to be the Euclidean ball of radius $R > 0$.

## B NTK DEVIATION AND PARAMETER NORM BOUNDS

Let $\Gamma > 1$. At the end of this section we will prove a high probability bound of the form

$$\sup_{(x,x') \in B_M \times B_M} |K_t(x, x') - K^\infty(x, x')| = \tilde{O}\left(\frac{\sqrt{d}}{\sqrt{m}}\left[1 + t\Gamma^3 \|\hat{r}(0)\|_{\mathbb{R}^n}\right]\right).$$

Ideally we would like to use the results in Huang & Yau (2020) where they prove for a deep feed-forward network without biases:

$$\sup_{1 \leq i,j \leq n} |K_t(x_i, x_j) - K^\infty(x_i, x_j)| = \tilde{O}\left(\frac{t^2}{m} + \frac{1}{\sqrt{m}}\right).$$

However there are three problems that prevent this. The first is that the have a constant under the $\tilde{O}$ above that depends on the training data. Specifically their Assumption 2.2 requires that the smallest singular value of the data matrix $[x_{\alpha_1}, \ldots, x_{\alpha_r}]$ is greater than $c_r > 0$ where $1 \leq \alpha_1, \ldots, \alpha_r \leq n$ are arbitrary distinct indices. As you send the number of samples to infinity you will have $c_r \to 0$, thus it is not clear how the bound will scale in the large sample regime. The second is that their bound only holds on the training data, whereas we need a bound that is uniform over all inputs. The final one is their network does not have biases. In the following section we will overcome these issues. The main difference between our argument and theirs is how we prove convergence at initialization. In their argument for convergence at initialization they make repeated use of a Gaussian conditioning lemma as they pass through the layers, and this relies on their Assumption 2.2. By contrast we will use Lipschitzness of the NTK and convergence over an $\epsilon$ net to prove convergence at initialization. As we see it, our deviation bounds for the time derivative $\partial_t K_t$ are proved in a very similar fashion and the rest of the argument is very much inspired by their approach.

At the time of submission of this manuscript, we were made aware of the work by Liu et al. (2020) that provides an alternative uniform NTK deviation bound by providing a uniform bound on the operator norm of the Hessian. Their work is very nice, and it opens the door to extending the results of this paper to the other architectures they consider. Nevertheless, we proceed with our original analysis below.

This section is conceptually simple but technical. We will take care to outline the high level structure of each section to prevent the technicalities from obfuscating the overall simplicity of the approach. Our argument runs through the following steps:

- Control parameter norms throughout training.
- Bound the Lipschitz constant of the $NTK$ with respect to spatial inputs.
- Use concentration of subexponential random variables (Bernstein's inequality) to show that $|K^\infty(z') - K_0(z')| = \tilde{O}(1/\sqrt{m})$ (roughly) for all $z'$ in an $\epsilon$ net of the spatial domain. Combine with the Lipschitz property of the $NTK$ to show convergence over *all* inputs, namely $\sup_{z \in B_M} |K^\infty(z) - K_0(z)| = \tilde{O}(1/\sqrt{m})$ (roughly).
- Produce the bound $\sup_{z \in B_M \times B_M} |\partial_t K_t(z)| = \tilde{O}(1/\sqrt{m})$ (roughly).
- Conclude that $\sup_{z \in B_M \times B_M} |K_t(z) - K_0(z)| = \tilde{O}(t/\sqrt{m})$ (roughly).

### B.1 IMPORTANT EQUATIONS

The following list contains the equations that are relevant for this section. We found it easier to read the following proofs by keeping these equations on a separate piece of paper or in a separate tab. We write $a \otimes x = ax^T$. Also throughout this section the training data will be considered fixed and thus the randomness of the inputs is not relevant to this section. The randomness will come entirely from the parameter initialization $\theta_0$.

$$f(x; \theta) = \frac{1}{\sqrt{m}} a^T \sigma(Wx + b) + b_0$$

$$x^{(1)} := \frac{1}{\sqrt{m}} \sigma(Wx + b)$$

$$\sigma_1'(x) := diag(\sigma'(Wx + b))$$
$$\sigma_1''(x) := diag(\sigma''(Wx + b))$$

$$\partial_a f(x; \theta) = \frac{1}{\sqrt{m}} \sigma(Wx + b) = x^{(1)}$$

$$\partial_W f(x; \theta) = \frac{1}{\sqrt{m}} \sigma_1'(x) a \otimes x$$

$$\partial_b f(x; \theta) = \frac{1}{\sqrt{m}} \sigma_1'(x) a$$

$$\partial_{b_0} f(x; \theta) = 1$$

$$\partial_t a = -\frac{1}{n} \sum_{i=1}^{n} \hat{r}_i x_i^{(1)}$$

$$\partial_t W = -\frac{1}{n} \sum_{i=1}^{n} \hat{r}_i \frac{1}{\sqrt{m}} \sigma_1'(x_i) a \otimes x_i$$

$$\partial_t b = -\frac{1}{n} \sum_{i=1}^{n} \hat{r}_i \frac{1}{\sqrt{m}} \sigma_1'(x_i) a$$

$$\partial_t b_0 = -\frac{1}{n} \sum_{i=1}^{n} \hat{r}_i$$

$$\partial_t x^{(1)} = \partial_t \frac{1}{\sqrt{m}} \sigma(Wx + b) = \frac{1}{\sqrt{m}} \sigma_1'(x) [\partial_t Wx + \partial_t b]$$

$$= -\frac{1}{n} \sum_{i=1}^{n} \hat{r}_i \left[ \frac{1}{\sqrt{m}} \sigma_1'(x) \sigma_1'(x_i) \frac{a}{\sqrt{m}} \right] [\langle x, x_i \rangle_2 + 1]$$

$$\partial_t \sigma_1'(x) = \partial_t \sigma'(Wx + b) = \sigma_1''(x) diag(\partial_t Wx + \partial_t b)$$

$$= -\frac{1}{n} \sum_{i=1}^{n} \hat{r}_i \frac{1}{\sqrt{m}} \sigma_1''(x) \sigma_1'(x_i) diag(a) [\langle x, x_i \rangle_2 + 1]$$

## B.2 A PRIORI PARAMETER NORM BOUNDS

In this section we will provide bounds for the following quantities:

$$\xi(t) = \max\{ \frac{1}{\sqrt{m}} \|W(t)\|_{op} , \frac{1}{\sqrt{m}} \|b(t)\|_2 , \frac{1}{\sqrt{m}} \|a(t)\|_2 , 1\}$$

$$\tilde{\xi}(t) = \max\{ \max_{\ell \in [m]} \|w_\ell(t)\|_2 , \|a(t)\|_\infty , \|b(t)\|_\infty , 1\}.$$

Here $w_\ell = W_{\ell,:} \in \mathbb{R}^d$ is the vector of input weights to the $\ell$th unit. These quantities appear repeatedly throughout the rest of the proofs of this section and thus need to be controlled. The parameter norm bounds will also be useful for the purpose of the covering number argument in Section C.6. This section is broken down as follows:

- Prove Lemma B.1
- Bound $\xi(t)$
- Bound $\tilde{\xi}(t)$

The time derivatives throughout will repeatedly be of the form $\frac{1}{n} \sum_{i=1}^{n} \hat{r}_i(t) v_i$. Lemma B.1 provides a simple bound that we will use over and over again.

**Lemma B.1.** *Let $\|\bullet\|$ be any norm over a vector space $V$. Then for any $v_1, \ldots, v_n \in V$ we have*

$$\left\| \frac{1}{n} \sum_{i=1}^{n} \hat{r}_i(t) v_i \right\| \leq \max_{i \in [n]} \|v_i\| \|\hat{r}(t)\|_{\mathbb{R}^n} \leq \max_{i \in [n]} \|v_i\| \|\hat{r}(0)\|_{\mathbb{R}^n} .$$

*Proof.* Note that

$$\left\| \frac{1}{n} \sum_{i=1}^{n} \hat{r}_i(t) v_i \right\| \leq \frac{1}{n} \sum_{i=1}^{n} |\hat{r}_i(t)| \|v_i\| \leq \max_{i \in [n]} \|v_i\| \frac{1}{n} \sum_{i=1}^{n} |\hat{r}_i(t)|$$

$$\leq \max_{i \in [n]} \|v_i\| \frac{1}{\sqrt{n}} \|\hat{r}(t)\|_2 = \max_{i \in [n]} \|v_i\| \|\hat{r}(t)\|_{\mathbb{R}^n} \leq \max_{i \in [n]} \|v_i\| \|\hat{r}(0)\|_{\mathbb{R}^n} ,$$

where the last inequality follows from $\|\hat{r}(t)\|_{\mathbb{R}^n} \leq \|\hat{r}(0)\|_{\mathbb{R}^n}$ from gradient flow. □

We now proceed to bound $\xi(t)$.

**Lemma B.2.** *Let* $\xi(t) = \max\{\frac{1}{\sqrt{m}} \|W(t)\|_{op}, \frac{1}{\sqrt{m}} \|b(t)\|_2, \frac{1}{\sqrt{m}} \|a(t)\|_2, 1\}$ *and*

$$D := 3 \max\{|\sigma(0)|, M \|\sigma'\|_\infty, \|\sigma'\|_\infty, 1\}.$$

*Then for any initial conditions* $W(0)$, $b(0)$, $a(0)$ *we have for all* $t$

$$\xi(t) \le \exp\left(\frac{D}{\sqrt{m}} \int_0^t \|\hat{r}(s)\|_{\mathbb{R}^n} ds\right) \xi(0) \le \exp\left(\frac{D}{\sqrt{m}} \|\hat{r}(0)\|_{\mathbb{R}^n} t\right) \xi(0).$$

*Proof.* Recall that

$$\partial_t a = -\frac{1}{n} \sum_{i=1}^n \hat{r}_i x_i^{(1)}$$

$$\partial_t W = -\frac{1}{n} \sum_{i=1}^n \hat{r}_i \frac{1}{\sqrt{m}} \sigma_1'(x_i) a \otimes x_i$$

$$\partial_t b = -\frac{1}{n} \sum_{i=1}^n \hat{r}_i \frac{1}{\sqrt{m}} \sigma_1'(x_i) a.$$

We will show that each of the above derivatives is $\lesssim \|\hat{r}(t)\|_{\mathbb{R}^n} \xi(t)$ then apply Grönwall's inequality. By Lemma B.1 it suffices to show that the terms multiplied by $\hat{r}_i$ in the above sums are $\lesssim \xi(t)$. First we note that

$$\left\| x_i^{(1)} \right\|_2 = \left\| \frac{1}{\sqrt{m}} \sigma(W x_i + b) \right\|_2 \le |\sigma(0)| + \frac{1}{\sqrt{m}} \|\sigma'\|_\infty \|W x_i + b\|_2$$

$$\le |\sigma(0)| + \frac{1}{\sqrt{m}} \|\sigma'\|_\infty \left[ \|W\|_{op} \|x_i\|_2 + \|b\|_2 \right] \le |\sigma(0)| + \frac{1}{\sqrt{m}} \|\sigma'\|_\infty \left[ \|W\|_{op} M + \|b\|_2 \right]$$

$$\le D\xi.$$

Second we have that

$$\left\| \frac{1}{\sqrt{m}} \sigma_1'(x_i) a \otimes x_i \right\|_{op} = \left\| \frac{1}{\sqrt{m}} \sigma_1'(x_i) a \right\|_2 \|x_i\|_2 \le M \|\sigma'\|_\infty \frac{1}{\sqrt{m}} \|a\|_2 \le D\xi.$$

Finally we have that

$$\left\| \frac{1}{\sqrt{m}} \sigma_1'(x_i) a \right\| \le \|\sigma'\|_\infty \frac{1}{\sqrt{m}} \|a\|_2 \le D\xi.$$

Thus by Lemma B.1 and the above bounds we have

$$\|\partial_t W(t)\|_{op}, \|\partial_t a(t)\|_2, \|\partial_t b(t)\|_2 \le D \|\hat{r}(t)\|_{\mathbb{R}^n} \xi(t).$$

Let $v(t)$ be a placeholder for one of the functions $\frac{1}{\sqrt{m}} a(t), \frac{1}{\sqrt{m}} W(t), \frac{1}{\sqrt{m}} b(t)$ with corresponding norm $\|\bullet\|$. Then we have that

$$\|v(t)\| \le \|v(0)\| + \|v(t) - v(0)\| = \|v(0)\| + \left\| \int_0^t \partial_s v(s) ds \right\|$$

$$\le \|v(0)\| + \int_0^t \|\partial_s v(s)\| ds \le \xi(0) + \int_0^t \frac{\|\hat{r}(s)\|_{\mathbb{R}^n}}{\sqrt{m}} D\xi(s) ds.$$

This inequality holds for any of the three choices of $v$ thus we get that

$$\xi(t) \le \xi(0) + \int_0^t \frac{\|\hat{r}(s)\|_{\mathbb{R}^n}}{\sqrt{m}} D\xi(s) ds.$$

Therefore by Grönwall's inequality we get that

$$\xi(t) \le \exp\left(\frac{D}{\sqrt{m}} \int_0^t \|\hat{r}(s)\|_{\mathbb{R}^n} ds\right) \xi(0) \le \exp\left(\frac{D}{\sqrt{m}} \|\hat{r}(0)\|_{\mathbb{R}^n} t\right) \xi(0).$$

$\square$

We will now bound $\tilde{\xi}(t)$ using essentially the same argument as in the previous lemma.

**Lemma B.3.** *Let* $\tilde{\xi}(t) = \max\{\max_{\ell \in [m]} \|w_\ell(t)\|_2, \|a(t)\|_\infty, \|b(t)\|_\infty, 1\}$ *and*

$$D = 3 \max\{|\sigma(0)|, M \|\sigma'\|_\infty, \|\sigma'\|_\infty, 1\}.$$

*Then for any initial conditions* $W(0)$, $b(0)$, $a(0)$ *we have for all* $t$

$$\tilde{\xi}(t) \le \exp\left(\frac{D}{\sqrt{m}} \int_0^t \|\hat{r}(s)\|_{\mathbb{R}^n} \, ds\right) \tilde{\xi}(0) \le \exp\left(\frac{D}{\sqrt{m}} \|\hat{r}(0)\|_{\mathbb{R}^n} t\right) \tilde{\xi}(0).$$

*Proof.* The proof is basically the same as Lemma B.2. We have that

$$\partial_t w_\ell = -\frac{1}{n} \sum_{i=1}^n \hat{r}_i \frac{a_\ell}{\sqrt{m}} \sigma'(\langle w_\ell, x_i \rangle_2 + b_\ell) x_i.$$

Now note

$$\left\| \frac{a_\ell}{\sqrt{m}} \sigma'(\langle w_\ell, x_i \rangle_2 + b_\ell) x_i \right\|_2 \le \frac{1}{\sqrt{m}} \|a\|_\infty \|\sigma'\|_\infty M \le \frac{D}{\sqrt{m}} \tilde{\xi}.$$

Thus by Lemma B.1 we have that

$$\|\partial_t w_\ell(t)\|_2 \le \frac{D}{\sqrt{m}} \|\hat{r}(t)\|_{\mathbb{R}^n} \tilde{\xi}(t).$$

On the other hand

$$\partial_t a = -\frac{1}{n} \sum_{i=1}^n \hat{r}_i x_i^{(1)}$$

with

$$\left\| x_i^{(1)} \right\|_\infty = \left\| \frac{1}{\sqrt{m}} \sigma(Wx_i + b) \right\|_\infty \le \frac{1}{\sqrt{m}} \left[ |\sigma(0)| + \|\sigma'\|_\infty \|Wx_i + b\|_\infty \right]$$

$$\le \frac{1}{\sqrt{m}} \left[ |\sigma(0)| + \|\sigma'\|_\infty \left( M \max_\ell \|w_\ell\|_2 + \|b\|_\infty \right) \right] \le \frac{D}{\sqrt{m}} \tilde{\xi}.$$

Thus again by Lemma B.1 we have

$$\|\partial_t a(t)\|_\infty \le \frac{D}{\sqrt{m}} \|\hat{r}(t)\|_{\mathbb{R}^n} \tilde{\xi}(t).$$

Finally we have

$$\partial_t b = -\frac{1}{n} \sum_{i=1}^n \hat{r}_i \frac{1}{\sqrt{m}} \sigma_1'(x_i) a$$

with

$$\left\| \frac{1}{\sqrt{m}} \sigma_1'(x_i) a \right\|_\infty \le \frac{1}{\sqrt{m}} \|\sigma'\|_\infty \|a\|_\infty \le \frac{D}{\sqrt{m}} \tilde{\xi}.$$

Again applying Lemma B.1 one last time we get

$$\|\partial_t b(t)\|_\infty \le \frac{D}{\sqrt{m}} \|\hat{r}(t)\|_{\mathbb{R}^n} \tilde{\xi}(t).$$

Therefore by the same argument as in Lemma B.2 using Grönwall's inequality we get that

$$\tilde{\xi}(t) \le \exp\left(\frac{D}{\sqrt{m}} \int_0^t \|\hat{r}(s)\|_{\mathbb{R}^n} \, ds\right) \tilde{\xi}(0) \le \exp\left(\frac{D}{\sqrt{m}} \|\hat{r}(0)\|_{\mathbb{R}^n} t\right) \tilde{\xi}(0).$$

$\square$

### B.3  $NTK$ Is Lipschitz With Respect To Spatial Inputs

The $NTK$ being Lipschitz with respect to spatial inputs is essential to our proof. The Lipschitz property means that to show convergence uniformly for *all* inputs it suffices to show convergence on an $\epsilon$ net of the spatial domain. Since the parameters are changing throughout time, the Lipschitz constant of the $NTK$ will change throughout time. We will see that the Lipschitz constant depends on the quantities $\xi(t)$ and $\tilde{\xi}(t)$ from the previous Section B.2.

The $NTK$ $K_t(x, x')$ is a sum of terms of the form $g(x)^T g(x')$ where $g$ is one of the derivatives $\partial_a f(x; \theta_t), \partial_b f(x; \theta_t), \partial_W f(x; \theta_t), \partial_{b_0} f(x; \theta_t)$. Since $\partial_{b_0} f(x; \theta_t) \equiv 1$ this term can be ignored for the rest of the section. The upcomming Lemma B.4 shows that if $g$ is Lipschitz and bounded then $(x, x') \mapsto g(x)^T g(x')$ is Lipschitz. This lemma guides the structure of this section:

- Prove Lemma B.4
- Show that $\partial_a f(x; \theta), \partial_b f(x; \theta), \partial_W f(x; \theta)$ are bounded and Lipschitz
- Conclude the $NTK$ is Lipschitz

**Lemma B.4.** *Let $g : \mathbb{R}^k \to \mathbb{R}^l$ be L-Lipschitz with respect to the 2-norm, i.e.*

$$\|g(x) - g(z)\|_2 \leq L \|x - z\|_2$$

*and satisfy $\|g(x)\|_2 \leq M$ for all $x$ in some set $\mathcal{X}$. Then $K_g : \mathcal{X} \times \mathcal{X} \to \mathbb{R}$*

$$K_g(x, x') := g(x)^T g(x')$$

*is $ML$-Lipschitz with respect to the norm*

$$\|(x, x')\| := \|x\|_2 + \|x'\|_2.$$

*Proof.* We have

$$\begin{aligned}
|K_g(x, x') - K_g(z, z')| &= |g(x)^T g(x') - g(z)^T g(z')| \\
&= |g(x)^T (g(x') - g(z'))| + |(g(x) - g(z))^T g(z')| \\
&\leq \|g(x)\|_2 \|g(x') - g(z')\|_2 + \|g(x) - g(z)\|_2 \|g(z')\|_2 \\
&\leq ML \|x' - z'\|_2 + ML \|x - z\|_2 \leq ML \|(x, x') - (z, z')\|.
\end{aligned}$$

$\square$

By the previous Lemma B.4, to show that the $NTK$ is Lipschitz it suffices to show that $\partial_a f(x; \theta), \partial_b f(x; \theta), \partial_W f(x; \theta), \partial_{b_0} f(x; \theta)$ are bounded and Lipschitz. The following lemma bounds the norms of the derivatives $\partial_a f(x; \theta), \partial_W f(x; \theta), \partial_b f(x; \theta)$.

**Lemma B.5.** *Let $D = 3 \max\{|\sigma(0)|, M \|\sigma'\|_\infty, \|\sigma'\|_\infty, 1\}$ and*

$$\xi = \max\{\frac{1}{\sqrt{m}} \|W\|_{op}, \frac{1}{\sqrt{m}} \|b\|_2, \frac{1}{\sqrt{m}} \|a\|_2, 1\}.$$

*Then*

$$\|\partial_a f(x; \theta)\|_2, \|\partial_W f(x; \theta)\|_F, \|\partial_b f(x; \theta)\|_2 \leq D\xi.$$

*Proof.* We have

$$\begin{aligned}
\|\partial_a f(x; \theta)\|_2 &= \left\| \frac{1}{\sqrt{m}} \sigma(Wx + b) \right\|_2 \leq |\sigma(0)| + \frac{1}{\sqrt{m}} \|\sigma'\|_\infty \|Wx + b\|_2 \\
&\leq |\sigma(0)| + \frac{1}{\sqrt{m}} \|\sigma'\|_\infty \left[ \|W\|_{op} \|x\|_2 + \|b\|_2 \right] \\
&\leq |\sigma(0)| + \frac{1}{\sqrt{m}} \|\sigma'\|_\infty \left[ \|W\|_{op} M + \|b\|_2 \right] \leq D\xi,
\end{aligned}$$

$$\|\partial_W f(x; \theta)\|_F = \left\| \frac{1}{\sqrt{m}} \sigma'_1(x) a \otimes x \right\|_F = \left\| \frac{1}{\sqrt{m}} \sigma'_1(x) a \right\|_2 \|x\|_2 \leq \frac{M}{\sqrt{m}} \|\sigma'\|_\infty \|a\|_2 \leq D\xi,$$

$$\|\partial_b f(x;\theta)\|_2 = \left\|\frac{1}{\sqrt{m}}\sigma_1'(x)a\right\|_2 \le \frac{1}{\sqrt{m}}\|\sigma'\|_\infty \|a\|_2 \le D\xi.$$

$\square$

The following lemma demonstrates that $\partial_a f(x;\theta)$, $\partial_W f(x;\theta)$, and $\partial_b f(x;\theta)$ are Lipschitz as functions of the input $x$.

**Lemma B.6.** *Let*

$$\xi = \max\{\frac{1}{\sqrt{m}}\|W\|_{op}, \frac{1}{\sqrt{m}}\|b\|_2, \frac{1}{\sqrt{m}}\|a\|_2, 1\},$$

$$\tilde{\xi} = \max\{\max_{\ell\in[m]}\|w_\ell\|_2, \|a\|_\infty, \|b\|_\infty, 1\},$$

$$D' = \max\{\|\sigma'\|_\infty, M\|\sigma''\|_\infty, \|\sigma''\|_\infty\},$$

$$L = 2\xi\tilde{\xi}D'.$$

*Then $\partial_a f(x;\theta)$, $\partial_b f(x;\theta)$, $\partial_W f(x;\theta)$ are all L-Lipschitz with respect to the Euclidean norm $\|\bullet\|_2$. In symbols:*

$$\|\partial_a f(x;\theta) - \partial_a f(y;\theta)\|_2 \le L\|x-y\|_2$$
$$\|\partial_W f(x;\theta) - \partial_W f(y;\theta)\|_F \le L\|x-y\|_2$$
$$\|\partial_b f(x;\theta) - \partial_b f(y;\theta)\|_2 \le L\|x-y\|_2.$$

*Proof.* We have

$$\|\partial_a f(x;\theta) - \partial_a f(y;\theta)\|_2 = \left\|\frac{1}{\sqrt{m}}(\sigma(Wx+b) - \sigma(Wy+b))\right\|_2$$

$$\le \frac{1}{\sqrt{m}}\|\sigma'\|_\infty\|W(x-y)\|_2 \le \frac{1}{\sqrt{m}}\|\sigma'\|_\infty\|W\|_{op}\|x-y\|_2 \le L\|x-y\|_2,$$

$$\|\partial_W f(x;\theta) - \partial_W f(y;\theta)\|_F = \left\|\frac{1}{\sqrt{m}}\sigma_1'(x)a\otimes x - \frac{1}{\sqrt{m}}\sigma_1'(y)a\otimes y\right\|_F$$

$$\le \left\|\frac{1}{\sqrt{m}}\sigma_1'(x)a\otimes[x-y]\right\|_F + \left\|\frac{1}{\sqrt{m}}[\sigma_1'(x)a - \sigma_1'(y)a]\otimes y\right\|_F$$

$$\le \frac{1}{\sqrt{m}}\|\sigma_1'(x)a\|_2\|x-y\|_2 + \frac{1}{\sqrt{m}}\|[\sigma_1'(x)-\sigma_1'(y)]a\|_2\|y\|_2$$

$$\le \frac{1}{\sqrt{m}}\|\sigma'\|_\infty\|a\|_2\|x-y\|_2 + \frac{1}{\sqrt{m}}\|\sigma'(Wx+b)-\sigma'(Wy+b)\|_\infty\|a\|_2 M$$

$$\le \frac{1}{\sqrt{m}}\|\sigma'\|_\infty\|a\|_2\|x-y\|_2 + \frac{1}{\sqrt{m}}\|\sigma''\|_\infty\|W(x-y)\|_\infty\|a\|_2 M$$

$$\le \frac{1}{\sqrt{m}}\|\sigma'\|_\infty\|a\|_2\|x-y\|_2 + \frac{1}{\sqrt{m}}\|\sigma''\|_\infty\max_{\ell\in[m]}\|w_\ell\|_2\|x-y\|_2\|a\|_2 M$$

$$\le L\|x-y\|_2,$$

$$\|\partial_b f(x;\theta) - \partial_b f(y;\theta)\|_2 = \left\|\frac{1}{\sqrt{m}}\sigma_1'(x)a - \frac{1}{\sqrt{m}}\sigma_1'(y)a\right\|_2$$

$$\le \frac{1}{\sqrt{m}}\|\sigma'(Wx+b)-\sigma'(Wy+b)\|_\infty\|a\|_2$$

$$\le \frac{1}{\sqrt{m}}\|\sigma''\|_\infty\|W(x-y)\|_\infty\|a\|_2$$

$$\le \frac{1}{\sqrt{m}}\|\sigma''\|_\infty\max_{\ell\in[m]}\|w_\ell\|_2\|x-y\|_2\|a\|_2 \le L\|x-y\|_2.$$

$\square$

Finally we can prove that the Neural Tangent Kernel is Lipschitz.

**Theorem B.7.** *Consider the Neural Tangent Kernel*

$$K(x,y) = \langle \partial_a f(x;\theta), \partial_a f(y;\theta) \rangle_2 + \langle \partial_b f(x;\theta), \partial_b f(y;\theta) \rangle_2 + \langle \partial_W f(x;\theta), \partial_W f(y;\theta) \rangle + 1$$

*and let*

$$\xi = \max\{ \frac{1}{\sqrt{m}} \|W\|_{op}, \frac{1}{\sqrt{m}} \|b\|_2, \frac{1}{\sqrt{m}} \|a\|_2, 1 \},$$

$$\tilde{\xi} = \max\{ \max_{\ell \in [m]} \|w_\ell\|_2, \|a\|_\infty, \|b\|_\infty, 1 \},$$

$$D = 3 \max\{ |\sigma(0)|, M \|\sigma'\|_\infty, \|\sigma'\|_\infty, 1 \},$$

$$D' = \max\{ \|\sigma'\|_\infty, M \|\sigma''\|_\infty, \|\sigma''\|_\infty \}.$$

*Then the Neural Tangent Kernel is Lipschitz with respect to the norm*

$$\|(x,y)\| := \|x\|_2 + \|y\|_2$$

*with Lipschitz constant $L := 6DD'\xi^2 \tilde{\xi}$. In symbols:*

$$|K(x,y) - K(x',y')| \leq L \|(x,y) - (x',y')\|.$$

*Proof.* By Lemma B.5, we have that the gradients are bounded

$$\|\partial_a f(x;\theta)\|_2, \|\partial_W f(x;\theta)\|_F, \|\partial_b f(x;\theta)\|_2 \leq D\xi.$$

Also by Lemma B.6 the gradients are Lipschitz with Lipschitz constant $2\xi\tilde{\xi}D'$. Thus these two facts combined with Lemma B.4 tell us that each of the three terms $\langle \partial_a f(x;\theta), \partial_a f(y;\theta) \rangle$, $\langle \partial_b f(x;\theta), \partial_b f(y;\theta) \rangle$, and $\langle \partial_W f(x;\theta), \partial_W f(y;\theta) \rangle$ are individually Lipschitz with constant $(D\xi) \cdot (2\xi\tilde{\xi}D')$. Thus the Lipschitz constant of the $NTK$ itself is bounded by the sum of the 3 Lipschitz constants, for a total of $6DD'\xi^2\tilde{\xi}$. $\qquad\square$

Using that the $NTK$ at time zero $K_0(x,y)$ is Lipschitz we can prove that the analytical NTK $K^\infty = \mathbb{E}[K_0(x,y)]$ is Lipschitz. We will use this primarily as a qualitative statement, meaning that the estimate that we derive for the Lipschitz constant will not be used as it is not very explicit. Rather, in theorems where we use the fact that $K^\infty$ is Lipschitz we will simply take the Lipschitz constant of $K^\infty$ as an external parameter.

**Theorem B.8.** *Assume that $W_{i,j}(0) \sim \mathcal{W}$, $b_\ell(0) \sim \mathcal{B}$, $a_\ell(0) \sim \mathcal{A}$ are all i.i.d. zero-mean, unit variance subgaussian random variables. Let*

$$\xi(0) = \max\{ \frac{1}{\sqrt{m}} \|W(0)\|_{op}, \frac{1}{\sqrt{m}} \|b(0)\|_2, \frac{1}{\sqrt{m}} \|a(0)\|_2, 1 \},$$

$$\tilde{\xi}(0) = \max\{ \max_{\ell \in [m]} \|w_\ell(0)\|_2, \|a(0)\|_\infty, \|b(0)\|_\infty, 1 \},$$

$$D = 3 \max\{ |\sigma(0)|, M \|\sigma'\|_\infty, \|\sigma'\|_\infty, 1 \},$$

$$D' = \max\{ \|\sigma'\|_\infty, M \|\sigma''\|_\infty, \|\sigma''\|_\infty \}.$$

*Then the analytical Neural Tangent Kernel $K^\infty(x,y) = \mathbb{E}[K_0(x,y)]$ is Lipschitz with respect to the norm*

$$\|(x,y)\| := \|x\|_2 + \|y\|_2$$

*with Lipschitz constant $\leq 6DD'\mathbb{E}[\xi^2\tilde{\xi}] < \infty$. If one instead does the doubling trick then the same conclusion holds.*

*Proof.* First assume we are not doing the doubling trick. We note that

$$|K^\infty(x,y) - K^\infty(x',y')| = |\mathbb{E}[K_0(x,y)] - \mathbb{E}[K_0(x',y')]|$$

$$\leq \mathbb{E}|K_0(x,y) - K_0(x',y')| \leq 6DD'\mathbb{E}[\xi^2\tilde{\xi}] \|(x,y) - (x',y')\|$$

where the last line follows from the Lipschitzness of $K_0$ provided by Theorem B.7. Using that $\|W(0)\|_{op} \leq \|W(0)\|_F$ and the fact that the Euclidean norm of a vector with i.i.d. subgaussian entries is subgaussian (Vershynin, 2018, Theorem 3.1.1), we have that $\xi(0)$ and $\tilde{\xi}(0)$ are maximums of subgaussian random variables. Since a maximum of subgaussian random variables is subgaussian, we have that $\xi(0)$ and $\tilde{\xi}(0)$ are subgaussian. From the inequality $ab \leq \frac{1}{2}(a^2 + b^2)$ we get $\mathbb{E}[\xi^2\tilde{\xi}] \leq \frac{1}{2}\mathbb{E}[\xi^4] + \frac{1}{2}\mathbb{E}[\tilde{\xi}^2] < \infty$ since moments of subgaussian random variables are all finite. Since the doubling trick does not change the distribution of $K_0$, the same conclusion holds under that initialization scheme. $\qquad\square$

B.4  $NTK$ CONVERGENCE AT INITIALIZATION

In this section we prove that $\sup_{z \in B_M \times B_M} |K_0(z) - K^\infty(z)| = \tilde{O}(1/\sqrt{m})$. Our argument traces the following steps:

- Show that $K_0$ is sum of averages of $m$ independent subexponential random variables
- Use subexponential concentration to show that $\sup_{z' \in \Delta} |K_0(z') - K^\infty(z')| = \tilde{O}(1/\sqrt{m})$ for all $z'$ in an $\epsilon$ net $\Delta$ of $B_M \times B_M$
- Use that $K_0$ is Lipschitz and convergence over the epsilon net $\Delta$ to show that

$$\sup_{z \in B_M \times B_M} |K_0(z) - K^\infty(z)| = \tilde{O}(1/\sqrt{m}) \text{ (roughly)}$$

We recall the following definitions 2.5.6 and 2.7.5 from Vershynin (2018).

**Definition B.9.** *(Vershynin 2018) Let $Y$ be a random variable. Then we define the subgaussian norm of $Y$ to be*

$$\|Y\|_{\psi_2} = \inf\{t > 0 : \mathbb{E}\exp(Y^2/t^2) \leq 2\}$$

*If $\|Y\|_{\psi_2} < \infty$, then we say $Y$ is subgaussian.*

**Definition B.10.** *(Vershynin 2018) Let $Y$ be a random variable. Then we define the subexponential norm of $Y$ to be*

$$\|Y\|_{\psi_1} = \inf\{t > 0 : \mathbb{E}\exp(|Y|/t) \leq 2\}$$

*If $\|Y\|_{\psi_1} < \infty$, then we say $Y$ is subexponential.*

We also recall the following useful lemma Vershynin (2018, Lemma 2.7.7).

**Lemma B.11.** *(Vershynin 2018) Let $X$ and $Y$ be subgaussian random variables. Then $XY$ is subexponential. Moreover*

$$\|XY\|_{\psi_1} \leq \|X\|_{\psi_2} \|Y\|_{\psi_2}$$

We recall one last definition Vershynin (2018, Definition 3.4.1)

**Definition B.12.** *(Vershynin 2018) A random vector $Y \in \mathbb{R}^k$ is called subgaussian if the one dimensional marginals $\langle Y, x \rangle$ are subgaussian random variables for all $x \in \mathbb{R}^k$. The subgaussian norm of $Y$ is defined as*

$$\|Y\|_{\psi_2} = \sup_{x \in S^{k-1}} \|\langle Y, x \rangle\|_{\psi_2}$$

The typical example of a subgaussian random vector is a random vector with independent subgaussian coordinates. The following lemma demonstrates that the $NTK$ at initializeation is a sum of terms that are averages of independent subexponential random variables, which will enable us to use concentration arguments later.

**Theorem B.13.** *Let $w_\ell$, $b_\ell$, $a_\ell$ all be independent subgaussian random variables with subgaussian norms satisfying $\|\bullet\|_{\psi_2} \leq K$. Furthermore assume $\|1\|_{\psi_2} \leq K$. Also let*

$$D = 3\max\{|\sigma(0)|, M\|\sigma'\|_\infty, \|\sigma'\|_\infty, 1\}.$$

*Then for fixed $x, y$, each of the following*

$$\langle \partial_a f(x;\theta), \partial_a f(y;\theta) \rangle, \langle \partial_b f(x;\theta), \partial_b f(y;\theta) \rangle, \langle \partial_W f(x;\theta), \partial_W f(y;\theta) \rangle$$

*is an average of $m$ independent subexponential random variables with subexponential norms bounded by $D^2 K^2$.*

*Proof.* We first observe that

$$\langle \partial_a f(x;\theta), \partial_a f(y;\theta) \rangle_2 = \frac{1}{m} \langle \sigma(Wx+b), \sigma(Wy+b) \rangle_2$$

$$= \frac{1}{m} \sum_{\ell=1}^{m} \sigma(\langle w_\ell, x \rangle_2 + b_\ell)\sigma(\langle w_\ell, y \rangle_2 + b_\ell),$$

$$\langle \partial_b f(x;\theta), \partial_b f(y;\theta) \rangle_2 = \langle \frac{1}{\sqrt{m}} \sigma_1'(x)a, \frac{1}{\sqrt{m}} \sigma_1'(y)a \rangle_2$$

$$= \frac{1}{m} \sum_{\ell=1}^{m} a_\ell^2 \sigma'(\langle w_\ell, x \rangle_2 + b_\ell) \sigma'(\langle w_\ell, y \rangle_2 + b_\ell),$$

$$\langle \partial_W f(x;\theta), \partial_W f(y;\theta) \rangle = \langle \frac{1}{\sqrt{m}} \sigma_1'(x)a \otimes x, \frac{1}{\sqrt{m}} \sigma_1'(y)a \otimes y \rangle_2$$

$$= \frac{1}{m} \langle \sigma_1'(x)a, \sigma_1'(y)a \rangle_2 \langle x, y \rangle_2 = \frac{\langle x, y \rangle_2}{m} \sum_{\ell=1}^{m} a_\ell^2 \sigma'(\langle w_\ell, x \rangle_2 + b_\ell) \sigma'(\langle w_\ell, y \rangle_2 + b_\ell).$$

Note that

$$|\sigma(\langle w_\ell, x \rangle_2 + b_\ell)| \le |\sigma(0)| + \|\sigma'\|_\infty [|\langle w_\ell, x \rangle| + |b_\ell|].$$

Thus

$$\|\sigma(\langle w_\ell, x \rangle_2 + b_\ell)\|_{\psi_2} \le |\sigma(0)| \|1\|_{\psi_2} + \|\sigma'\|_\infty [\||\langle w_\ell, x \rangle|\|_{\psi_2} + \||b_\ell|\|_{\psi_2}]$$
$$\le |\sigma(0)| \|1\|_{\psi_2} + \|\sigma'\|_\infty [M \|w_\ell\|_{\psi_2} + \||b_\ell|\|_{\psi_2}]$$
$$\le 3 \max\{|\sigma(0)|, M \|\sigma'\|_\infty, \|\sigma'\|_\infty\} K \le DK.$$

Also

$$|a_\ell \sigma'(\langle w_\ell, x \rangle_2 + b_\ell)| \le |a_\ell| \|\sigma'\|_\infty \le D|a_\ell|,$$

therefore

$$\|a_\ell \sigma'(\langle w_\ell, x \rangle_2 + b_\ell)\|_{\psi_2} \le D \|a_\ell\|_{\psi_2} \le DK.$$

Finally

$$\left\| |\langle x, y \rangle_2|^{1/2} a_\ell \sigma'(\langle w_\ell, x \rangle_2 + b_\ell) \right\|_{\psi_2} \le M \|\sigma'\|_\infty \|a_\ell\|_{\psi_2} \le DK.$$

It follows by Lemma B.11 that each of $\langle \partial_a f(x;\theta), \partial_a f(y;\theta) \rangle$, $\langle \partial_W f(x;\theta), \partial_W f(y;\theta) \rangle$, and $\langle \partial_b f(x;\theta), \partial_b f(y;\theta) \rangle$ is an average of $m$ independent subexponential random variables with subexponential norm $\|\bullet\|_{\psi_1} \le D^2 K^2$. $\qquad\square$

We now recall the following Theorem from Vershynin (2012, Theorem 5.39) which will be useful.

**Lemma B.14** (Vershynin 2012). *Let $A$ be an $N \times n$ matrix whose rows $A_i$ are independent subgaussian isotropic random vectors in $\mathbb{R}^n$. Then for every $t \ge 0$, with probability at least $1 - 2 \exp(-ct^2)$ one has the following bounds on the singular values*

$$\sqrt{N} - C\sqrt{n} - t \le s_{min}(A) \le s_{max}(A) \le \sqrt{N} + C\sqrt{n} + t.$$

*Here $C = C_K > 0$ depends only on the subgaussian norms $K = \max_i \|A_i\|_{\psi_2}$ of the rows.*

Also the following special case of Vershynin (2012, Lemma 5.5) will be useful for us.

**Lemma B.15** (Vershynin 2012). *Let $Y$ be subgaussian. Then*

$$\mathbb{P}(|Y| > t) \le C \exp(-ct^2 / \|Y\|_{\psi_2}^2).$$

It will be useful to remind the reader that $C, c > 0$ denote absolute constants whose meaning will vary from statement-to-statement, as this abuse of notation becomes especially prevalent during the concentration of measure arguments of the rest of the section. The following lemma provides a concentration inequality for the maximum of subgaussian random variables which will be useful for bounding $\xi$ and $\tilde{\xi}$ later which is necessary for bounding the Lipschitz constant of $K_0$.

**Lemma B.16.** *Let $Y_1, \ldots, Y_n$ be subgaussian random variables with $\|Y_i\|_{\psi_2} \le K$ for $i \in [n]$. Then there exists absolute constants $c, c', C > 0$ such that*

$$\mathbb{P}\left( \max_{i \in [n]} |Y_i| > t + K\sqrt{c' \log n} \right) \le C \exp(-ct^2 / K^2).$$

*Proof.* Since each $Y_i$ is subgaussian we have for any $t \geq 0$ (Lemma B.15)

$$\mathbb{P}\left(|Y_i| > t\right) \leq C \exp\left(-ct^2/\|Y_i\|_{\psi_2}^2\right).$$

By the union bound,

$$\mathbb{P}\left(\max_{i \in [n]} |Y_i| > t + K\sqrt{c^{-1}\log n}\right) \leq \sum_{i=1}^n \mathbb{P}\left(|Y_i| > t + K\sqrt{c^{-1}\log n}\right)$$

$$\leq nC \exp\left(-c\left[t + K\sqrt{c^{-1}\log n}\right]^2/K^2\right) = C \exp(-ct^2/K^2).$$

Thus by setting $c' := c^{-1}$ we get the desired result. $\qquad\square$

We now introduce a high probability bound for $\xi$.

**Lemma B.17.** *Assume that $W_{i,j} \sim \mathcal{W}$, $b_\ell \sim \mathcal{B}$, $a_\ell \sim \mathcal{A}$ are all i.i.d zero-mean, subgaussian random variables with unit variance. Furthermore assume $\|w_\ell\|_{\psi_2}, \|a_\ell\|_{\psi_2}, \|b_\ell\|_{\psi_2} \leq K$ for each $\ell \in [m]$ where $K \geq 1$. Let*

$$\xi = \max\{\frac{1}{\sqrt{m}}\|W\|_{op}, \frac{1}{\sqrt{m}}\|b\|_2, \frac{1}{\sqrt{m}}\|a\|_2\}$$

*Then with probability at least $1 - \delta$*

$$\xi \leq 1 + C\frac{\sqrt{d} + K^2\sqrt{\log(c/\delta)}}{\sqrt{m}}$$

*Proof.* Note that by setting $t = \sqrt{c^{-1}\log(2/\delta)}$ in Lemma B.14 we have that with probability at least $1 - \delta$

$$\frac{1}{\sqrt{m}}\|W\|_{op} \leq 1 + \frac{C\sqrt{d}}{\sqrt{m}} + \frac{\sqrt{c^{-1}\log(2/\delta)}}{\sqrt{m}}$$

Also by Theorem 3.1.1 in (Vershynin, 2018)

$$\left\|\|a\|_2 - \sqrt{m}\right\|_{\psi_2} \leq CK^2$$

$$\left\|\|b\|_2 - \sqrt{m}\right\|_{\psi_2} \leq CK^2$$

Thus by Lemma B.15 and a union bound we have with probability at least $1 - 2\delta$

$$\frac{1}{\sqrt{m}}\|a\|_2, \frac{1}{\sqrt{m}}\|b\|_2 \leq 1 + \frac{C}{\sqrt{m}}K^2\sqrt{\log(c/\delta)}$$

Thus by replacing every $\delta$ in the above arguments with $\delta/3$ and using the union bound we have with probability at least $1 - \delta$

$$\xi \leq 1 + C\frac{\sqrt{d} + K^2\sqrt{\log(c/\delta)}}{\sqrt{m}}.$$

$\qquad\square$

Similarly we now introduce a high probability bound for $\tilde{\xi}$.

**Lemma B.18.** *Assume that $W_{i,j} \sim \mathcal{W}$, $b_\ell \sim \mathcal{B}$, $a_\ell \sim \mathcal{A}$ are all i.i.d zero-mean, subgaussian random variables with unit variance. Furthermore assume $\|w_\ell\|_{\psi_2}, \|a_\ell\|_{\psi_2}, \|b_\ell\|_{\psi_2} \leq K$ for each $\ell \in [m]$ where $K \geq 1$. Let*

$$\tilde{\xi} = \max\{\max_{\ell \in [m]}\|w_\ell\|_2, \|a\|_\infty, \|b\|_\infty\}$$

*Then with probability at least $1 - \delta$ we have*

$$\tilde{\xi} \leq \sqrt{d} + CK^2\left[\sqrt{\log(c/\delta)} + \sqrt{\log m}\right]$$

*Proof.* By Theorem 3.1.1 in (Vershynin, 2018) we have

$$\left\| \|w_\ell\|_2 - \sqrt{d} \right\|_{\psi_2} \leq CK^2$$

Well then by Lemma B.16 there is a constant $c' > 0$ so that

$$\mathbb{P}\left( \max_{\ell \in [m]} \|w_\ell\|_2 - \sqrt{d} > t + CK^2\sqrt{c' \log m} \right)$$

$$\leq \mathbb{P}\left( \max_{\ell \in [m]} \left| \|w_\ell\|_2 - \sqrt{d} \right| > t + CK^2\sqrt{c' \log m} \right) \leq C\exp(-ct^2/K^4).$$

Thus by setting $t = CK^2\sqrt{\log(c/\delta)}$ we have with probability at least $1 - \delta$

$$\max_{\ell \in [m]} \|w_\ell\|_2 \leq \sqrt{d} + CK^2\sqrt{\log(c/\delta)} + CK^2\sqrt{\log m}$$

where we have absorbed the constant $\sqrt{c'}$ into $C$. Similarly by Lemma B.16 and a union bound we get with probability at least $1 - 2\delta$ that

$$\|a\|_\infty, \|b\|_\infty \leq CK\sqrt{\log(c/\delta)} + CK\sqrt{\log m}$$

Thus by replacing each $\delta$ with $\delta/3$ in the above arguments and using the union bound we get with probability at least $1 - \delta$

$$\tilde{\xi} \leq \sqrt{d} + CK^2\left[ \sqrt{\log(c/\delta)} + \sqrt{\log m} \right]$$

$\square$

We are now finally ready to prove the main theorem of this section.

**Theorem B.19.** *Assume that $W_{i,j} \sim \mathcal{W}$, $b_\ell \sim \mathcal{B}$, $a_\ell \sim \mathcal{A}$ are all i.i.d zero-mean, subgaussian random variables with unit variance. Furthermore assume $\|w_\ell\|_{\psi_2}, \|a_\ell\|_{\psi_2}, \|b_\ell\|_{\psi_2} \leq K$ for each $\ell \in [m]$ where $K \geq 1$. Let*

$$D = 3\max\{|\sigma(0)|, M\|\sigma'\|_\infty, \|\sigma'\|_\infty, 1\},$$

$$D' = \max\{\|\sigma'\|_\infty, M\|\sigma''\|_\infty, \|\sigma''\|_\infty\}.$$

*Define*

$$\rho(M, \sigma, d, K, \delta, m) :=$$

$$CDD'\left\{ 1 + C\frac{\sqrt{d} + K^2\sqrt{\log(c/\delta)}}{\sqrt{m}} \right\}^2 \left\{ \sqrt{d} + CK^2\left[ \sqrt{\log(c/\delta)} + \sqrt{\log m} \right] \right\}.$$

*Let $L(K^\infty)$ denote the Lipschitz constant of $K^\infty$. If*

$$m \geq C[\log(c/\delta) + 2d\log(CM\max\{\rho, L(K^\infty)\}\sqrt{m})],$$

*then with probability at least $1 - \delta$*

$$\sup_{z \in B_M \times B_M} |K_0(z) - K^\infty(z)| \leq \frac{1}{\sqrt{m}}\left[ 1 + CD^2K^2\sqrt{\log(c/\delta) + 2d\log(CM\max\{\rho, L(K^\infty)\}\sqrt{m})} \right].$$

*If one instead does the doubling trick then the same conclusion holds.*

*Proof.* First assume we are not doing the doubling trick. Recall that by Theorem B.7 that $K_0$ is Lipschitz with constant at most

$$CDD'\xi(0)^2\tilde{\xi}(0),$$

where $\xi$ and $\tilde{\xi}$ are defined as in the theorem. Well then by Lemmas B.17, B.18 and a union bound we have with probability at least $1 - 2\delta$

$$\xi(0)^2\tilde{\xi}(0) \leq \left\{ 1 + C\frac{\sqrt{d} + K^2\sqrt{\log(c/\delta)}}{\sqrt{m}} \right\}^2 \left\{ \sqrt{d} + CK^2\left[ \sqrt{\log(c/\delta)} + \sqrt{\log m} \right] \right\}.$$

Let $L(K_0), L(K^\infty)$ denote the Lipschitz constant of $K_0$ and $K^\infty$ respectively. Then assuming the above inequality holds we have that

$$L(K_0) \leq \rho(M, \sigma, d, K, \delta, m). \tag{1}$$

For conciseness from now on we will suppress the arguments of $\rho$. Now set

$$\gamma := \frac{1}{2 \max\{\rho, L(K^\infty)\} \sqrt{m}}.$$

Let $\mathcal{N}_\gamma(B_M)$ be the cardinality of a maximal $\gamma$-net of the ball $B_M = \{x : \|x\|_2 \leq M\}$ with respect to the L2 norm $\|\bullet\|_2$. By a standard volume argument we have that

$$\mathcal{N}_\gamma(B_M) \leq \left(\frac{CM}{\gamma}\right)^d.$$

By taking the product of two $\gamma/2$ nets of $B_M$ it follows that we can choose a $\gamma$ net of $B_M \times B_M$, say $\Delta$, with respect to the norm

$$\|(x, y)\| = \|x\|_2 + \|y\|_2$$

such that

$$|\Delta| \leq \left|\mathcal{N}_{\gamma/2}(B_M)\right|^2 \leq \left(\frac{CM}{\gamma}\right)^{2d} =: \mathcal{M}_\gamma.$$

By Theorem B.13 for $(x, y) \in B_M \times B_M$ fixed each of the following

$$\langle \partial_a f(x; \theta), \partial_a f(y; \theta)\rangle_2, \langle \partial_b f(x; \theta), \partial_b f(y; \theta)\rangle_2, \langle \partial_W f(x; \theta), \partial_W f(y; \theta)\rangle$$

is an average of $m$ subexponential random variables with subexponential norm at most $D^2 K^2$. Therefore separately from the randomness discussed before by Bernstein's inequality Vershynin (2018, Theorem 2.8.1) and a union bound we have

$$\mathbb{P}(|K_0(x, y) - K^\infty(x, y)| > t) \leq 3 \times 2 \exp\left(-c \min\left\{\frac{mt^2}{D^4 K^4}, \frac{mt}{D^2 K^2}\right\}\right).$$

Thus for $t \leq D^2 K^2$ we have

$$\mathbb{P}(|K_0(x, y) - K^\infty(x, y)| > t) \leq 6 \exp\left(-c \frac{mt^2}{D^4 K^4}\right).$$

Then by a union bound and the previous inequality we have that for $t \leq D^2 K^2$

$$\mathbb{P}\left(\max_{z' \in \Delta} |K_0(z') - K^\infty(z')| > t\right) \leq 6 \mathcal{M}_\gamma \exp\left(-c \frac{mt^2}{D^4 K^4}\right).$$

Thus by setting $t = CD^2 K^2 \frac{\sqrt{\log(c/\delta) + \log \mathcal{M}_\gamma}}{\sqrt{m}}$ (note that the condition on $m$ in the hypothesis ensures that $t \leq D^2 K^2$) we get that with probability $1 - \delta$

$$\max_{z' \in \Delta} |K_0(z') - K^\infty(z')| \leq t.$$

Now fix $z \in B_M \times B_M$ and choose $z' \in \Delta$ such that $\|z - z'\| \leq \gamma$. Then

$$|K_0(z) - K^\infty(z)| \leq |K_0(z) - K_0(z')|$$
$$+ |K_0(z') - K^\infty(z')| + |K^\infty(z') - K^\infty(z)|$$

$$\leq 2 \max\{L(K_0), L(K^\infty)\} \gamma + t. \tag{2}$$

Note that this argument runs through for any $z \in B_M \times B_M$ therefore

$$\sup_{z \in B_M \times B_M} |K_0(z) - K^\infty(z)| \leq 2 \max\{L(K_0), L(K^\infty)\} \gamma + t.$$

Well by replacing $\delta$ with $\delta/3$ in the previous arguments by taking a union bound we can assume that equations (1) and (2) hold simultaneously. In which case

$$\sup_{z \in B_M \times B_M} |K_0(z) - K^\infty(z)| \le 2\max\{L(K_0), L(K^\infty)\}\gamma + t \le 2\max\{\rho, L(K^\infty)\}\gamma + t$$

$$\le \frac{1}{\sqrt{m}} + CD^2K^2 \frac{\sqrt{\log(c/\delta) + \log \mathcal{M}_\gamma}}{\sqrt{m}} = \frac{1}{\sqrt{m}} + CD^2K^2 \frac{\sqrt{\log(c/\delta) + 2d\log(CM/\gamma)}}{\sqrt{m}}$$

$$= \frac{1}{\sqrt{m}} \left[ 1 + CD^2K^2 \sqrt{\log(c/\delta) + 2d\log(CM\max\{\rho, L(K^\infty)\}\sqrt{m})} \right],$$

where we have used the definition of $\mathcal{M}_\gamma$ in the second-to-last equality and the definition of $\gamma$ in the last equality. Since the doubling trick does not change the distribution of $K_0$, the same conclusion holds under that initialization scheme. $\square$

We immediately get the following corollary.

**Corollary B.20.** *Assume that $W_{i,j} \sim \mathcal{W}$, $b_\ell \sim \mathcal{B}$, $a_\ell \sim \mathcal{A}$ are all i.i.d zero-mean, subgaussian random variables with unit variance. Furthermore assume $\|w_\ell\|_{\psi_2}, \|a_\ell\|_{\psi_2}, \|b_\ell\|_{\psi_2} \le K$ for each $\ell \in [m]$ where $K \ge 1$. Then*

$$m \ge C[\log(c/\delta) + \tilde{O}(d)]$$

*suffices to ensure that with probability at least $1 - \delta$*

$$\sup_{z \in B_M \times B_M} |K_0(z) - K^\infty(z)| = \tilde{O}\left(\frac{\sqrt{d}}{\sqrt{m}}\right).$$

*If one instead does the doubling trick then the same conclusion holds.*

## B.5 CONTROL OF NETWORK AT INITIALIZATION

Many of our previous results depend on the quantity $\|\hat{r}(0)\|_{\mathbb{R}^n}$ which depends on the network at initialization. Before we proceed we must control the infinity norm of the network at initialization and work out a few consequences of this. The following lemma controls $\|f(\bullet; \theta_0)\|_\infty$.

**Lemma B.21.** *Assume that $W_{i,j} \sim \mathcal{W}$, $b_\ell \sim \mathcal{B}$, $a_\ell \sim \mathcal{A}$, $b_0 \sim \mathcal{B}'$ are all i.i.d zero-mean, subgaussian random variables with unit variance. Furthermore assume $\|1\|_{\psi_2}, \|w_\ell\|_{\psi_2}, \|a_\ell\|_{\psi_2}, \|b_\ell\|_{\psi_2} \le K$ for each $\ell \in [m]$ where $K \ge 1$. Let*

$$D = 3\max\{|\sigma(0)|, M\|\sigma'\|_\infty, \|\sigma'\|_\infty, 1\}$$

$$L(m, \sigma, d, K, \delta) := \sqrt{m}\|\sigma'\|_\infty \left\{ 1 + C\frac{\sqrt{d} + K^2\sqrt{\log(c/\delta)}}{\sqrt{m}} \right\}^2.$$

*Assume that*

$$m \ge C[\log(c/\delta) + d\log(CML)].$$

*Then with probability at least $1 - \delta$*

$$\sup_{x \in B_M} |f(x; \theta_0)| \le CDK^2 \sqrt{d\log(CML) + \log(c/\delta)} = \tilde{O}(\sqrt{d}).$$

*Proof.* First we note that

$$\left| \frac{a^T}{\sqrt{m}}\sigma(Wx + b) - \frac{a^T}{\sqrt{m}}\sigma(Wy + b) \right| \le \frac{\|a\|_2}{\sqrt{m}} \|\sigma(Wx + b) - \sigma(Wy + b)\|_2$$

$$\le \frac{\|a\|_2}{\sqrt{m}} \|\sigma'\|_\infty \|W(x - y)\|_2 \le \frac{\|a\|_2}{\sqrt{m}} \|\sigma'\|_\infty \|W\|_{op} \|x - y\|_2 \le \sqrt{m}\|\sigma'\|_\infty \xi(0)^2 \|x - y\|_2,$$

where $\xi(0)$ is defined as in Lemma B.17. Thus $f(\bullet; \theta_0)$ is Lipschitz with constant $L = \sqrt{m}\|\sigma'\|_\infty \xi(0)^2$. Well then by Lemma B.17 we have with probability at least $1 - \delta$

$$\xi(0)^2 \le \left\{ 1 + C\frac{\sqrt{d} + K^2\sqrt{\log(c/\delta)}}{\sqrt{m}} \right\}^2. \tag{3}$$

When the above holds we have that $f(\bullet; \theta_0)$ is Lipschitz with constant

$$L := \sqrt{m} \left\| \sigma' \right\|_\infty \left\{ 1 + C \frac{\sqrt{d} + K^2 \sqrt{\log(c/\delta)}}{\sqrt{m}} \right\}^2 .$$

On the other hand note that

$$\left| \sigma(\langle w_\ell, x \rangle_2 + b_\ell) \right| \leq \left| \sigma(0) \right| + \left\| \sigma' \right\|_\infty \left[ \left| \langle w_\ell, x \rangle_2 \right| + \left| b_\ell \right| \right].$$

Thus

$$\left\| \sigma(\langle w_\ell, x \rangle_2 + b_\ell) \right\|_{\psi_2} \leq \left| \sigma(0) \right| \left\| 1 \right\|_{\psi_2} + \left\| \sigma' \right\|_\infty \left[ \left\| \left| \langle w_\ell, x \rangle_2 \right| \right\|_{\psi_2} + \left\| \left| b_\ell \right| \right\|_{\psi_2} \right]$$
$$\leq \left| \sigma(0) \right| \left\| 1 \right\|_{\psi_2} + \left\| \sigma' \right\|_\infty \left[ M \left\| w_\ell \right\|_{\psi_2} + \left\| \left| b_\ell \right| \right\|_{\psi_2} \right]$$
$$\leq 3 \max \{ \left| \sigma(0) \right|, M \left\| \sigma' \right\|_\infty, \left\| \sigma' \right\|_\infty \} K \leq DK.$$

Therefore by Lemma B.11 we have

$$\left\| a_\ell \sigma(\langle w_\ell, x \rangle_2 + b_\ell) \right\|_{\psi_1} \leq DK^2.$$

Thus for each $x$ fixed we have by Bernstein's inequality Vershynin (2018, Theorem 2.8.1)

$$\mathbb{P} \left( \left| \sum_{\ell=1}^m a_\ell \sigma(\langle w_\ell, x \rangle_2 + b_\ell) \right| > t\sqrt{m} \right) \leq 2 \exp \left( -c \min \left[ \frac{t^2}{[DK^2]^2}, \frac{t\sqrt{m}}{DK^2} \right] \right).$$

Thus for $t \leq \sqrt{m} DK^2$ this simplifies to

$$\mathbb{P} \left( \left| \sum_{\ell=1}^m a_\ell \sigma(\langle w_\ell, x \rangle_2 + b_\ell) \right| > t\sqrt{m} \right) \leq 2 \exp \left( -c \frac{t^2}{D^2 K^4} \right).$$

Let $\Delta$ be a $\gamma$ net of the ball $B_M = \{ x : \left\| x \right\|_2 \leq M \}$ with respect to the Euclidean $\left\| \bullet \right\|_2$ norm. Then by a standard volume argument we have that

$$\left| \Delta \right| \leq \left( \frac{CM}{\gamma} \right)^d =: \mathcal{M}_\gamma.$$

Thus by a union bound we have for $t \leq \sqrt{m} DK^2$

$$\mathbb{P} \left( \max_{x \in \Delta} \left| \sum_{\ell=1}^m a_\ell \sigma(\langle w_\ell, x \rangle_2 + b_\ell) \right| > t\sqrt{m} \right) \leq 2 \mathcal{M}_\gamma \exp \left( -c \frac{t^2}{D^2 K^4} \right).$$

Thus by setting $t = CDK^2 \sqrt{\log(c\mathcal{M}_\gamma/\delta)}$ assuming $t \leq \sqrt{m} DK^2$ we have with probability at least $1 - \delta$

$$\max_{x \in \Delta} \left| \sum_{\ell=1}^m \frac{a_\ell}{\sqrt{m}} \sigma(\langle w_\ell, x \rangle_2 + b_\ell) \right| \leq t. \tag{4}$$

On the other hand by Lemma B.15 our prior definition of $t$ is large enough (up to a redefinition of the constants $c, C$) to ensure that with probability at least $1 - \delta$

$$\left| b_0 \right| \leq t. \tag{5}$$

When (4) and (5) hold simultaneously we have that $\max_{x' \in \Delta} \left| f(x', \theta_0) \right| \leq 2t$. By a union bound we have with probability at least $1 - 3\delta$ that (3), (4), (5) hold simultaneously. Well then for any $x \in B_M$ we may choose $x' \in \Delta$ so that $\left\| x - x' \right\|_2 \leq \gamma$. Then

$$\left| f(x; \theta_0) \right| \leq \left| f(x'; \theta_0) \right| + \left| f(x; \theta_0) - f(x'; \theta_0) \right| \leq 2t + L\gamma.$$

Therefore

$$\sup_{x \in B_M} \left| f(x; \theta_0) \right| \leq 2t + L\gamma$$

and this argument runs through for any $\gamma > 0$. We will set $\gamma = 1/L$. Note that for this choice of $\gamma$ the hypothesis on $m$ ensures that $t \leq \sqrt{m} DK^2$. Thus the preceding argument goes through in this case. Thus by replacing $\delta$ with $\delta/3$ in the previous argument we get the desired conclusion up to a redefinition of $c, C$. $\qquad \square$

We quickly introduce the following lemma.

**Lemma B.22.**
$$\|\hat{r}(0)\|_{\mathbb{R}^n} \leq \|f(\bullet; \theta_0)\|_\infty + \|y\|_{\mathbb{R}^n} .$$

*Proof.* Let $\hat{y} \in \mathbb{R}^n$ be the vector whose $i$th entry is equal to $f(x_i; \theta_0)$. Well then note that $\|\hat{y}\|_{\mathbb{R}^n} \leq \|f(\bullet; \theta_0)\|_\infty$. Therefore

$$\|\hat{r}(0)\|_{\mathbb{R}^n} = \|\hat{y} - y\|_{\mathbb{R}^n} \leq \|\hat{y}\|_{\mathbb{R}^n} + \|y\|_{\mathbb{R}^n} \leq \|f(\bullet; \theta_0)\|_\infty + \|y\|_{\mathbb{R}^n} .$$

$\square$

Finally we prove one last lemma that will be useful later.

**Lemma B.23.** *Assume that $W_{i,j} \sim \mathcal{W}$, $b_\ell \sim \mathcal{B}$, $a_\ell \sim \mathcal{A}$ are all i.i.d zero-mean, subgaussian random variables with unit variance. Furthermore assume $\|1\|_{\psi_2}, \|w_\ell\|_{\psi_2}, \|a_\ell\|_{\psi_2}, \|b_\ell\|_{\psi_2} \leq K$ for each $\ell \in [m]$ where $K \geq 1$. Let $\Gamma > 1$, $D = 3 \max\{|\sigma(0)|, M \|\sigma'\|_\infty, \|\sigma'\|_\infty, 1\}$,*

$$L(m, \sigma, d, K, \delta) := \sqrt{m} \|\sigma'\|_\infty \left\{ 1 + C \frac{\sqrt{d} + K^2 \sqrt{\log(c/\delta)}}{\sqrt{m}} \right\}^2 ,$$

$$\rho := CDK^2 \sqrt{d \log(CML) + \log(c/\delta)} = \tilde{O}(\sqrt{d}).$$

*Suppose*

$$m \geq \frac{4D^2 \|y\|_{\mathbb{R}^n}^2 T^2}{[\log(\Gamma)]^2} \text{ and } m \geq \max\left\{ \frac{4D^2 \rho^2 T^2}{[\log(\Gamma)]^2}, \left(\frac{\rho}{DK^2}\right)^2 \right\} .$$

*Then with probability at least $1 - \delta$*

$$\max_{t \leq T} \xi(t) \leq \Gamma \xi(0) \quad \max_{t \leq T} \tilde{\xi}(t) \leq \Gamma \tilde{\xi}(0),$$

*where $\xi(t)$ and $\tilde{\xi}(t)$ are defined as in Lemmas B.2 and B.3. If one instead does the doubling trick the second hypothesis on $m$ can be removed and the conclusion holds with probability $1$.*

*Proof.* First assume we are not doing the doubling trick. Well from the condition $m \geq \left(\frac{\rho}{DK^2}\right)^2$ we have by Lemma B.21 that with probability at least $1 - \delta$

$$\sup_{x \in B_M} |f(x; \theta_0)| \leq CDK^2 \sqrt{d \log(CML) + \log(c/\delta)} =: \rho.$$

Also by Lemma B.22 and the above bound we have

$$\|\hat{r}(0)\|_{\mathbb{R}^n} \leq \|y\|_{\mathbb{R}^n} + \rho.$$

Well in this case

$$\frac{D \|\hat{r}(0)\|_{\mathbb{R}^n} t}{\sqrt{m}} \leq \frac{D[\|y\|_{\mathbb{R}^n} + \rho]t}{\sqrt{m}} \leq \frac{2D \max\{\|y\|_{\mathbb{R}^n}, \rho\}t}{\sqrt{m}} \leq \log(\Gamma),$$

where we have used the hypothesis on $m$ in the last inequality. Therefore by Lemmas B.2, B.3 we have in this case that

$$\max_{t \leq T} \xi(t) \leq \Gamma \xi(0) \quad \max_{t \leq T} \tilde{\xi}(t) \leq \Gamma \tilde{\xi}(0)$$

Now assume we are performing the doubling trick so that $f(\bullet; \theta_0) \equiv 0$. Then $\rho$ in the previous argument can simply be replaced with zero and the same argument runs through, except using Lemma B.21 is no longer necessary (and thus the second hypothesis on $m$ is not needed). Without using Lemma B.21 the whole argument is deterministic so that the conclusion holds with probability $1$. $\square$

### B.6 $NTK$ TIME DEVIATIONS BOUNDS

In this section we bound the deviations of the $NTK$ throughout time. This section runs through the following steps

- Bound $\sup_{(x,y)\in B_M\times B_M}|\partial_t K_t(x,y)|$
- Bound $\sup_{(x,y)\in B_M\times B_M}|K_t(x,y)-K_0(x,y)|$

In the following lemma we will provide an upper bound on the $NTK$ derivative $\sup_{x,y\in B_M\times B_M}|\partial_t K_t(x,y)|$.

**Lemma B.24.** *Let*

$$\xi(t)=\max\{\frac{1}{\sqrt{m}}\left\|W(t)\right\|_{op},\frac{1}{\sqrt{m}}\left\|b(t)\right\|_2,\frac{1}{\sqrt{m}}\left\|a(t)\right\|_2,1\},$$

$$\tilde{\xi}(t)=\max\{\max_{\ell\in[m]}\left\|w_\ell(t)\right\|_2,\left\|a(t)\right\|_\infty,\left\|b(t)\right\|_\infty,1\}$$

$$D=3\max\{|\sigma(0)|,M\left\|\sigma'\right\|_\infty,\left\|\sigma'\right\|_\infty,1\}$$

$$D':=\left[\max\{\left\|\sigma'\right\|_\infty,\left\|\sigma''\right\|_\infty\}^2[M^2+1]+D\left\|\sigma'\right\|_\infty\right]\max\{1,M\}.$$

*Then for any initial conditions $W(0)$, $b(0)$, $a(0)$ we have for all $t$*

$$\sup_{x,y\in B_M\times B_M}|\partial_t K_t(x,y)|\leq\frac{CDD'}{\sqrt{m}}\xi(t)^2\tilde{\xi}(t)\left\|\hat{r}(t)\right\|_{\mathbb{R}^n}.$$

*Proof.* We need to bound the following time derivatives

$$\partial_t\partial_a f(x;\theta)=\partial_t x^{(1)}=-\frac{1}{n}\sum_{i=1}^n\hat{r}_i\left[\frac{1}{\sqrt{m}}\sigma_1'(x)\sigma_1'(x_i)\frac{a}{\sqrt{m}}\right][\langle x,x_i\rangle_2+1]$$

$$\partial_t\partial_W f(x;\theta)=\partial_t\frac{1}{\sqrt{m}}\sigma_1'(x)a\otimes x$$

$$=\frac{1}{\sqrt{m}}[(\partial_t\sigma_1'(x))a+\sigma_1'(x)(\partial_t a)]\otimes x$$

$$\partial_t\partial_b f(x;\theta)=\partial_t\frac{1}{\sqrt{m}}\sigma_1'(x)a=\frac{1}{\sqrt{m}}\left([\partial_t\sigma_1'(x)]a+\sigma_1'(x)\partial_t a\right).$$

Note that

$$\left\|\frac{1}{\sqrt{m}}\sigma_1'(x)\sigma_1'(x_i)\frac{a}{\sqrt{m}}[\langle x,x_i\rangle_2+1]\right\|_2\leq\frac{1}{\sqrt{m}}\left\|\sigma'\right\|_\infty^2\frac{1}{\sqrt{m}}\left\|a\right\|_2[M^2+1].$$

Thus by Lemma B.1

$$\left\|\partial_t\partial_a f(x;\theta)\right\|_2\leq\frac{1}{\sqrt{m}}\left\|\sigma'\right\|_\infty^2\frac{1}{\sqrt{m}}\left\|a\right\|_2[M^2+1]\left\|\hat{r}(t)\right\|_{\mathbb{R}^n}$$

$$\leq\frac{\left\|\sigma'\right\|_\infty^2[M^2+1]]}{\sqrt{m}}\xi(t)\left\|\hat{r}(t)\right\|_{\mathbb{R}^n}.$$

On the other hand

$$[\partial_t\sigma_1'(x)]a=-\frac{1}{n}\sum_{i=1}^n\hat{r}_i\frac{1}{\sqrt{m}}\sigma_1''(x)\sigma_1'(x_i)diag(a)a[\langle x,x_i\rangle_2+1].$$

Well

$$\frac{1}{\sqrt{m}}\left\|\sigma_1''(x)\sigma_1'(x_i)diag(a)a[\langle x,x_i\rangle_2+1\right\| \leq \frac{1}{\sqrt{m}}\left\|\sigma''\right\|_\infty\left\|\sigma'\right\|_\infty\left\|a\right\|_\infty\left\|a\right\|_2[M^2+1]$$

$$\leq\left\|\sigma''\right\|_\infty\left\|\sigma'\right\|_\infty[M^2+1]\xi(t)\tilde{\xi}(t).$$

Thus by Lemma B.1 we have that

$$\left\|[\partial_t\sigma_1'(x)]a\right\| \leq \left\|\sigma''\right\|_\infty\left\|\sigma'\right\|_\infty[M^2+1]\xi(t)\tilde{\xi}(t)\left\|\hat{r}(t)\right\|_{\mathbb{R}^n}.$$

Finally we have

$$\sigma_1'(x)\partial_t a = -\frac{1}{n}\sum_{i=1}^n \hat{r}_i\sigma_1'(x)x_i^{(1)}.$$

Well

$$\left\|\sigma_1'(x)x_i^{(1)}\right\| \leq \left\|\sigma'\right\|_\infty\left\|x_i^{(1)}\right\|_2 = \left\|\sigma'\right\|_\infty\frac{1}{\sqrt{m}}\left\|\sigma(Wx_i+b)\right\|_2$$

$$\leq \left\|\sigma'\right\|_\infty\left[|\sigma(0)|+\frac{1}{\sqrt{m}}\left\|\sigma'\right\|_\infty(\left\|W\right\|_{op}M+\left\|b\right\|_2)\right]$$

$$\leq \left\|\sigma'\right\|_\infty\left[|\sigma(0)|+M\left\|\sigma'\right\|_\infty+\left\|\sigma'\right\|_\infty\right]\xi(t) \leq \left\|\sigma'\right\|_\infty D\xi(t).$$

Thus we finally by Lemma B.1 again we get that

$$\left\|\sigma_1'(x)\partial_t a\right\| \leq \left\|\sigma'\right\|_\infty D\xi(t)\left\|\hat{r}(t)\right\|_{\mathbb{R}^n}.$$

It follows that

$$\left\|\partial_t\partial_b f(x;\theta)\right\|$$

$$\leq \frac{1}{\sqrt{m}}\left[\left\|\sigma''\right\|_\infty\left\|\sigma'\right\|_\infty[M^2+1]+\left\|\sigma'\right\|_\infty D\right]\xi(t)\tilde{\xi}(t)\left\|\hat{r}(t)\right\|_{\mathbb{R}^n}$$

and similarly

$$\left\|\partial_t\partial_W f(x;\theta)\right\|$$

$$\leq \frac{M}{\sqrt{m}}\left[\left\|\sigma''\right\|_\infty\left\|\sigma'\right\|_\infty[M^2+1]+\left\|\sigma'\right\|_\infty D\right]\xi(t)\tilde{\xi}(t)\left\|\hat{r}(t)\right\|_{\mathbb{R}^n}.$$

Thus in total we can say

$$\left\|\partial_t\partial_a f(x;\theta)\right\|_2, \left\|\partial_t\partial_b f(x;\theta)\right\|_2, \left\|\partial_t\partial_w f(x;\theta)\right\|_F \leq \frac{D'}{\sqrt{m}}\xi(t)\tilde{\xi}(t)\left\|\hat{r}(t)\right\|_{\mathbb{R}^n}.$$

It thus follows by the chain rule and Lemma B.5 that

$$\sup_{(x,y)\in B_M\times B_M}\left|\partial_t K_t(x,y)\right| \leq \frac{CDD'}{\sqrt{m}}\xi(t)^2\tilde{\xi}(t)\left\|\hat{r}(t)\right\|_{\mathbb{R}^n}.$$

$\square$

Using the previous lemma we can now bound the deviations of the $NTK$.

**Theorem B.25.** *Assume that $W_{i,j}\sim\mathcal{W}$, $b_\ell\sim\mathcal{B}$, $a_\ell\sim\mathcal{A}$ are all i.i.d zero-mean, subgaussian random variables with unit variance. Furthermore assume $\left\|w_\ell\right\|_{\psi_2}, \left\|a_\ell\right\|_{\psi_2}, \left\|b_\ell\right\|_{\psi_2}\leq K$ for each $\ell\in[m]$ where $K\geq 1$. Let $\Gamma>1$ and $T>0$ be positive constants,*

$$D := 3\max\{|\sigma(0)|, M\left\|\sigma'\right\|_\infty, \left\|\sigma'\right\|_\infty, 1\},$$

$$D' := \left[\max\{\left\|\sigma'\right\|_\infty, \left\|\sigma''\right\|_\infty\}^2[M^2+1]+D\left\|\sigma'\right\|_\infty\right]\max\{1,M\},$$

*and assume*

$$m \geq \frac{4D^2\left\|y\right\|_{\mathbb{R}^n}^2 T^2}{[\log(\Gamma)]^2} \text{ and } m \geq \max\left\{\frac{4D^2 O(\log(c/\delta)+\tilde{O}(d))T^2}{[\log(\Gamma)]^2}, O(\log(c/\delta)+\tilde{O}(d))\right\}.$$

*Then with probability at least $1 - \delta$*

$$\sup_{(x,y) \in B_M \times B_M} |K_t(x,y) - K_0(x,y)| \leq$$

$$t\Gamma^3 \frac{CDD'}{\sqrt{m}} \|\hat{r}(0)\|_{\mathbb{R}^n} \left\{ 1 + C\frac{\sqrt{d} + K^2 \sqrt{\log(c/\delta)}}{\sqrt{m}} \right\}^2 \left\{ \sqrt{d} + CK^2 \left[ \sqrt{\log(c/\delta)} + \sqrt{\log m} \right] \right\}.$$

*If one instead does the doubling trick then the second condition on $m$ can be removed from the hypothesis and the same conclusion holds.*

*Proof.* First assume we are not doing the doubling trick. By Lemmas B.17, B.18 and a union bound we have with probability at least $1 - \delta$

$$\xi(0)^2 \tilde{\xi}(0) \leq \left\{ 1 + C\frac{\sqrt{d} + K^2 \sqrt{\log(c/\delta)}}{\sqrt{m}} \right\}^2 \left\{ \sqrt{d} + CK^2 \left[ \sqrt{\log(c/\delta)} + \sqrt{\log m} \right] \right\}. \quad (6)$$

Note that $\rho$ as defined in Lemma B.23 satisfies $\rho^2 = O(\log(c/\delta) + \tilde{O}(d))$. Thus the hypothesis on $m$ is strong enough to apply Lemma B.23, therefore by applying this lemma we have with probability at least $1 - \delta$

$$\max_{t \leq T} \xi(t) \leq \Gamma \xi(0) \quad \max_{t \leq T} \tilde{\xi}(t) \leq \Gamma \tilde{\xi}(0). \quad (7)$$

Thus by replacing $\delta$ with $\delta/2$ and taking a union bound we have that with probability at least $1 - \delta$ (6) and (7) hold simultaneously. Then using Lemma B.24 and the fact that $\|\hat{r}(t)\|_{\mathbb{R}^n} \leq \|\hat{r}(0)\|_{\mathbb{R}^n}$ we have for $t \leq T$

$$\sup_{(x,y) \in B_M \times B_M} |\partial_t K_t(x,y)| \leq \Gamma^3 \frac{CDD'}{\sqrt{m}} \xi(0)^2 \tilde{\xi}(0) \|\hat{r}(0)\|_{\mathbb{R}^n}.$$

Therefore by the fundamental theorem of calculus for $t \leq T$

$$\sup_{(x,y) \in B_M \times B_M} |K_t(x,y) - K_0(x,y)| \leq t\Gamma^3 \frac{CDD'}{\sqrt{m}} \xi(0)^2 \tilde{\xi}(0) \|\hat{r}(0)\|_{\mathbb{R}^n}$$

$$\leq t\Gamma^3 \frac{CDD'}{\sqrt{m}} \|\hat{r}(0)\|_{\mathbb{R}^n} \left\{ 1 + C\frac{\sqrt{d} + K^2 \sqrt{\log(c/\delta)}}{\sqrt{m}} \right\}^2 \left\{ \sqrt{d} + CK^2 \left[ \sqrt{\log(c/\delta)} + \sqrt{\log m} \right] \right\}.$$

Now consider if one instead does the doubling trick where one does the following swaps $W(0) \to \begin{bmatrix} W(0) \\ W(0) \end{bmatrix}$, $b(0) \to \begin{bmatrix} b(0) \\ b(0) \end{bmatrix}$, $a(0) \to \begin{bmatrix} a(0) \\ -a(0) \end{bmatrix}$ and $m \to 2m$ where $W(0)$, $b(0)$, and $a(0)$ are initialized as before. Then $\xi(0)$ and $\tilde{\xi}(0)$ do not change. We can then run through the same exact proof as before except when we apply Lemma B.23 the second hypothesis on $m$ is no longer needed. $\quad \square$

**Theorem B.26.** *Assume that $W_{i,j} \sim \mathcal{W}$, $b_\ell \sim \mathcal{B}$, $a_\ell \sim \mathcal{A}$ are all i.i.d zero-mean, subgaussian random variables with unit variance. Let $\Gamma > 1$ and $T > 0$ be positive constants and let $D := 3\max\{|\sigma(0)|, M\|\sigma'\|_\infty, \|\sigma'\|_\infty, 1\}$. Assume*

$$m \geq \frac{4D^2 \|y\|_{\mathbb{R}^n}^2 T^2}{[\log(\Gamma)]^2} \text{ and } m \geq \max\left\{ \frac{4D^2 O(\log(c/\delta) + \tilde{O}(d))T^2}{[\log(\Gamma)]^2}, O(\log(c/\delta) + \tilde{O}(d)) \right\}.$$

*Then with probability at least $1 - \delta$ we have for $t \leq T$*

$$\sup_{(x,y) \in B_M \times B_M} |K_t(x,y) - K^\infty(x,y)| = \tilde{O}\left( \frac{\sqrt{d}}{\sqrt{m}} \left[ 1 + t\Gamma^3 \|\hat{r}(0)\|_{\mathbb{R}^n} \right] \right).$$

*If one instead does the doubling trick then one can remove the assumption $m \geq \frac{4D^2 O(\log(c/\delta) + \tilde{O}(d))T^2}{[\log(\Gamma)]^2}$ and have the same conclusion hold.*

*Proof.* The condition $m \geq O(\log(c/\delta) + \tilde{O}(d))$ is sufficient to satisfy the hypothesis of Theorem B.19. The condition on $m$ also immediately satisfies the hypothesis of Theorem B.25. The desired result then follows from a union bound. $\quad \square$

**Theorem B.27.** *Under the same assumptions as Theorem B.26 we have that with probability at least $1 - \delta$ for all $t \leq T$*

$$\|H^\infty - H_t\|_{op} \leq n\tilde{O}\left(\frac{\sqrt{d}}{\sqrt{m}}\left[1 + t\Gamma^3\|\hat{r}(0)\|_{\mathbb{R}^n}\right]\right)$$

$$\sup_{s \leq T}\|G^\infty - G_t\|_{op} \leq \tilde{O}\left(\frac{\sqrt{d}}{\sqrt{m}}\left[1 + t\Gamma^3\|\hat{r}(0)\|_{\mathbb{R}^n}\right]\right).$$

*Proof.* Recall that for a matrix $A \in \mathbb{R}^{m \times n}$ $\|A\|_{op} \leq \sqrt{mn}\max_{i,j}|A_{i,j}|$. Thus by Theorem B.26 with probability at least $1 - \delta$

$$\|H^\infty - H_t\|_{op} \leq n\max_{i,j}|H^\infty_{i,j} - (H_t)_{i,j}| \leq n\sup_{(x,y)\in B_M \times B_M}|K_t(x,y) - K^\infty(x,y)|$$

$$= n\tilde{O}\left(\frac{\sqrt{d}}{\sqrt{m}}\left[1 + t\Gamma^3\|\hat{r}(0)\|_{\mathbb{R}^n}\right]\right).$$

The second bound follows from $G_s = \frac{1}{n}H_s$ and $G^\infty = \frac{1}{n}H^\infty$. $\qquad\qquad\square$

### B.7 NTK DEVIATIONS FOR RELU APPROXIMATIONS

The NTK deviation bounds given in the previous subsections assumed $\|\sigma''\|_\infty < \infty$. For ReLU this assumption is not satisfied. It is natural to ask to what extent we might expect the results to hold when the activation function is $\sigma(x) = ReLU(x) = \max\{0, x\}$. The closest we can get to ReLU without modifying the proofs is to use the Softmax approximation to ReLU, namely $\sigma(x) = \frac{1}{\alpha}\ln(1 + \exp(\alpha x))$, and consider what happens as $\alpha \to \infty$. For this choice of $\sigma$ we have that $\|\sigma''\|_\infty = O(\alpha)$. In Subsection B.6 where you will pay the biggest penalty is in Theorem B.25 via the constant $D' = O(\|\sigma''\|_\infty^2) = O(\alpha^2)$. Since the final bound depends on the ratio $\frac{D'}{\sqrt{m}}$ you will have that $m$ will grow like $O(\alpha^4)$. This is no moderate penalty, although we might expect the results to hold for wide ReLU networks if a finite $\alpha$ provides a reasonable approximation. In particular Softmax $\ln(1 + \exp(x))$ leads to a fixed constant for $D'$.

## C  UNDERPARAMETERIZED REGIME

In this section we build the tools to study the implicit bias in the underparameterized case. Our ultimate goal is prove Theorem 3.5.

Outline of this section

- Review operator theory
- Prove damped deviations equation
- Bound $\|(T_{K^\infty} - T_n^s)r_t\|_2$
    - Bound $\|(T_n - T_n^s)r_t\|_2$ using $NTK$ deviation results (comparatively easy)
    - Bound $\|(T_{K^\infty} - T_n)r_t\|_2$
        * Derive covering number for a class of functions $\mathcal{C}$
        * Use covering number to bound $\sup_{g \in \mathcal{C}}\|(T_{K^\infty} - T_n)g\|$
        * Show that $r_t$ is in class $\mathcal{C}$
- Prove Theorem 3.5

### C.1 RKHS AND MERCER'S THEOREM

We recall some facts about Reproducing Kernel Hilbert Spaces (RKHS) and Mercer's Theorem. For additional background we suggest Berlinet & Thomas-Agnan (2004). Let $X \subset \mathbb{R}^d$ be a compact space equipped with a strictly positive (regular Borel) probability measure $\rho$. Let $K : X \times X \to \mathbb{R}$ be

a continuous, symmetric, positive definite function. We define the integral operator $T_K : L^2_\rho(X) \to L^2_\rho(X)$

$$T_K f(x) := \int_X K(x, s) f(s) d\rho(s).$$

In this setting $T_K$ is a compact, positive, self-adjoint operator. By the spectral theorem there is a countable nonincreasing sequence of nonnegative values $\{\sigma_i\}_{i=1}^\infty$ and an orthonormal set $\{\phi_i\}_{i=1}^\infty$ in $L^2$ such that $T_K \phi_i = \sigma_i \phi_i$. We will assume that $T_K$ is strictly positive, i.e. $\langle f, T_K f \rangle_2 > 0$ for $f \neq 0$, so that we have further that $\{\phi_i\}_{i=1}^\infty$ is an orthonormal basis of $L^2$ and $\sigma_i > 0$ for all $i$. Moreover since $K$ is continuous we may select the $\phi_i$ so that they are continuous functions, i.e. $\phi_i \in C(X)$ for each $i$. Then by Mercer's theorem we can decompose

$$K(x, y) = \sum_{i=1}^\infty \sigma_i \phi_i(x) \phi_i(y),$$

where the convergence is uniform. Furthermore the RKHS $\mathcal{H}$ associated with $K$ is given by the set of functions

$$\mathcal{H} = \left\{ f \in L^2 : \sum_{i=1}^\infty \frac{|\langle f, \phi_i \rangle_2|^2}{\sigma_i} < \infty \right\},$$

where the inner product on $\mathcal{H}$ is given by

$$\langle f, g \rangle_\mathcal{H} = \sum_{i=1}^\infty \frac{\langle f, \phi_i \rangle_2 \langle g, \phi_i \rangle_2}{\sigma_i}.$$

Note that in this setting $\{\sqrt{\sigma_i} \phi_i\}_{i=1}^\infty$ is an orthonormal basis of $\mathcal{H}$. Define $K_x := K(\bullet, x)$. Recall the RKHS has the defining properties

$$K_x \in \mathcal{H} \quad \forall x \in X$$

$$h(x) = \langle h, K_x \rangle_\mathcal{H} \quad \forall (x, h) \in X \times \mathcal{H}.$$

We will let $\kappa := \sup_{x \in X} K(x, x) < \infty$. From this we will have the useful inequality: for $h \in \mathcal{H}$

$$|h(x)| = |\langle h, K_x \rangle_\mathcal{H}| \leq \|h\|_\mathcal{H} \|K_x\|_\mathcal{H} = \|h\|_\mathcal{H} \sqrt{\langle K_x, K_x \rangle_\mathcal{H}} = \|h\|_\mathcal{H} \sqrt{K(x, x)}$$
$$\leq \kappa^{1/2} \|h\|_\mathcal{H}.$$

Furthermore the elements of $\mathcal{H}$ are bounded continuous functions and $\mathcal{H}$ is seperable.

## C.2 Hilbert-Schmidt and Trace Class Operators

We will recall some definitions from (Rosasco et al., 2010). A bounded operator on a separable Hilbert space with associated norm $\|\bullet\|$ is called *Hilbert-Schmidt* if

$$\sum_{i=1}^\infty \|A e_i\|^2 < \infty$$

for some (any) orthonormal basis $\{e_i\}_i$. For such an operator we define its Hilbert-Schmidt norm $\|A\|_{HS}$ to be the square root of the above sum. The Hilbert-Schmidt norm is the analog of the Frobenius norm for matrices. It is useful to note that every Hilbert-Schmidt operator is compact. The space of Hilbert-Schmidt operators is a Hilbert space with respect to the inner product

$$\langle A, B \rangle = \sum_j \langle A e_j, B e_j \rangle.$$

A stronger notion is that of a *trace class* operator. We say a bounded operator on a separable Hilbert space is *trace class* if

$$\sum_{i=1}^\infty \langle \sqrt{A^* A} e_i, e_i \rangle < \infty$$

for some (any) orthonormal bases $\{e_i\}_i$. For such an operator we may define

$$Tr(A) := \sum_{i=1}^{\infty} \langle Ae_i, e_i \rangle.$$

By Lidskii's theorem the above sum is also equal to the sum of the eigenvalues of $A$ repeated by multiplicity. The space of trace class operators is a Banach space with the norm $\|A\|_{TC} = Tr(\sqrt{A^*A})$. The following inequalities will be useful

$$\|A\| \le \|A\|_{HS} \le \|A\|_{TC}.$$

Furthermore if $A$ is Hilbert-Schmidt and $B$ is bounded we have

$$\|BA\|_{HS}, \|AB\|_{HS} \le \|A\|_{HS} \|B\|.$$

Note that in our setting we have

$$\kappa \ge \int_X K(x,x)d\rho(x) = \int_X \sum_{i=1}^{\infty} \sigma_i |\phi_i(x)|^2 d\rho(x) = \sum_{i=1}^{\infty} \sigma_i \int_X |\phi_i(x)|^2 d\rho(x)$$

$$= \sum_{i=1}^{\infty} \sigma_i = Tr(T_K),$$

where the interchange of integration and summation is justified by the monotone convergence theorem. Thus $T_K$ is a trace class operator and we have the inequality

$$\kappa \ge \sum_{i=1}^{\infty} \sigma_i$$

which will prove useful later.

### C.3 Damped Deviations

Let $x \mapsto g_s(x) \in L^2$ for each $s \in [0,t]$ such that $s \mapsto \langle \phi_i, g_s \rangle_2$ is measureable for each $i$ and $\int_0^t \|g_s\|_2^2 < \infty$. Then we define the integral

$$\int_0^t g_s ds$$

coordinatewise, meaning that $\int_0^t g_s ds$ is the $L^2$ function $h$ such that

$$\langle h, \phi_i \rangle_2 = \int_0^t \langle g_s, \phi_i \rangle_2 ds.$$

Using this definition, we can now prove the "Damped Deviations" lemma.

**Lemma 2.3.** *Let $K(x,x')$ be an arbitrary continuous, symmetric, positive-definite kernel. Let $[T_K h](\bullet) = \int_X K(\bullet, s)h(s)d\rho(s)$ be the integral operator associated with $K$ and let $[T_n^s h](\bullet) = \frac{1}{n}\sum_{i=1}^{n} K_s(\bullet, x_i)h(x_i)$ denote the operator associated with the time-dependent NTK $K_s$. Then*

$$r_t = \exp(-T_K t)r_0 + \int_0^t \exp(-T_K(t-s))(T_K - T_n^s)r_s ds,$$

*where the equality is in the $L^2$ sense.*

*Proof.* We have that

$$\partial_s r_s(x) = -\frac{1}{n}\sum_{i=1}^{n} K_s(x, x_i)r_s(x_i) = -[T_n^s r_s](x),$$

where the equality is pointwise over $x$. $\partial_s r_s(x)$ is a continuous function of $x$ since $K_s$ is continuous and is thus in $L^2$. Therefore we can consider

$$\langle \partial_s r_s, \phi_i \rangle_2 = \langle -T_n^s r_s, \phi_i \rangle_2.$$

By the continuity of $s \mapsto \theta_s$ we have the parameters are locally bounded in time and thus by Lemma B.5 we have that $\|K_s\|_\infty$ is also locally bounded therefore for any $\delta > 0$, $s_0$: $\sup_{|s-s_0|\leq\delta} \|K_s\|_\infty < \infty$. Note then that

$$|\partial_s r_s(x)| \leq \frac{1}{n} \sum_{i=1}^n |K_s(x, x_i)||r_s(x_i)| \leq \|K_s\|_\infty \|\hat{r}_s\|_{\mathbb{R}^n} \leq \|K_s\|_\infty \|\hat{r}_0\|_{\mathbb{R}^n}.$$

It follows that $\|\partial_s r_s\|_\infty$ is bounded locally uniformly in $s$. Therefore the following differentiation under the integral sign is justified

$$\frac{d}{ds}\langle r_s, \phi_i \rangle_2 = \langle \partial_s r_s, \phi_i \rangle_2.$$

Thus combined with our previous equality we get

$$\frac{d}{ds}\langle r_s, \phi_i \rangle_2 = \langle -T_n^s r_s, \phi_i \rangle_2 = \langle -T_K r_s, \phi_i \rangle_2 + \langle (T_K - T_n^s)r_s, \phi_i \rangle_2$$

$$= \langle r_s, -T_K \phi_i \rangle_2 + \langle (T_K - T_n^s)r_s, \phi_i \rangle_2 = -\sigma_i \langle r_s, \phi_i \rangle_2 + \langle (T_K - T_n^s)r_s, \phi_i \rangle_2,$$

where we have used that $T_K$ is self-adjoint. Therefore

$$\frac{d}{ds}\langle r_s, \phi_i \rangle_2 + \sigma_i \langle r_s, \phi_i \rangle_2 = \langle (T_K - T_n^s)r_s, \phi_i \rangle_2.$$

Multiplying by the integrating factor $\exp(\sigma_i s)$ we get

$$\frac{d}{ds}[\exp(\sigma_i s)\langle r_s, \phi_i \rangle_2] = \exp(\sigma_i s)\langle (T_K - T_n^s)r_s, \phi_i \rangle_2.$$

Therefore applying the fundamental theorem of calculus after rearrangement we get

$$\langle r_t, \phi_i \rangle_2 = \exp(-\sigma_i t)\langle r_0, \phi_i \rangle_2 + \int_0^t \exp(-\sigma_i(t-s))\langle (T_K - T_n^s)r_s, \phi_i \rangle_2 ds,$$

which is just the coordinatewise version of the desired result.

$$r_t = \exp(-T_K t)r_0 + \int_0^t \exp(-T_K(t-s))(T_K - T_n^s)r_s ds.$$

$\square$

## C.4 COVERING NUMBER OF CLASS

We will now estimate the covering number of the class of shallow networks with bounds on their parameter norms. This lemma is slightly more general than what we will use but we will particularize it latter as it's general formulation presents no additional difficulty.

**Lemma C.1.** *Let*

$$\mathcal{C} = \{ \frac{a^T}{\sqrt{m}}\sigma(Wx+b) + b_0 : \|a - a'\|_2 \leq \rho_1, \|W - W'\|_F \leq \rho_2,$$

$$\|b - b'\|_2 \leq \rho_3, |b_0 - b'_0| \leq \rho_4$$

$$\frac{1}{\sqrt{m}}\|a\|_2 \leq \rho'_1, \frac{1}{\sqrt{m}}\|W\|_{op} \leq \rho'_2, \frac{1}{\sqrt{m}}\|b\|_2 \leq \rho'_3 \}$$

*and*

$$\gamma' := |\sigma(0)| + \|\sigma'\|_\infty [\rho'_2 M + \rho'_3].$$

*Then the (proper) covering number of $\mathcal{C}$ in the uniform norm satisfies*

$$\mathcal{N}(\mathcal{C}, \epsilon, \|\|_\infty) \leq \left(\frac{C'}{\epsilon}\right)^p$$

*where $p = md + 2m + 1$ is the total number of parameters and $C'$ equals*

$$C' = C\max\{\rho_1\gamma', \rho_2 M \|\sigma'\|_\infty \rho'_1, \rho_3 \|\sigma'\|_\infty \rho'_1, \rho_4\},$$

*where $C > 0$ is an absolute constant.*

*Proof.* We will bound the pertubation of the function when changing the weights, specifically we will bound

$$\sup_{x \in B_M} \left| \frac{a^T}{\sqrt{m}} \sigma(Wx + b) - \frac{\tilde{a}^T}{\sqrt{m}} \sigma(\tilde{W}x + \tilde{b}) \right|.$$

Let $x^{(1)} = \frac{1}{\sqrt{m}} \sigma(Wx + b)$ and $\tilde{x}^{(1)} = \frac{1}{\sqrt{m}} \sigma(\tilde{W}x + \tilde{b})$. Then note that we have

$$\left\| x^{(1)} - \tilde{x}^{(1)} \right\|_2 \leq \frac{1}{\sqrt{m}} \left\| \sigma' \right\|_\infty \left\| (W - \tilde{W})x + b - \tilde{b} \right\|_2$$

$$\leq \frac{1}{\sqrt{m}} \left\| \sigma' \right\|_\infty \left[ \left\| W - \tilde{W} \right\|_{op} \|x\|_2 + \left\| b - \tilde{b} \right\|_2 \right]$$

$$\leq \frac{1}{\sqrt{m}} \left\| \sigma' \right\|_\infty \left[ \left\| W - \tilde{W} \right\|_F M + \left\| b - \tilde{b} \right\|_2 \right] =: \gamma.$$

Well then

$$|a^T x^{(1)} - \tilde{a}^T \tilde{x}^{(1)}| \leq |a^T(x^{(1)} - \tilde{x}^{(1)})| + |(a - \tilde{a})^T \tilde{x}^{(1)}|$$

$$\leq \|a\|_2 \gamma + \|a - \tilde{a}\|_2 \left\| \tilde{x}^{(1)} \right\|_2.$$

Finally

$$\left\| \tilde{x}^{(1)} \right\|_2 = \left\| \frac{1}{\sqrt{m}} \sigma(\tilde{W}x + \tilde{b}) \right\|_2 \leq |\sigma(0)| + \frac{1}{\sqrt{m}} \left\| \sigma' \right\|_\infty \left\| \tilde{W}x + \tilde{b} \right\|_2$$

$$\leq |\sigma(0)| + \frac{1}{\sqrt{m}} \left\| \sigma' \right\|_\infty \left[ \left\| \tilde{W} \right\|_{op} \|x\|_2 + \left\| \tilde{b} \right\|_2 \right]$$

$$\leq |\sigma(0)| + \frac{1}{\sqrt{m}} \left\| \sigma' \right\|_\infty \left[ \left\| \tilde{W} \right\|_{op} M + \left\| \tilde{b} \right\|_2 \right]$$

$$\leq |\sigma(0)| + \left\| \sigma' \right\|_\infty [\rho_2' M + \rho_3'] =: \gamma'.$$

Therefore

$$|a^T x^{(1)} - \tilde{a}^T \tilde{x}^{(1)}| \leq \|a\|_2 \gamma + \|a - \tilde{a}\|_2 \gamma'.$$

Thus if we have

$$\|a - \tilde{a}\|_2 \leq \frac{\epsilon}{4\gamma'} =: \epsilon_1, \quad \left\| W - \tilde{W} \right\|_F \leq \frac{\epsilon}{8M \left\| \sigma' \right\|_\infty \rho_1'} =: \epsilon_2, \quad \left\| b - \tilde{b} \right\|_2 \leq \frac{\epsilon}{8 \left\| \sigma' \right\|_\infty \rho_1'} =: \epsilon_3,$$

then

$$\|a\|_2 \gamma \leq \frac{\epsilon \|a\|_2}{4\rho_1' \sqrt{m}} \leq \frac{\epsilon}{4}.$$

Therefore

$$|a^T x^{(1)} - \tilde{a}^T \tilde{x}^{(1)}| \leq \epsilon/2$$

and this bound holds for any $x \in B_M$. If add biases $b_0$ and $\tilde{b}_0$ such that $|b_0 - \tilde{b}_0| \leq \epsilon/2$ we simply get by the triangle inequality

$$|a^T x^{(1)} + b_0 - (\tilde{a}^T \tilde{x}^{(1)} + \tilde{b}_0)| \leq \epsilon.$$

Thus to get a cover we can simply cover the sets

$$\{a : \|a - a'\|_2 \leq \rho_1\} \quad \{W : \|W - W'\|_F \leq \rho_2\}$$

$$\{b : \|b - b'\|_2 \leq \rho_3\} \quad \{b_0 : |b_0 - b_0'| \leq \rho_4\}$$

in the Euclidean norm and multiply the covering numbers. Recall that the $\epsilon$ covering number for a Euclidean ball of radius $R$ in $\mathbb{R}^s$, say $\mathcal{N}_\epsilon$, using the Euclidean norm satisfies

$$\left( \frac{cR}{\epsilon} \right)^s \leq \mathcal{N}_\epsilon \leq \left( \frac{CR}{\epsilon} \right)^s$$

for two absolute constants $c, C > 0$. Therefore we get that

$$\mathcal{N}(\mathcal{C}, \epsilon, \|\cdot\|_\infty) \leq \left( \frac{C\rho_1}{\epsilon_1} \right)^m \left( \frac{C\rho_2}{\epsilon_2} \right)^{md} \left( \frac{C\rho_3}{\epsilon_3} \right)^m \left( \frac{2C\rho_4}{\epsilon} \right).$$

The desired result follows from

$$\max\left\{\frac{\rho_1}{\epsilon_1}, \frac{\rho_2}{\epsilon_2}, \frac{\rho_3}{\epsilon_3}, \frac{2\rho_4}{\epsilon}\right\} \le \frac{C}{\epsilon}\max\left\{\rho_1\gamma', \rho_2 M\|\sigma'\|_\infty \rho_1', \rho_3\|\sigma'\|_\infty \rho_1', \rho_4\right\}.$$

$\square$

We can now prove the following corollary which is the version of the previous lemma that we will actually use for our neural network.

**Corollary C.2.** *Let*

$$\mathcal{C} = \{\,\frac{a^T}{\sqrt{m}}\sigma(Wx+b) + b_0 : \frac{1}{\sqrt{m}}\|a\|_2, \frac{1}{\sqrt{m}}\|W\|_{op}, \frac{1}{\sqrt{m}}\|b\|_2 \le A, |b_0| \le B\,\}$$

*and*

$$\gamma' := |\sigma(0)| + \|\sigma'\|_\infty [AM + A]$$

*and assume $m \ge d$. Then the (proper) covering number of $\mathcal{C}$ in the uniform norm satisfies*

$$\mathcal{N}(\mathcal{C}, \epsilon, \|\|_\infty) \le \left(\frac{\Psi(m,d)}{\epsilon}\right)^p,$$

*where*

$$\Psi(m,d) = C\max\{\sqrt{m}A\gamma', \sqrt{md}A^2\|\sigma'\|_\infty M, \sqrt{m}A^2\|\sigma'\|_\infty, B\}$$
$$= \sqrt{md}O\left(\max\left\{A^2, \frac{B}{\sqrt{md}}\right\}\right).$$

*Proof.* The idea is to apply Lemma C.1 with $a' = 0$, $W' = 0$, $b' = 0$, and $b_0' = 0$. Note that $\|W\|_F \le \sqrt{d}\|W\|_{op} \le \sqrt{md}A$. The result then follows by applying lemma with $\rho_1 = \sqrt{m}A$, $\rho_2 = \sqrt{md}A$, $\rho_3 = \sqrt{m}A$ and $\rho_4 = B$ and $\rho_1' = \rho_2' = \rho_3' = A$. $\square$

## C.5 Uniform Convergence Over The Class

We now show that $\|(T_n - T_{K^\infty})g\|_2$ is uniformly small for all $g$ in a suitable class of functions $\mathcal{C}'$. Ultimately we will show that $r_t \in \mathcal{C}'$ and thus this result is towards proving that $\|(T_n - T_{K^\infty})r_t\|_2$ is small.

**Lemma C.3.** *Let $K(x,x')$ by a continuous, symmetric, positive-definite kernel and let $\kappa = \max_{x\in X}K(x,x) < \infty$. Let $T_K h(\bullet) = \int_X K(\bullet, s)h(s)d\rho(s)$ and $T_n h(\bullet) = \frac{1}{n}\sum_{i=1}^n K(\bullet, x_i)h(x_i)$ be the associated operators. Let $\sigma_1$ denote the largest eigenvalue of $T_K$. Let $\mathcal{C}$ and $\Psi(m,d)$ be defined as in Corollary C.2. We let $\mathcal{C}' = \{g - f^* : g \in \mathcal{C}\} \cap \{g : \|g\|_\infty \le S\}$ be the set where $\mathcal{C}$ is translated by the target function $f^*$ then intersected with the $L^\infty$ ball of radius $S > 0$. Then with probability at least $1 - \delta$ over the sampling of $x_1, \ldots, x_n$*

$$\sup_{g\in\mathcal{C}'}\|(T_n - T_K)g\|_2 \le \frac{2S\sqrt{\sigma_1\kappa}\sqrt{2\log(c/\delta) + 2p\log(\|K\|_\infty \Psi(m,d)\sqrt{n})}}{\sqrt{n}} + \frac{2}{\sqrt{n}}$$

$$= \frac{2\left[1 + S\sqrt{\sigma_1\kappa}\sqrt{2\log(c/\delta) + \tilde{O}(p)}\right]}{\sqrt{n}}.$$

*Proof.* Let $g \in \mathcal{C}'$. We introduce the random variables $Y_i := K_{x_i}g(x_i) - \mathbb{E}_{x\sim\rho}[K_x g(x)]$ taking values in the Hilbert space $\mathcal{H}$ for $i \in [n]$ where $\mathcal{H}$ is the RKHS associated with $K$. Note that for any $x$

$$\|K_x g(x)\|_\mathcal{H} = |g(x)|\sqrt{\langle K_x, K_x\rangle_\mathcal{H}} \le S\sqrt{K(x,x)} \le S\kappa^{1/2}.$$

Thus $\|Y_i\|_\mathcal{H} \le 2S\kappa^{1/2}$ a.s. Thus by Hoeffding's inequality for random variables taking values in a separable Hilbert space (see Rosasco et al. (2010, Section 2.4)) we have

$$\mathbb{P}\left(\left\|\frac{1}{n}\sum_{i=1}^n Y_i\right\|_\mathcal{H} > t\right) \le 2\exp\left(-nt^2/2[2S\kappa^{1/2}]^2\right).$$

Note that by basic properties of the covering number we have that $\mathcal{N}(\mathcal{C}', \epsilon, \|\|_\infty) \leq \mathcal{N}(\mathcal{C}, \epsilon/2, \|\|_\infty)$, thus by Corollary C.2 the covering number of $\mathcal{C}'$ satisfies (up to a redefinition of $C$)

$$\mathcal{N}(\mathcal{C}', \epsilon, \|\|_\infty) \leq \left(\frac{\Psi(m, d)}{\epsilon}\right)^p.$$

Let $\Delta$ be an $\epsilon$ net of $\mathcal{C}'$ in the uniform norm. Note that $\frac{1}{n}\sum_{i=1}^n Y_i = (T_n - T_K)g$. Thus by taking a union bound we have

$$\mathbb{P}\left(\max_{g \in \Delta} \|(T_n - T_K)g\|_\mathcal{H} \geq t\right) \leq \left(\frac{\Psi(m, d)}{\epsilon}\right)^p 2\exp\left(-nt^2/2[2S\kappa^{1/2}]^2\right).$$

Note that for any probability measure $\nu$ and $h \in L^\infty$

$$\left|\int_X K(x, s)h(s)d\nu(s)\right| \leq \int_X |K(x, s)||h(s)|d\nu(s) \leq \|K\|_\infty \|h\|_\infty.$$

It follows that for any $h \in L^\infty$

$$\|(T_K - T_n)h\|_\infty \leq 2\|K\|_\infty \|h\|_\infty.$$

Note for any $g \in \mathcal{C}'$ we can pick $\hat{g}$ in $\Delta$ such that $\|g - \hat{g}\|_\infty \leq \epsilon$. Then

$$\|(T_n - T_K)g\|_2 \leq \|(T_n - T_K)\hat{g}\|_2 + \|(T_n - T_K)(g - \hat{g})\|_2$$
$$\leq \sqrt{\sigma_1}\|(T_n - T_K)\hat{g}\|_\mathcal{H} + \|(T_n - T_K)(g - \hat{g})\|_\infty$$
$$\leq \sqrt{\sigma_1}t + 2\|K\|_\infty \|g - \hat{g}\|_\infty$$
$$\leq \sqrt{\sigma_1}t + 2\|K\|_\infty \epsilon,$$

where we have used the fact that $\|\bullet\|_2 \leq \sqrt{\sigma_1}\|\bullet\|_\mathcal{H}$ and $\|\bullet\|_2 \leq \|\bullet\|_\infty$ in the second inequality. Thus by setting

$$t = \frac{2S\kappa^{1/2}\sqrt{2\log(c/\delta) + 2p\log(\Psi(m, d)/\epsilon)}}{\sqrt{n}}$$

we have with probability at least $1 - \delta$

$$\sup_{g \in \mathcal{C}'} \|(T_n - T_K)g\|_2 \leq$$

$$\sqrt{\sigma_1}\frac{2S\kappa^{1/2}\sqrt{2\log(c/\delta) + 2p\log(\Psi(m, d)/\epsilon)}}{\sqrt{n}} + 2\|K\|_\infty \epsilon.$$

This argument runs through for any $\epsilon > 0$. Thus by setting $\epsilon = \frac{1}{\|K\|_\infty \sqrt{n}}$ we get the desired result. $\square$

## C.6 NEURAL NETWORK IS IN THE CLASS

In this section we demonstrate that the neural network in such a class as $\mathcal{C}$ as defined in Lemma C.1. Once we have this we can use Lemma C.3 to show that $\|(T_{K^\infty} - T_n)r_t\|_2$ is uniformly small. The first step is to bound the parameter norms, hence the following lemma.

**Lemma C.4.** *Assume that $W_{i,j} \sim \mathcal{W}$, $b_\ell \sim \mathcal{B}$, $a_\ell \sim \mathcal{A}$ are all i.i.d zero-mean, subgaussian random variables with unit variance. Furthermore assume $\|1\|_{\psi_2}, \|w_\ell\|_{\psi_2}, \|a_\ell\|_{\psi_2}, \|b_\ell\|_{\psi_2} \leq K$ for each $\ell \in [m]$ where $K \geq 1$. Let $\Gamma > 1$, $T > 0$, $D := 3\max\{|\sigma(0)|, M\|\sigma'\|_\infty, \|\sigma'\|_\infty, 1\}$, and*

$$\xi(t) = \max\{\frac{1}{\sqrt{m}}\|W\|_{op}, \frac{1}{\sqrt{m}}\|b\|_2, \frac{1}{\sqrt{m}}\|a\|_2, 1\}.$$

*Furthermore assume*

$$m \geq \frac{4D^2\|y\|_{\mathbb{R}^n}^2 T^2}{[\log(\Gamma)]^2} \text{ and } m \geq \max\left\{\frac{4D^2 O(\log(c/\delta) + \tilde{O}(d))T^2}{[\log(\Gamma)]^2}, O(\log(c/\delta) + \tilde{O}(d))\right\}.$$

*Then with probability at least $1 - \delta$*

$$\max_{t \in [0,T]} \xi(t) \leq \Gamma\left[1 + C\frac{\sqrt{d} + K^2\sqrt{\log(c/\delta)}}{\sqrt{m}}\right].$$

*If one instead does the doubling trick then the second condition on $m$ can be removed from the hypothesis and the same conclusion holds.*

*Proof.* First assume we are not doing the doubling trick. Note that the hypothesis on $m$ is strong enough to satisfy the hypothesis of Lemma B.23, therefore we have with probability at least $1 - \delta$

$$\max_{t \leq T} \xi(t) \leq \Gamma \xi(0).$$

Well then separately by Lemma B.17 with probability at least $1 - \delta$

$$\xi(0) \leq 1 + C \frac{\sqrt{d} + K^2 \sqrt{\log(c/\delta)}}{\sqrt{m}}.$$

Thus by replacing $\delta$ with $\delta/2$ in the previous statements and taking a union bound we have with probability at least $1 - \delta$

$$\max_{t \in [0,T]} \xi(t) \leq \Gamma \left[ 1 + C \frac{\sqrt{d} + K^2 \sqrt{\log(c/\delta)}}{\sqrt{m}} \right]$$

which is the desired result. Now suppose instead one does the doubling trick. We recall that the doubling trick does not change $\xi(0)$. Thus we can run through the exact same argument as before except when we apply Lemma B.23 we can remove the second condition on $m$ from the hypothesis. $\square$

The following lemma bounds the bias term.

**Lemma C.5.** *For any initial conditions we have*

$$|b_0(t)| \leq |b_0(0)| + t \left\| \hat{r}(0) \right\|_{\mathbb{R}^n}.$$

*Proof.* Note that

$$|\partial_t b_0(t)| = \left| \frac{1}{n} \sum_{i=1}^n \hat{r}(t)_i \right| \leq \left\| \hat{r}(t) \right\|_{\mathbb{R}^n} \leq \left\| \hat{r}(0) \right\|_{\mathbb{R}^n}$$

Thus by the fundamental theorem of calculus

$$|b_0(t)| \leq |b_0(0)| + t \left\| \hat{r}(0) \right\|_{\mathbb{R}^n}.$$

$\square$

The following lemma demonstrates that the residual $r_t = f_t - f^*$ is bounded.

**Lemma C.6.** *Assume that $W_{i,j} \sim \mathcal{W}$, $b_\ell \sim \mathcal{B}$, $a_\ell \sim \mathcal{A}$ are all i.i.d zero-mean, subgaussian random variables with unit variance. Furthermore assume $\|1\|_{\psi_2}, \|w_\ell\|_{\psi_2}, \|a_\ell\|_{\psi_2}, \|b_\ell\|_{\psi_2} \leq K$ for each $\ell \in [m]$ where $K \geq 1$. Let $\Gamma > 1$, $T > 0$, $D := 3 \max\{|\sigma(0)|, M \|\sigma'\|_\infty, \|\sigma'\|_\infty, 1\}$, and assume*

$$m \geq \frac{4D^2 \|y\|_{\mathbb{R}^n}^2 T^2}{[\log(\Gamma)]^2} \text{ and } m \geq \max \left\{ \frac{4D^2 O(\log(c/\delta) + \tilde{O}(d))T^2}{[\log(\Gamma)]^2}, O(\log(c/\delta) + \tilde{O}(d)) \right\}.$$

*Then with probability at least $1 - \delta$ for $t \leq T$*

$$\|f_t - f^*\|_\infty \leq \|f_0 - f^*\|_\infty + t \left\| \hat{r}(0) \right\|_{\mathbb{R}^n} C D^2 \Gamma^2 \left[ 1 + C \frac{\sqrt{d} + K^2 \sqrt{\log(c/\delta)}}{\sqrt{m}} \right]^2$$

*If one instead does the doubling trick then the second condition on $m$ can be removed from the hypothesis and the same conclusion holds.*

*Proof.* Recall that

$$\partial_t(f_t(x) - f^*(x)) = -\frac{1}{n} \sum_{i=1}^n K_t(x, x_i)(f_t(x_i) - f^*(x_i)) = -\frac{1}{n} \sum_{i=1}^n K_t(x, x_i) \hat{r}(t)_i.$$

Thus

$$|\partial_t(f_t(x) - f^*(x))| \leq \frac{1}{n} \sum_{i=1}^n |K_t(x, x_i)| |\hat{r}(t)_i| \leq \|K_t\|_\infty \|\hat{r}(t)\|_{\mathbb{R}^n} \leq \|K_t\|_\infty \|\hat{r}(0)\|_{\mathbb{R}^n}.$$

Well by Lemma B.5 we have that $\|K_t\|_\infty \leq CD^2\xi^2(t)$ where

$$\xi(t) = \max\{\frac{1}{\sqrt{m}}\|W\|_{op}, \frac{1}{\sqrt{m}}\|b\|_2, \frac{1}{\sqrt{m}}\|a\|_2, 1\}.$$

Well by Lemma C.4 we have that with probability at least $1 - \delta$

$$\max_{t \in [0,T]} \xi(t) \leq \Gamma \left[1 + C\frac{\sqrt{d} + K^2\sqrt{\log(c/\delta)}}{\sqrt{m}}\right].$$

Thus by the fundamental theorem of calculus for $t \leq T$

$$|f_t(x) - f^*(x)| \leq |f_0(x) - f^*(x)| + t\|\hat{r}(0)\|_{\mathbb{R}^n} CD^2\Gamma^2\left[1 + C\frac{\sqrt{d} + K^2\sqrt{\log(c/\delta)}}{\sqrt{m}}\right]^2$$

Thus by taking the supremum over $x$ we get

$$\|f_t - f^*\|_\infty \leq \|f_0 - f^*\|_\infty + t\|\hat{r}(0)\|_{\mathbb{R}^n} CD^2\Gamma^2\left[1 + C\frac{\sqrt{d} + K^2\sqrt{\log(c/\delta)}}{\sqrt{m}}\right]^2$$

which is the desired conclusion. $\qquad\square$

We can now finally prove that $\|(T_{K^\infty} - T_n)r_t\|_2$ is uniformly small.

**Lemma C.7.** *Let $K(x,x')$ by a continuous, symmetric, positive-definite kernel and let $\kappa = \max_x K(x,x) < \infty$. Let $T_K h(\bullet) = \int_X K(\bullet, s)h(s)d\rho(s)$ and $T_n h(\bullet) = \frac{1}{n}\sum_{i=1}^n K(\bullet, x_i)h(x_i)$ be the associated operators. Assume that $W_{i,j} \sim \mathcal{W}$, $b_\ell \sim \mathcal{B}$, $a_\ell \sim \mathcal{A}$ are all i.i.d zero-mean, subgaussian random variables with unit variance. Furthermore assume $\|1\|_{\psi_2}, \|w_\ell\|_{\psi_2}, \|a_\ell\|_{\psi_2}, \|b_\ell\|_{\psi_2}, \|b_0\|_{\psi_2} \leq K'$ for each $\ell \in [m]$ where $K' \geq 1$. Let $\Gamma > 1$, $T > 0$, $D := 3\max\{|\sigma(0)|, M\|\sigma'\|_\infty, \|\sigma'\|_\infty, 1\}$, and assume*

$$m \geq \frac{4D^2\|y\|_{\mathbb{R}^n}^2 T^2}{[\log(\Gamma)]^2} \text{ and } m \geq \max\left\{\frac{4D^2 O(\log(c/\delta) + \tilde{O}(d))T^2}{[\log(\Gamma)]^2}, O(\log(c/\delta) + \tilde{O}(d))\right\}.$$

*If we are doing the doubling trick set $S' = 0$ and otherwise set*

$$S' = CD(K')^2\sqrt{d\log(CM\tilde{O}(\sqrt{m})) + \log(c/\delta)} = \tilde{O}(\sqrt{d}).$$

*Then with probability at least $1 - \delta$*

$$\sup_{t \leq T}\|(T_n - T_K)r_t\|_2 = \tilde{O}\left(\frac{(\|f^*\|_\infty + S')(1 + T\Gamma^2)\sqrt{\sigma_1\kappa p}}{\sqrt{n}}\right)$$

*and*

$$\|r_0\|_\infty \leq \|f^*\|_\infty + S'.$$

*If we are performing the doubling trick the second condition on $m$ can be removed and the same conclusion holds.*

*Proof.* By Lemma C.4 we have with probability at least $1 - \delta$

$$\max_{t \in [0,T]} \xi(t) \leq \Gamma\left[1 + C\frac{\sqrt{d} + (K')^2\sqrt{\log(c/\delta)}}{\sqrt{m}}\right] =: A. \tag{8}$$

Also by Lemma C.5

$$|b_0(t)| \leq |b_0(0)| + t\|\hat{r}(0)\|_{\mathbb{R}^n}.$$

If we are doing the doubling trick then $b_0(0) = 0$. Otherwise by Lemma B.15 we have with probability at least $1 - \delta$

$$|b_0(0)| \leq CK'\sqrt{\log(c/\delta)}.$$

Furthermore by Lemma B.22 we have

$$\|\hat{r}(0)\|_{\mathbb{R}^n} \leq \|y\|_{\mathbb{R}^n} + \|f(\bullet; \theta_0)\|_\infty \,.$$

Let $L$ be defined as in Lemma B.21, i.e.

$$L(m, \sigma, d, K', \delta) := \sqrt{m}\,\|\sigma'\|_\infty \left\{ 1 + C\frac{\sqrt{d} + (K')^2\sqrt{\log(c/\delta)}}{\sqrt{m}} \right\}^2 = \tilde{O}(\sqrt{m}).$$

If we are not performing the doubling trick set

$$S' = CD(K')^2 \sqrt{d\log(CML) + \log(c/\delta)}.$$

Otherwise if we are performing the doubling trick set $S' = 0$. In either case by Lemma B.21 we have with probability at least $1 - \delta$

$$\|f(\bullet; \theta_0)\|_\infty \leq S'. \tag{9}$$

In particular by Lemma B.22 we have

$$\|\hat{r}(0)\|_{\mathbb{R}^n} \leq \|y\|_{\mathbb{R}^n} + \|f(\bullet; \theta_0)\|_\infty \leq \|y\|_{\mathbb{R}^n} + S'.$$

Thus we can say

$$|b_0(t)| \leq |b_0(0)| + t\,\|\hat{r}(0)\|_{\mathbb{R}^n} \leq CK'\sqrt{\log(c/\delta)} + T(\|y\|_{\mathbb{R}^n} + S') =: B$$

and this holds whether or not we are performing the doubling trick. Thus up until time $T$ the neural network is in class $\mathcal{C}$ as defined in Corollary C.2 with parameters $A$ and $B$ as defined above. Moreover by Lemma C.6 separate from the randomness before we have that with probability at least $1 - \delta$

$$\|r_t\|_\infty \leq \|r_0\|_\infty + t\,\|\hat{r}(0)\|_{\mathbb{R}^n}\, CD^2\Gamma^2 \left[ 1 + C\frac{\sqrt{d} + (K')^2\sqrt{\log(c/\delta)}}{\sqrt{m}} \right]^2$$

Well note that when (9) holds we have

$$\|\hat{r}(0)\|_{\mathbb{R}^n} \leq \|r_0\|_\infty \leq \|f^*\|_\infty + \|f(\bullet; \theta_0)\|_\infty \leq \|f^*\|_\infty + S'.$$

Thus

$$\|r_t\|_\infty \leq (\|f^*\|_\infty + S')\left\{ 1 + TCD^2\Gamma^2 \left[ 1 + C\frac{\sqrt{d} + (K')^2\sqrt{\log(c/\delta)}}{\sqrt{m}} \right]^2 \right\} =: S$$

Thus by taking a union bound and redefining $\delta$ we have by an application of Lemma C.3 with $S$ as defined in the hypothesis of the current theorem that with probability at least $1 - \delta$

$$\sup_{t \leq T} \|(T_n - T_K)r_t\|_2 \leq \frac{2\left[1 + S\sqrt{\sigma_1 \kappa}\sqrt{2\log(c/\delta) + \tilde{O}(p)}\right]}{\sqrt{n}}$$

$$= \tilde{O}\left( \frac{(\|f^*\|_\infty + S')(1 + T\Gamma^2)\sqrt{\sigma_1 \kappa p}}{\sqrt{n}} \right)$$

where we have used that $S = \tilde{O}([\|f^*\|_\infty + S'][1 + T\Gamma^2])$. $\qquad\square$

## C.7 PROOF OF THEOREM 3.5

We are almost ready to prove Theorem 3.5. However first we must introduce a couple lemmas. The following lemma uses the damped deviations equation to bound the difference between $r_t$ and $\exp(-T_K t)r_0$.

**Lemma C.8.** *Let $K(x, x')$ by a continuous, symmetric, positive-definite kernel with associated operator $T_K h(\bullet) = \int_X K(\bullet, s)h(s)d\rho(s)$. Let $T_n^s h(\bullet) = \frac{1}{n}\sum_{i=1}^n K_s(\bullet, x_i)h(x_i)$ denote the operator associated with the time-dependent NTK. Then*

$$\|P_k(r_t - \exp(-T_K t)r_0)\|_2 \leq \frac{1 - \exp(-\sigma_k t)}{\sigma_k} \sup_{s \leq t} \|(T_K - T_n^s)r_s\|_2 \,.$$

*and*

$$\|r_t - \exp(-T_K t)r_0\|_2 \leq t \cdot \sup_{s \leq t} \|(T_K - T_n^s)r_s\|_2 \,.$$

*Proof.* From Lemma 2.3 we have

$$r_t = \exp(-T_K t)r_0 + \int_0^t \exp(-T_K(t-s))(T_K - T_n^s)r_s ds.$$

Thus for any $k \in \mathbb{N}$

$$P_k(r_t - \exp(-T_K t)r_0) = P_k \int_0^t \exp(-T_K(t-s))(T_K - T_n^s)r_s ds$$

$$= \int_0^t P_k \exp(-T_K(t-s))(T_K - T_n^s)r_s ds.$$

Therefore

$$\|P_k(r_t - \exp(-T_K t)r_0)\|_2 = \left\| \int_0^t P_k \exp(-T_K(t-s))(T_K - T_n^s)r_s ds \right\|_2$$

$$\leq \int_0^t \|P_k \exp(-T_K(t-s))(T_K - T_n^s)r_s\|_2 \, ds$$

$$\leq \int_0^t \|P_k \exp(-T_K(t-s))\| \, \|(T_K - T_n^s)r_s\|_2 \, ds$$

$$\leq \int_0^t \exp(-\sigma_k(t-s)) \, \|(T_K - T_n^s)r_s\|_2 \, ds$$

$$\leq \frac{1 - \exp(-\sigma_k t)}{\sigma_k} \sup_{s \leq t} \|(T_K - T_n^s)r_s\|_2 \, .$$

Similarly

$$\|r_t - \exp(-T_K t)r_0\|_2 = \left\| \int_0^t \exp(-T_K(t-s))(T_K - T_n^s)r_s ds \right\|_2$$

$$\leq \int_0^t \|\exp(-T_K(t-s))(T_K - T_n^s)r_s\|_2 \, ds$$

$$\leq \int_0^t \|\exp(-T_K(t-s))\| \, \|(T_K - T_n^s)r_s\|_2 \, ds$$

$$\leq \int_0^t \|(T_K - T_n^s)r_s\|_2 \, ds \leq t \cdot \sup_{s \leq t} \|(T_K - T_n^s)r_s\|_2 \, .$$

$\square$

In light of the previous lemma we would like to have a bound for $\|(T_K - T_n^s)r_s\|_2$. This is accomplished by the following lemma.

**Lemma C.9.** *Let $K(x,x')$ by a continuous, symmetric, positive-definite kernel. Let $T_K h(\bullet) = \int_X K(\bullet, s)h(s)d\rho(s)$ and $T_n h(\bullet) = \frac{1}{n}\sum_{i=1}^n K(\bullet, x_i)h(x_i)$ be the associated operators. Let $T_n^s h(\bullet) = \frac{1}{n}\sum_{i=1}^n K_s(\bullet, x_i)h(x_i)$ denote the operator associated with the time-dependent NTK. Then*

$$\sup_{s \leq T} \|(T_K - T_n^s)r_s\|_2 \leq \sup_{s \leq T} \|(T_K - T_n)r_s\|_2 + \sup_{s \leq T} \|K - K_s\|_\infty \|\hat{r}(0)\|_{\mathbb{R}^n} \, .$$

*Proof.* We have that

$$\|(T_K - T_n^s)r_s\|_2 \leq \|(T_K - T_n)r_s\|_2 + \|(T_n - T_n^s)r_s\|_2 \, .$$

Now observe that

$$|(T_n - T_n^s)r_s(x)| = \left| \frac{1}{n}\sum_{i=1}^n [K(x,x_i) - K_s(x,x_i)]r_s(x_i) \right|$$

$$\leq \frac{1}{n}\sum_{i=1}^n |K(x,x_i) - K_s(x,x_i)||r_s(x_i)|$$

$$\leq \|K - K_s\|_\infty \|\hat{r}(s)\|_{\mathbb{R}^n} \leq \|K - K_s\|_\infty \|\hat{r}(0)\|_{\mathbb{R}^n} \, .$$

Therefore
$$\|(T_n - T_n^s)r_s\|_2 \leq \|(T_n - T_n^s)r_s\|_\infty \leq \|K - K_s\|_\infty \|\hat{r}(0)\|_{\mathbb{R}^n} .$$
Thus
$$\sup_{s \leq T} \|(T_K - T_n^s)r_s\|_2 \leq \sup_{s \leq T} \|(T_K - T_n)r_s\|_2 + \sup_{s \leq T} \|K - K_s\|_\infty \|\hat{r}(0)\|_{\mathbb{R}^n} .$$

$\square$

We are almost ready to finally prove Theorem 3.5. We must prove one final lemma that combines Lemma C.7 with the $NTK$ deviation bounds in Theorem B.26 to show that $\|(T_{K^\infty} - T_n^s)r_s\|_2$ is uniformly small.

**Lemma C.10.** *Assume that $W_{i,j} \sim \mathcal{W}$, $b_\ell \sim \mathcal{B}$, $a_\ell \sim \mathcal{A}$ are all i.i.d zero-mean, subgaussian random variables with unit variance. Furthermore assume $\|1\|_{\psi_2}, \|w_\ell\|_{\psi_2}, \|a_\ell\|_{\psi_2}, \|b_\ell\|_{\psi_2} \leq K$ for each $\ell \in [m]$ where $K \geq 1$. Let $\Gamma > 1$, $T > 0$, $D := 3\max\{|\sigma(0)|, M\|\sigma'\|_\infty, \|\sigma'\|_\infty, 1\}$, and assume*
$$m \geq \frac{4D^2 \|y\|_{\mathbb{R}^n}^2 T^2}{[\log(\Gamma)]^2} \text{ and } m \geq \max\left\{ \frac{4D^2 O(\log(c/\delta) + \tilde{O}(d))T^2}{[\log(\Gamma)]^2}, O(\log(c/\delta) + \tilde{O}(d)) \right\}.$$
*If we are doing the doubling trick set $S' = 0$ and otherwise set*
$$S' = CDK^2\sqrt{d\log(CM\tilde{O}(\sqrt{m})) + \log(c/\delta)} = \tilde{O}(\sqrt{d}), S = S' + \|f^*\|_\infty .$$
*Then with probability at least $1 - \delta$*
$$\sup_{s \leq t} \|(T_{K^\infty} - T_n^s)r_s\|_2 = \tilde{O}\left( S\frac{\sqrt{d}}{\sqrt{m}}\left[1 + t\Gamma^3 S\right] + \frac{S(1 + T\Gamma^2)\sqrt{\sigma_1 \kappa p}}{\sqrt{n}} \right).$$
*If we are performing the doubling trick the condition $m \geq \frac{4D^2 O(\log(c/\delta) + \tilde{O}(d))T^2}{[\log(\Gamma)]^2}$ can be removed and the same conclusion holds.*

*Proof.* Note by Lemma C.9 we have
$$\sup_{s \leq T} \|(T_{K^\infty} - T_n^s)r_s\|_2 \leq \sup_{s \leq T} \|(T_{K^\infty} - T_n)r_s\|_2 + \sup_{s \leq T} \|K^\infty - K_s\|_\infty \|\hat{r}(0)\|_{\mathbb{R}^n} .$$
Well then by Theorem B.26 we have with probability at least $1 - \delta$ that
$$\sup_{t \leq T} \|K_t - K^\infty\|_\infty = \tilde{O}\left( \frac{\sqrt{d}}{\sqrt{m}}\left[1 + t\Gamma^3 \|\hat{r}(0)\|_{\mathbb{R}^n}\right] \right).$$
Separately by Lemma C.7 we have with probability at least $1 - \delta$
$$\sup_{s \leq T} \|(T_{K^\infty} - T_n)r_s\|_2 = \tilde{O}\left( \frac{(\|f^*\|_\infty + S')(1 + T\Gamma^2)\sqrt{\sigma_1 \kappa p}}{\sqrt{n}} \right) = \tilde{O}\left( \frac{S(1 + T\Gamma^2)\sqrt{\sigma_1 \kappa p}}{\sqrt{n}} \right)$$
and
$$\|\hat{r}(0)\|_{\mathbb{R}^n} \leq \|r_0\|_\infty \leq S.$$
The result follows then from taking a union bound and replacing $\delta$ with $\delta/2$. $\square$

We now proceed to prove the main theorem of this paper.

**Theorem 3.5.** *Assume that Assumptions 3.3 and 3.4 hold. Let $P_k$ be the orthogonal projection in $L^2$ onto $span\{\phi_1, \ldots, \phi_k\}$ and let $D := 3\max\{|\sigma(0)|, M\|\sigma'\|_\infty, \|\sigma'\|_\infty, 1\}$. If we are doing the doubling trick set $S' = 0$ and otherwise set $S' = O\left(\sqrt{\tilde{O}(d) + \log(c/\delta)}\right)$, $S = \|f^*\|_\infty + S'$. Also let $T > 0$. Assume $m \geq D^2 \|y\|_{\mathbb{R}^n}^2 T^2$, and $m \geq O(\log(c/\delta) + \tilde{O}(d))\max\{T^2, 1\}$. Then with probability at least $1 - \delta$ we have that for all $t \leq T$ and $k \in \mathbb{N}$*
$$\|P_k(r_t - \exp(-T_{K^\infty}t)r_0)\|_2 \leq \frac{1 - \exp(-\sigma_k t)}{\sigma_k}\tilde{O}\left( S[1 + tS]\frac{\sqrt{d}}{\sqrt{m}} + S(1 + T)\frac{\sqrt{p}}{\sqrt{n}} \right)$$
*and*
$$\|r_t - \exp(-T_{K^\infty}t)r_0\|_2 \leq t\tilde{O}\left( S[1 + tS]\frac{\sqrt{d}}{\sqrt{m}} + S(1 + T)\frac{\sqrt{p}}{\sqrt{n}} \right).$$

*Proof.* By Lemma C.8 we have for any $k \in \mathbb{N}$

$$\|P_k(r_t - \exp(-T_{K^\infty}t)r_0)\|_2 \leq \frac{1 - \exp(-\sigma_k t)}{\sigma_k} \sup_{s \leq t} \|(T_{K^\infty} - T_n^s)r_s\|_2$$

and furthermore

$$\|r_t - \exp(-T_{K^\infty}t)r_0\| \leq t \sup_{s \leq t} \|(T_{K^\infty} - T_n^s)r_s\|_2$$

Well the conditions on $m$ in the hypothesis suffice to apply Lemma C.10 with $\Gamma = e^2$ ensure that with probability at least $1 - \delta$

$$\sup_{s \leq t} \|(T_{K^\infty} - T_n^s)r_s\|_2 = \tilde{O}\left(S\frac{\sqrt{d}}{\sqrt{m}}[1 + tS] + \frac{S(1+T)\sqrt{\sigma_1 \kappa p}}{\sqrt{n}}\right).$$

Since $\kappa$ and $\sigma_1$ only depend on $K^\infty$ which is fixed we will treat them as constants for simplicity of presentation of the main result (note that they were tracked in all previous results for anyone interested in the specific constants). The desired result follows from plugging in the above expression into the previous bounds after setting $\sigma_1$ and $\kappa$ as constants. $\square$

Theorem 3.5 is strong enough to get a bound on the test error, which is demonstrated by the following corollary.

**Corollary 3.6.** *Assume Assumptions 3.3 and 3.4 hold. Suppose that $f^* = O(1)$ and assume we are performing the doubling trick where $f_0 \equiv 0$ so that $r_0 = -f^*$. Let $k \in \mathbb{N}$ and let $P_k$ be the orthogonal projection onto $span\{\phi_1, \ldots, \phi_k\}$. Set $t = \frac{\log(\sqrt{2}\|P_k f^*\|_2/\epsilon^{1/2})}{\sigma_k}$ Then we have that $m = \tilde{\Omega}(\frac{d}{\epsilon \sigma_k^4})$ and $n = \tilde{\Omega}\left(\frac{p}{\sigma_k^4 \epsilon}\right)$ suffices to ensure with probability at least $1 - \delta$*

$$\frac{1}{2}\|r_t\|_2^2 \leq 2\epsilon + 2\|(I - P_k)f^*\|_2^2.$$

*Proof.* Set $t = \frac{\log(\sqrt{2}\|P_k f^*\|_2/\epsilon^{1/2})}{\sigma_k}$. Note that

$$\frac{1}{2}\|r_t\|_2^2 \leq \frac{1}{2}\left[\|\exp(-T_{K^\infty}t)r_0\|_2 + \|r_t - \exp(-T_{K^\infty}t)r_0\|_2\right]^2$$

$$\leq 2\max\{\|\exp(-T_{K^\infty}t)r_0\|_2, \|r_t - \exp(-T_{K^\infty}t)r_0\|_2\}^2$$

$$\leq 2\left[\|\exp(-T_{K^\infty}t)r_0\|_2^2 + \|r_t - \exp(-T_{K^\infty}t)r_0\|_2^2\right].$$

Note that

$$\|\exp(-T_{K^\infty}t)r_0\|_2^2 = \|\exp(-T_{K^\infty}t)f^*\|_2^2 = \sum_{i=1}^\infty \exp(-2\sigma_i t)|\langle f^*, \phi_i \rangle_2|^2$$

$$\leq \exp(-2\sigma_k t)\sum_{i=1}^k |\langle f^*, \phi_i \rangle_2|^2 + \sum_{i=k+1}^\infty |\langle f^*, \phi_i \rangle_2|^2$$

$$= \frac{\epsilon}{2} + \|(I - P_k)f^*\|_2^2.$$

We want to apply Theorem 3.5 with $T = t$. We need

$$m \geq D^2\|y\|_{\mathbb{R}^n}^2 t^2 \text{ and } m \geq O(\log(c/\delta) + \tilde{O}(d))\max\{t^2, 1\}.$$

Note that since $f^* = O(1)$ we have that $\|\hat{r}(0)\|_{\mathbb{R}^n} = \|y\|_{\mathbb{R}^n} = O(1)$. Then

$$D^2\|y\|_{\mathbb{R}^n}^2 t^2 = \tilde{O}(t^2) = \tilde{O}\left(\frac{1}{\sigma_k^2}\right).$$

thus our condition on $m$ is strong enough to satisfy the first condition. Also $O(\log(c/\delta) + \tilde{O}(d))\max\{t^2, 1\} = \tilde{O}(dt^2)$ which is satisfied by our condition on $m$. Thus by an application of Theorem 3.5 with $T = t$ we have with probability at least $1 - \delta$

$$\|r_t - \exp(-T_{K^\infty}t)r_0\|_2 \leq t\tilde{O}\left(\|f^*\|_\infty[1 + t\|f^*\|_\infty]\frac{\sqrt{d}}{\sqrt{m}} + \|f^*\|_\infty(1+t)\frac{\sqrt{p}}{\sqrt{n}}\right).$$

Recall that $f^* = O(1)$. Thus the first term above is

$$\tilde{O}\left(t^2 \frac{\sqrt{d}}{\sqrt{m}}\right) = \tilde{O}\left(\frac{\sqrt{d}}{\sigma_k^2 \sqrt{m}}\right).$$

Thus setting $m = \tilde{\Omega}(\frac{d}{\epsilon \sigma_k^4})$ suffices to ensure the first term is bounded by $\epsilon^{1/2}/(2\sqrt{2})$. Similarly the second term is

$$\tilde{O}\left(\frac{t^2 \sqrt{p}}{\sqrt{n}}\right) = \tilde{O}\left(\frac{\sqrt{p}}{\sigma_k^2 \sqrt{n}}\right).$$

Thus setting $n = \tilde{\Omega}\left(\frac{p}{\sigma_k^4 \epsilon}\right)$ suffices to ensure that the second term bounded by $\epsilon^{1/2}/(2\sqrt{2})$. Thus in this case we have

$$\|r_t - \exp(-T_{K^\infty}t)r_0\|_2 \le \frac{\epsilon^{1/2}}{\sqrt{2}}.$$

Thus we have

$$\frac{1}{2}\|r_t\|_2^2 \le 2\left[\|\exp(-T_{K^\infty}t)r_0\|_2^2 + \|r_t - \exp(-T_{K^\infty}t)r_0\|_2^2\right] \le 2\epsilon + 2\|(I - P_k)f^*\|_2^2.$$

$\square$

## C.8 DETERMINISTIC INITIALIZATION

In this section we will prove a version of Theorem 3.5 where instead of $\theta_0$ being chosen randomly we take $\theta_0$ to be some deterministic value. $\theta_0$ could represent the parameters given by the output of some pretraining procedure that is independent of the training data, or selected with a priori knowledge.

**Lemma C.11.** *Let $\theta_0$ be a fixed parameter initialization. Let $\Gamma > 1$, $T > 0$, $D := 3\max\{|\sigma(0)|, M\|\sigma'\|_\infty, \|\sigma'\|_\infty, 1\}$,*

$$\xi(t) = \max\{\frac{1}{\sqrt{m}}\|W(t)\|_{op}, \frac{1}{\sqrt{m}}\|b(t)\|_2, \frac{1}{\sqrt{m}}\|a(t)\|_2, 1\},$$

$$\tilde{\xi}(t) = \max\{\max_{\ell \in [m]}\|w_\ell(t)\|_2, \|a(t)\|_\infty, \|b(t)\|_\infty, 1\}.$$

*Furthermore assume*

$$m \ge \frac{D^2 \|\hat{r}(0)\|_{\mathbb{R}^n}^2 T^2}{[\log(\Gamma)]^2}.$$

*Then*

$$\max_{t \in [0,T]} \xi(t) \le \Gamma\xi(0) \quad \max_{t \in [0,T]} \tilde{\xi}(t) \le \Gamma\tilde{\xi}(0).$$

*Proof.* By the hypothesis on $m$ we have that for $t \le T$

$$\frac{D\|\hat{r}(0)\|_{\mathbb{R}^n} t}{\sqrt{m}} \le \log \Gamma.$$

Therefore by Lemmas B.2 and B.3 the desired result holds. $\square$

**Lemma C.12.** *Let $\theta_0$ be a fixed parameter initialization. Let $\Gamma > 1$, $T > 0$, $D := 3\max\{|\sigma(0)|, M\|\sigma'\|_\infty, \|\sigma'\|_\infty, 1\}$,*

$$\xi(t) = \max\{\frac{1}{\sqrt{m}}\|W(t)\|_{op}, \frac{1}{\sqrt{m}}\|b(t)\|_2, \frac{1}{\sqrt{m}}\|a(t)\|_2, 1\},$$

*and assume*

$$m \ge \frac{D^2 \|\hat{r}(0)\|_{\mathbb{R}^n}^2 T^2}{[\log(\Gamma)]^2}.$$

*Then for $t \le T$*

$$\|f_t - f^*\|_\infty \le \|f_0 - f^*\|_\infty + t\|\hat{r}(0)\|_{\mathbb{R}^n} CD^2\Gamma^2\xi(0)^2.$$

*Proof.* Recall that

$$\partial_t(f_t(x) - f^*(x)) = -\frac{1}{n}\sum_{i=1}^{n} K_t(x, x_i)(f_t(x_i) - f^*(x_i)) = -\frac{1}{n}\sum_{i=1}^{n} K_t(x, x_i)\hat{r}(t)_i.$$

Thus

$$|\partial_t(f_t(x) - f^*(x))| \leq \frac{1}{n}\sum_{i=1}^{n} |K_t(x, x_i)||\hat{r}(t)_i| \leq \|K_t\|_\infty \|\hat{r}(t)\|_{\mathbb{R}^n} \leq \|K_t\|_\infty \|\hat{r}(0)\|_{\mathbb{R}^n}.$$

Well by Lemma B.5 we have that $\|K_t\|_\infty \leq CD^2\xi^2(t)$. Also by Lemma C.11 we have that

$$\max_{t\in[0,T]} \xi(t) \leq \Gamma\xi(0).$$

Thus by the fundamental theorem of calculus for $t \leq T$

$$|f_t(x) - f^*(x)| \leq |f_0(x) - f^*(x)| + t\|\hat{r}(0)\|_{\mathbb{R}^n} CD^2\Gamma^2\xi(0)^2.$$

Thus by taking the supremum over $x$ we get

$$\|f_t - f^*\|_\infty \leq \|f_0 - f^*\|_\infty + t\|\hat{r}(0)\|_{\mathbb{R}^n} CD^2\Gamma^2\xi(0)^2$$

which is the desired conclusion. $\square$

**Lemma C.13.** *Let $\theta_0$ be a fixed parameter initialization. Let $K_0$ denote the time-dependent NTK at initialization $\theta_0$. Let $T_{K_0}h(\bullet) = \int_X K_0(\bullet, s)h(s)d\rho(s)$ and $T_n h(\bullet) = \frac{1}{n}\sum_{i=1}^{n} K_0(\bullet, x_i)h(x_i)$ be the associated operators. Let $\kappa = \max_x K_0(x, x)$ and let $\sigma_1$ denote the largest eigenvalue of $T_{K_0}$. Let $\Gamma > 1$, $T > 0$, $D := 3\max\{|\sigma(0)|, M\|\sigma'\|_\infty, \|\sigma'\|_\infty, 1\}$,*

$$\xi(0) = \max\{\frac{1}{\sqrt{m}}\|W(0)\|_{op}, \frac{1}{\sqrt{m}}\|b(0)\|_2, \frac{1}{\sqrt{m}}\|a(0)\|_2, 1\},$$

*and assume*

$$m \geq \frac{D^2\left[\|f^*\|_\infty + \|f_0\|_\infty\right]^2 T^2}{[\log(\Gamma)]^2}.$$

*Then with probability at least $1 - \delta$ over the sampling of $x_1, \ldots, x_n$ we have that*

$$\sup_{t\leq T}\|(T_n - T_{K_0})r_t\|_2 = \tilde{O}\left(\frac{(\|f^*\|_\infty + \|f_0\|_\infty)(1 + T\Gamma^2\xi(0)^2)\sqrt{\sigma_1\kappa p}}{\sqrt{n}}\right).$$

*Proof.* First note that

$$\|\hat{r}(0)\|_{\mathbb{R}^n} \leq \|r_0\|_\infty \leq \|f^*\|_\infty + \|f_0\|_\infty. \tag{10}$$

Thus our hypothesis on $m$ is strong enough to apply Lemma C.11 so that we have

$$\max_{t\in[0,T]} \xi(t) \leq \Gamma\xi(0) =: A. \tag{11}$$

Also by Lemma C.5

$$|b_0(t)| \leq |b_0(0)| + t\|\hat{r}(0)\|_{\mathbb{R}^n},$$

therefore

$$\max_{t\leq T}|b_0(t)| \leq |b_0(0)| + T\|\hat{r}(0)\|_{\mathbb{R}^n} := B.$$

Thus up until time $T$ the neural network is in class $\mathcal{C}$ as defined in Corollary C.2 with parameters $A$ and $B$ as defined above. Furthermore by Lemma C.12 we have

$$\|f_t - f^*\|_\infty \leq \|f_0 - f^*\|_\infty + t\|\hat{r}(0)\|_{\mathbb{R}^n} CD^2\Gamma^2\xi(0)^2.$$

Well then by (10) and the above we have

$$\|r_t\|_\infty \leq (\|f^*\|_\infty + \|f_0\|_\infty)\left\{1 + TCD^2\Gamma^2\xi(0)^2\right\} =: S.$$

Thus by an application of Lemma C.3 with $K = K_0$ we have with probability at least $1 - \delta$ over the sampling of $x_1, \ldots, x_n$ that

$$\sup_{t \leq T} \|(T_n - T_{K_0})r_t\|_2 \leq \frac{2\left[1 + S\sqrt{\sigma_1 \kappa}\sqrt{2\log(c/\delta) + \tilde{O}(p)}\right]}{\sqrt{n}}$$

$$= \tilde{O}\left(\frac{(\|f^*\|_\infty + \|f_0\|_\infty)(1 + T\Gamma^2 \xi(0)^2)\sqrt{\sigma_1 \kappa p}}{\sqrt{n}}\right)$$

where we have used that $S = \tilde{O}([\|f^*\|_\infty + \|f_0\|_\infty][1 + T\Gamma^2 \xi(0)^2])$. □

**Lemma C.14.** *Let* $\theta_0$ *be a fixed parameter initialization. Let* $\Gamma > 1$, $T > 0$, $D := 3\max\{|\sigma(0)|, M\|\sigma'\|_\infty, \|\sigma'\|_\infty, 1\}$, $D' := [\max\{\|\sigma'\|_\infty, \|\sigma''\|_\infty\}^2[M^2 + 1] + D\|\sigma'\|_\infty]\max\{1, M\}$,

$$\xi(t) = \max\{\frac{1}{\sqrt{m}}\|W(t)\|_{op}, \frac{1}{\sqrt{m}}\|b(t)\|_2, \frac{1}{\sqrt{m}}\|a(t)\|_2, 1\},$$

$$\tilde{\xi}(t) = \max\{\max_{\ell \in [m]}\|w_\ell(t)\|_2, \|a(t)\|_\infty, \|b(t)\|_\infty, 1\}.$$

*Furthermore assume*

$$m \geq \frac{D^2 \|\hat{r}(0)\|_{\mathbb{R}^n}^2 T^2}{[\log(\Gamma)]^2}.$$

*Then for* $t \leq T$

$$\|K_0 - K_t\|_\infty \leq t\frac{CDD'}{\sqrt{m}}\Gamma^3 \xi(0)^2\tilde{\xi}(0)\|\hat{r}(0)\|_{\mathbb{R}^n}.$$

*Proof.* Note by Lemma B.24 we have that

$$\sup_{x,y \in B_M \times B_M}|\partial_t K_t(x,y)| \leq \frac{CDD'}{\sqrt{m}}\xi(t)^2\tilde{\xi}(t)\|\hat{r}(t)\|_{\mathbb{R}^n}.$$

Now applying Lemma C.11 and the fact that $\|\hat{r}(t)\|_{\mathbb{R}^n} \leq \|\hat{r}(0)\|_{\mathbb{R}^n}$ from the above we get that for $t \leq T$

$$\sup_{x,y \in B_M \times B_M}|\partial_t K_t(x,y)| \leq \frac{CDD'}{\sqrt{m}}\Gamma^3 \xi(0)^2\tilde{\xi}(0)\|\hat{r}(0)\|_{\mathbb{R}^n}.$$

Thus by the fundamental theorem of calculus we have that for $t \leq T$

$$\|K_0 - K_t\|_\infty \leq t\frac{CDD'}{\sqrt{m}}\Gamma^3 \xi(0)^2\tilde{\xi}(0)\|\hat{r}(0)\|_{\mathbb{R}^n}.$$

□

**Theorem C.15.** *Let* $\theta_0$ *be a fixed parameter initialization. Assume that Assumption 3.3 holds. Let* $\{\phi_i\}_i$ *denote the eigenfunctions of* $T_{K_0}$ *corresponding to the nonzero eigenvalues, which we enumerate* $\sigma_1 \geq \sigma_2 \geq \cdots$. *Let* $P_k$ *be the orthogonal projection in* $L^2$ *onto* $\text{span}\{\phi_1, \ldots, \phi_k\}$ *and let* $D := 3\max\{|\sigma(0)|, M\|\sigma'\|_\infty, \|\sigma'\|_\infty, 1\}$. *Also let* $T > 0$ *and set*

$$\xi(0) = \max\{\frac{1}{\sqrt{m}}\|W(0)\|_{op}, \frac{1}{\sqrt{m}}\|b(0)\|_2, \frac{1}{\sqrt{m}}\|a(0)\|_2, 1\},$$

$$\tilde{\xi}(0) = \max\{\max_{\ell \in [m]}\|w_\ell(0)\|_2, \|a(0)\|_\infty, \|b(0)\|_\infty, 1\}.$$

$$S := \tilde{O}([\|f^*\|_\infty + \|f_0\|_\infty][1 + T\xi(0)^2]).$$

*Assume*

$$m \geq D^2[\|f^*\|_\infty + \|f_0\|_\infty]^2 T^2.$$

*Then with probability at least* $1 - \delta$ *over the sampling of* $x_1, \ldots, x_n$ *we have that for all* $t \leq T$ *and* $k \in \mathbb{N}$

$$\|P_k(r_t - \exp(-T_{K_0}t)r_0)\|_2 \leq \frac{1 - \exp(-\sigma_k t)}{\sigma_k}\tilde{O}\left(\frac{t}{\sqrt{m}}\xi(0)^2\tilde{\xi}(0)\|\hat{r}(0)\|_{\mathbb{R}^n}^2 + \frac{S\sqrt{\sigma_1 \kappa p}}{\sqrt{n}}\right)$$

*and*

$$\|r_t - \exp(-T_{K_0}t)r_0\|_2 \leq t\tilde{O}\left(\frac{t}{\sqrt{m}}\xi(0)^2\tilde{\xi}(0)\|\hat{r}(0)\|_{\mathbb{R}^n}^2 + \frac{S\sqrt{\sigma_1 \kappa p}}{\sqrt{n}}\right).$$

*Proof.* By Lemma C.8 we have for any $k \in \mathbb{N}$

$$\|P_k(r_t - \exp(-T_{K_0}t)r_0)\|_2 \leq \frac{1 - \exp(-\sigma_k t)}{\sigma_k} \sup_{s \leq t} \|(T_{K_0} - T_n^s)r_s\|_2$$

and furthermore

$$\|r_t - \exp(-T_{K_0}t)r_0\| \leq t \sup_{s \leq t} \|(T_{K_0} - T_n^s)r_s\|_2 .$$

Let $T_n h(\bullet) = \frac{1}{n} \sum_{i=1}^n K_0(\bullet, x_i)h(x_i)$ be the discretization of $T_{K_0}$. Thus by Lemma C.9 we have

$$\sup_{s \leq t} \|(T_{K_0} - T_n^s)r_s\|_2 \leq \sup_{s \leq t} \|(T_{K_0} - T_n)r_s\|_2 + \sup_{s \leq t} \|K_0 - K_s\|_\infty \|\hat{r}(0)\|_{\mathbb{R}^n} .$$

Note from the inequality

$$\|\hat{r}(0)\|_{\mathbb{R}^n} \leq \|r_0\|_\infty \leq \|f^*\|_\infty + \|f_0\|_\infty$$

the hypothesis on $m$ is strong enough to apply Lemma C.14 with $\Gamma = e$. Well then by an application of Lemma C.14 with $\Gamma = e$ we have that

$$\sup_{s \leq t} \|K_s - K_0\|_\infty = \tilde{O}\left(\frac{t}{\sqrt{m}} \xi(0)^2 \tilde{\xi}(0) \|\hat{r}(0)\|_{\mathbb{R}^n}\right).$$

Separately by Lemma C.13 we have with probability at least $1 - \delta$

$$\sup_{s \leq t} \|(T_{K_0} - T_n)r_s\|_2 = \tilde{O}\left(\frac{(\|f^*\|_\infty + \|f_0\|_\infty)(1 + T\xi(0)^2)\sqrt{\sigma_1 \kappa p}}{\sqrt{n}}\right).$$

Combining these results we get that

$$\sup_{s \leq t} \|(T_{K_0} - T_n^s)r_s\|_2 \leq \tilde{O}\left(\frac{t}{\sqrt{m}} \xi(0)^2 \tilde{\xi}(0) \|\hat{r}(0)\|_{\mathbb{R}^n}^2 + \frac{S\sqrt{\sigma_1 \kappa p}}{\sqrt{n}}\right).$$

The desired result follows from plugging in the above expression into the previous bounds. □

# D   DAMPED DEVIATIONS ON TRAINING SET

The damped deviations lemma for the training set is incredibly simple to prove and yet is incredibly powerful as we will see later. Here is the proof.

**Lemma 2.1.** *Let $G \in \mathbb{R}^{n \times n}$ be an arbitrary positive semidefinite matrix and let $G_s$ be the time dependent NTK matrix at time $s$. Then*

$$\hat{r}_t = \exp(-Gt)\hat{r}_0 + \int_0^t \exp(-G(t - s))(G - G_s)\hat{r}_s ds.$$

*Proof.* Note that we have the equation

$$\partial_t \hat{r}_t = -G_t \hat{r}_t = -G\hat{r}_t + (G - G_t)\hat{r}_t.$$

Thus by multiplying by the integrating factor $\exp(Gt)$ and using the fact that $\exp(Gt)$ and $G$ commute we have that

$$\partial_t \exp(Gt)\hat{r}_t = \exp(Gt)(G - G_t)\hat{r}_t.$$

Therefore by the fundamental theorem of calculus

$$\exp(Gt)\hat{r}_t - \hat{r}_0 = \int_0^t \exp(Gs)(G - G_s)\hat{r}_s ds,$$

which after rearrangement gives

$$\hat{r}_t = \exp(-Gt)\hat{r}_0 + \int_0^t \exp(-G(t - s))(G - G_s)\hat{r}_s ds.$$

□

Throughout we will let $u_1, \ldots, u_n$ denote the eigenvectors of $G^\infty$ with corresponding eigenvalues $\lambda_1, \ldots, \lambda_n$, normalized to have unit norm in $\|\bullet\|_{\mathbb{R}^n}$, i.e. $\|u_i\|_{\mathbb{R}^n} = 1$. The following corollary demonstrates that if one is only interested in approximating the top eigenvectors, then the deviations of the $NTK$ only need to be small relative to the cutoff eigenvalue $\lambda_i$ that you care about.

**Corollary D.1.** *Let $P_k$ be the orthogonal projection onto $span\{u_1, \ldots, u_k\}$. Then for any $k \in [n]$*

$$\|P_k(\hat{r}_t - \exp(-G^\infty t)\hat{r}_0)\|_{\mathbb{R}^n} \leq \sup_{s \leq t} \|G^\infty - G_s\|_{op} \|\hat{r}_0\|_{\mathbb{R}^n} \frac{1 - \exp(-\lambda_k t)}{\lambda_k}$$

$$\leq \sup_{s \leq t} \|G^\infty - G_s\|_{op} \|\hat{r}_0\|_{\mathbb{R}^n} t.$$

*In particular*

$$\|\hat{r}_t - \exp(-G^\infty t)\hat{r}_0\|_{\mathbb{R}^n} \leq \sup_{s \leq t} \|G^\infty - G_s\|_{\mathbb{R}^n} \|\hat{r}_0\|_{\mathbb{R}^n} \frac{1 - \exp(-\lambda_n t)}{\lambda_n} \leq \sup_{s \leq t} \|G^\infty - G_s\|_{op} \|\hat{r}_0\|_{\mathbb{R}^n} t.$$

*Proof.* Note by Lemma 2.1 we have that

$$\hat{r}_t - \exp(-G^\infty t)\hat{r}_0 = \int_0^t \exp(-G^\infty(t-s))(G^\infty - G_s)\hat{r}_s ds$$

Therefore for any $k \in [n]$

$$P_k(\hat{r}_t - \exp(-G^\infty t)\hat{r}_0) = P_k \int_0^t \exp(-G^\infty(t-s))(G^\infty - G_s)\hat{r}_s ds$$

$$= \int_0^t P_k \exp(-G^\infty(t-s))(G^\infty - G_s)\hat{r}_s ds$$

Thus

$$\|P_k(\hat{r}_t - \exp(-G^\infty t)\hat{r}_0)\|_{\mathbb{R}^n} = \left\|\int_0^t P_k \exp(-G^\infty(t-s))(G^\infty - G_s)\hat{r}_s ds\right\|_{\mathbb{R}^n}$$

$$\leq \int_0^t \|P_k \exp(-G^\infty(t-s))(G^\infty - G_s)\hat{r}_s\|_{\mathbb{R}^n} ds$$

$$\leq \int_0^t \|P_k \exp(-G^\infty(t-s))\|_{op} \|G^\infty - G_s\|_{op} \|\hat{r}_s\|_{\mathbb{R}^n} ds$$

$$\leq \int_0^t \exp(-\lambda_k(t-s)) \|G^\infty - G_s\|_{op} \|\hat{r}_0\|_{\mathbb{R}^n} ds$$

$$\leq \sup_{s \leq t} \|G^\infty - G_s\|_{op} \|\hat{r}_0\|_{\mathbb{R}^n} \int_0^t \exp(-\lambda_k(t-s)) ds$$

$$\leq \sup_{s \leq t} \|G^\infty - G_s\|_{op} \|\hat{r}_0\|_{\mathbb{R}^n} \frac{1 - \exp(-\lambda_k t)}{\lambda_k}$$

$$\leq \sup_{s \leq t} \|G^\infty - G_s\|_{op} \|\hat{r}_0\|_{\mathbb{R}^n} t$$

where we have used the inequality $1 + x \leq \exp(x)$ in the last inequality. By specializing to the case $k = n$ since $span\{u_1, \ldots, u_n\} = \mathbb{R}^n$ we have

$$\|\hat{r}_t - \exp(-G^\infty t)\hat{r}_0\|_{\mathbb{R}^n} \leq \sup_{s \leq t} \|G^\infty - G_s\|_{op} \|\hat{r}_0\|_{\mathbb{R}^n} \frac{1 - \exp(-\lambda_n t)}{\lambda_n}$$

$$\leq \sup_{s \leq t} \|G^\infty - G_s\|_{op} \|\hat{r}_0\|_{\mathbb{R}^n} t.$$

This completes the proof. $\square$

Theorem 3.5 uses the concept of damped deviations to compare $r_t$ with $\exp(-T_{K^\infty}t)r_0$. We can also prove the analogous statement on the training set that compares $\hat{r}_t$ to $\exp(-G^\infty t)\hat{r}_0$. The following is the analog of Theorem 3.5 on the training set.

**Theorem D.2.** *Let* $D := 3\max\{|\sigma(0)|, M\|\sigma'\|_\infty, \|\sigma'\|_\infty, 1\}$. *Also let* $\Gamma > 1, T > 0$. *Furthermore assume*

$$m \geq \frac{4D^2\|y\|_{\mathbb{R}^n}^2 T^2}{[\log(\Gamma)]^2} \text{ and } m \geq \max\left\{\frac{4D^2 O(\log(c/\delta) + \tilde{O}(d))T^2}{[\log(\Gamma)]^2}, O(\log(c/\delta) + \tilde{O}(d))\right\}$$

*Let* $P_k$ *be the orthogonal projection onto* $\text{span}\{u_1, \ldots, u_k\}$. *Then with probability at least* $1 - \delta$ *we have for any* $k \in [n]$ *and* $t \leq T$

$$\|P_k(\hat{r}_t - \exp(-G^\infty t)\hat{r}_0)\|_{\mathbb{R}^n} \leq \frac{1 - \exp(-\lambda_k t)}{\lambda_k}\|\hat{r}_0\|_{\mathbb{R}^n}\tilde{O}\left(\frac{\sqrt{d}}{\sqrt{m}}\left[1 + t\Gamma^3\|\hat{r}(0)\|_{\mathbb{R}^n}\right]\right)$$

$$\leq t\|\hat{r}_0\|_{\mathbb{R}^n}\tilde{O}\left(\frac{\sqrt{d}}{\sqrt{m}}\left[1 + t\Gamma^3\|\hat{r}(0)\|_{\mathbb{R}^n}\right]\right)$$

*in particular*

$$\|\hat{r}_t - \exp(-G^\infty t)\hat{r}_0\|_{\mathbb{R}^n} \leq \frac{1 - \exp(-\lambda_n t)}{\lambda_n}\|\hat{r}_0\|_{\mathbb{R}^n}\tilde{O}\left(\frac{\sqrt{d}}{\sqrt{m}}\left[1 + t\Gamma^3\|\hat{r}(0)\|_{\mathbb{R}^n}\right]\right)$$

$$\leq t\|\hat{r}_0\|_{\mathbb{R}^n}\tilde{O}\left(\frac{\sqrt{d}}{\sqrt{m}}\left[1 + t\Gamma^3\|\hat{r}(0)\|_{\mathbb{R}^n}\right]\right).$$

*If one instead does the doubling trick the condition* $\frac{4D^2 O(\log(c/\delta) + \tilde{O}(d))T^2}{[\log(\Gamma)]^2}$ *can be removed from the hypothesis and the same conclusion holds.*

*Proof.* By Corollary D.1 we have

$$\|P_k(\hat{r}_t - \exp(-G^\infty t)\hat{r}_0)\|_{\mathbb{R}^n} \leq \sup_{s \leq t}\|G^\infty - G_s\|_{\mathbb{R}^n}\|\hat{r}_0\|_{\mathbb{R}^n}\frac{1 - \exp(-\lambda_k t)}{\lambda_k}$$

$$\leq \sup_{s \leq t}\|G^\infty - G_s\|_{\mathbb{R}^n}\|\hat{r}_0\|_{\mathbb{R}^n}t$$

Well by Theorem B.27 we have with probability at least $1 - \delta$

$$\sup_{s \leq t}\|G^\infty - G_t\|_{op} \leq \tilde{O}\left(\frac{\sqrt{d}}{\sqrt{m}}\left[1 + t\Gamma^3\|\hat{r}(0)\|_{\mathbb{R}^n}\right]\right)$$

The desired result follows from plugging this in to the previous bounds. □

# E    PROOF OF THEOREM 3.7

We can now quickly prove our analog of Theorem 4.1 from Arora et al. (2019).

**Theorem 3.7.** *Assume* $m = \tilde{\Omega}(dn^5\epsilon^{-2}\lambda_n(H^\infty)^{-4})$ *and* $m \geq O(\log(c/\delta) + \tilde{O}(d))$ *and* $f^* = O(1)$. *Assume we are performing the doubling trick so that* $\hat{r}_0 = -y$. *Let* $v_1, \ldots, v_n$ *denote the eigenvectors of* $G^\infty$ *normalized to have unit L2 norm* $\|v_i\|_2 = 1$. *Then with probability at least* $1 - \delta$

$$\hat{r}_t = \exp(-G^\infty t)(-y) + \delta(t),$$

*where* $\sup_{t \geq 0}\|\delta(t)\|_2 \leq \epsilon$. *In particular*

$$\|\hat{r}_t\|_2 = \sqrt{\sum_{i=1}^n \exp(-2\lambda_i t)|\langle y, v_i\rangle_2|^2} \pm \epsilon.$$

*Proof.* Set $T = \log(\|\hat{r}(0)\|_{\mathbb{R}^n}\sqrt{n}/\epsilon)/\lambda_n$. Note that since $f^* = O(1)$ and we are performing the doubling trick we have that $\|\hat{r}_0\|_{\mathbb{R}^n} = \|y\|_{\mathbb{R}^n} = O(1)$. Recall that $\lambda_n := \frac{1}{n}\lambda_n(H^\infty)$ therefore $m = \tilde{\Omega}(dn^5\epsilon^{-2}\lambda_n(H^\infty)^{-4}) = \tilde{\Omega}(dn\epsilon^{-2}\lambda_n^{-4})$ is strong enough to ensure that

$$m \geq \frac{4D^2\|y\|_{\mathbb{R}^n}^2 T^2}{[\log(2)]^2} = \tilde{O}(\lambda_n^{-2}) \quad m \geq O(\log(c/\delta) + \tilde{O}(d)) = \tilde{O}(d)$$

Then by an application of Theorem D.2 with $\Gamma = 2$ we have with probability at least $1 - \delta$ that for all $t \leq T$

$$\|\hat{r}_t - \exp(-G^\infty t)\hat{r}_0\|_{\mathbb{R}^n} \leq \frac{1 - \exp(-\lambda_n t)}{\lambda_n} \|\hat{r}_0\|_{\mathbb{R}^n} \tilde{O}\left(\frac{\sqrt{d}}{\sqrt{m}}\left[1 + t\Gamma^3 \|\hat{r}(0)\|_{\mathbb{R}^n}\right]\right)$$

$$\leq \frac{\|\hat{r}_0\|_{\mathbb{R}^n}}{\lambda_n} \tilde{O}\left(\frac{\sqrt{d}}{\sqrt{m}}\left[1 + T\Gamma^3 \|\hat{r}(0)\|_{\mathbb{R}^n}\right]\right)$$

Since $f^* = O(1)$ we have that $\|\hat{r}_0\|_{\mathbb{R}^n} = \|y\|_{\mathbb{R}^n} = O(1)$ therefore the above bound is

$$\tilde{O}\left(\frac{\sqrt{d}}{\sqrt{m}} \frac{T}{\lambda_n}\right) = \tilde{O}\left(\frac{\sqrt{d}}{\sqrt{m}} \frac{1}{\lambda_n^2}\right)$$

Thus $m = \tilde{\Omega}(dn^5\epsilon^{-2}\lambda_n(H^\infty)^{-4}) = \tilde{\Omega}(dn\epsilon^{-2}\lambda_n^{-4})$ suffices to make the above term bounded by $\epsilon/\sqrt{n}$. Thus in this case

$$\sup_{t \leq T} \|\hat{r}_t - \exp(-G^\infty t)\hat{r}_0\|_{\mathbb{R}^n} \leq \epsilon/\sqrt{n}.$$

Let $\delta(t) = \hat{r}_t - \exp(-G^\infty t)\hat{r}_0$. We have just shown that $\sup_{t \leq T} \|\delta(t)\|_{\mathbb{R}^n} \leq \epsilon/\sqrt{n}$. We will now bound $\delta(t)$ for $t \geq T$. Note that for $t \geq T$

$$\|\exp(-G^\infty t)\hat{r}_0\|_{\mathbb{R}^n} \leq \exp(-\lambda_n t) \|\hat{r}_0\|_{\mathbb{R}^n} \leq \exp(-\lambda_n T) \|\hat{r}_0\|_{\mathbb{R}^n} \leq \epsilon/\sqrt{n}$$

Also for $t \geq T$

$$\|\hat{r}_t\|_{\mathbb{R}^n} \leq \|\hat{r}_T\|_{\mathbb{R}^n} \leq \|\exp(-G^\infty T)\hat{r}_0\|_{\mathbb{R}^n} + \|\delta(T)\|_{\mathbb{R}^n} \leq 2\epsilon/\sqrt{n}$$

where we have used that $\|\hat{r}_t\|_{\mathbb{R}^n}$ is nonincreasing for gradient flow. Therefore for $t \geq T$

$$\|\delta(t)\|_{\mathbb{R}^n} \leq \|\hat{r}_t\|_{\mathbb{R}^n} + \|\exp(-G^\infty t)\hat{r}_0\|_{\mathbb{R}^n} \leq 3\epsilon/\sqrt{n}$$

Thus we have shown

$$\sup_{t \geq 0} \|\delta(t)\|_{\mathbb{R}^n} \leq 3\epsilon/\sqrt{n}.$$

The desired result follows from replacing $\epsilon$ with $\epsilon/3$ in the previous argument and using the fact that $\|\bullet\|_2 = \sqrt{n} \|\bullet\|_{\mathbb{R}^n}$ and $\hat{r}_0 = -y$. $\qquad\square$

## F    PROOF OF THEOREM 3.8

Using some lemmas that we leave to the following section, we can prove Theorem 3.8 quite quickly using the damped deviations equation and the NTK deviation bounds.

### F.1    MAIN THEOREM

**Theorem 3.8.** *Assume Assumptions 3.3 and 3.4 hold. Furthermore assume $m = \tilde{\Omega}\left(\epsilon^{-2}dT^2 \|f^*\|_\infty^2 (1 + T \|f^*\|_\infty)^2\right)$ where $T > 0$ is a time parameter and $m \geq O(\log(c/\delta) + \tilde{O}(d))$ and $n \geq \frac{128\kappa^2 \log(2/\delta)}{(\sigma_k - \sigma_{k+1})^2}$. Also assume $f^* \in L^\infty(X) \subset L^2(X)$ and let $P^{T_{K^\infty}}$ be the orthogonal projection onto the eigenspaces of $T_{K^\infty}$ corresponding to the eigenvalue $\alpha \in \sigma(T_{K^\infty})$ and higher. Assume that $\left\|(I - P^{T_{K^\infty}})f^*\right\|_\infty \leq \epsilon'$ for some $\epsilon' \geq 0$. Pick $k$ so that $\sigma_k = \alpha$ and $\sigma_{k+1} < \alpha$, i.e. $k$ is the index of the last repeated eigenvalue corresponding to $\alpha$ in the ordered sequence $\{\sigma_i\}_i$. Also assume we are performing the doubling trick so that $\hat{r}(0) = -y$. Then we have with probability at least $1 - 3\delta$ over the sampling of $x_1, \ldots, x_n$ and $\theta_0$ that for $t \leq T$*

$$\|\hat{r}_t\|_{\mathbb{R}^n} \leq \exp(-\lambda_k t) \|y\|_{\mathbb{R}^n} + \frac{4\kappa \|f^*\|_2 \sqrt{10\log(2/\delta)}}{(\sigma_k - \sigma_{k+1})\sqrt{n}} + 2\epsilon' + \epsilon.$$

*Proof.* Recall that we have $\|\hat{r}(0)\|_{\mathbb{R}^n} = \|-y\|_{\mathbb{R}^n} \leq \|f^*\|_\infty$. Note that $m = \tilde{\Omega}\left(\epsilon^{-2}dT^2 \|f^*\|_\infty^2 (1 + T \|f^*\|_\infty)^2\right)$ and $m \geq O(\log(c/\delta) + \tilde{O}(d))$ are strong enough to ensure

the hypothesis of Theorem D.2 is satisfied with $\Gamma = 2$. From now on $\Gamma = 2 = O(1)$ and will be treated as a constant. Then by Theorem D.2 with probability at least $1 - \delta$ we have for $t \leq T$

$$\|\hat{r}_t - \exp(-G^\infty t)\hat{r}_0\|_{\mathbb{R}^n} \leq T \|\hat{r}_0\|_{\mathbb{R}^n} \tilde{O}\left(\frac{\sqrt{d}}{\sqrt{m}}\left[1 + T\Gamma^3 \|\hat{r}(0)\|_{\mathbb{R}^n}\right]\right)$$

Thus using the fact from the doubling trick that $\|\hat{r}(0)\|_{\mathbb{R}^n} = \|y\|_{\mathbb{R}^n} \leq \|f^*\|_\infty$ setting $m = \tilde{\Omega}\left(\epsilon^{-2}dT^2 \|f^*\|_\infty^2 (1 + T \|f^*\|_\infty)^2\right)$ suffices to ensure that $\|\hat{r}_t - \exp(-G^\infty t)\hat{r}_0\|_{\mathbb{R}^n} \leq \epsilon$ for $t \leq T$. Let $P_k$ be the orthogonal projection onto $\mathrm{span}\{u_1, \ldots, u_k\}$. Well then for $t \leq T$

$$\|\hat{r}_t\|_{\mathbb{R}^n} \leq \|\exp(-G^\infty t)\hat{r}_0\|_{\mathbb{R}^n} + \epsilon \leq \|P_k \exp(-G^\infty t)\hat{r}_0\|_{\mathbb{R}^n} + \|(I - P_k)\exp(-G^\infty t)\hat{r}_0\|_{\mathbb{R}^n} + \epsilon$$
$$\leq \exp(-\lambda_k t)\|\hat{r}_0\|_{\mathbb{R}^n} + \|(I - P_k)\hat{r}_0\|_{\mathbb{R}^n} + \epsilon$$

By Theorem F.6 we have with probability at least $1 - 2\delta$ over the sampling of $x_1, \ldots, x_n$ that

$$\|(I - P_k)y\|_{\mathbb{R}^n} \leq 2\epsilon' + \frac{4\kappa \|f^*\|_2 \sqrt{10\log(2/\delta)}}{(\sigma_k - \sigma_{k+1})\sqrt{n}}.$$

Since we are using the doubling trick we have $\hat{r}_0 = -y$. Thus we have

$$\|(I - P_k)\hat{r}_0\|_{\mathbb{R}^n} \leq 2\epsilon' + \frac{4\kappa \|f^*\|_2 \sqrt{10\log(2/\delta)}}{(\sigma_k - \sigma_{k+1})\sqrt{n}}.$$

Thus by taking a union bound we have with probability at least $1 - 3\delta$ for all $t \leq T$

$$\|\hat{r}_t\|_{\mathbb{R}^n} \leq \exp(-\lambda_k t)\|\hat{r}_0\|_{\mathbb{R}^n} + \frac{4\kappa \|f^*\|_2 \sqrt{10\log(2/\delta)}}{(\sigma_k - \sigma_{k+1})\sqrt{n}} + +2\epsilon' + \epsilon.$$

The desired result follows from $\hat{r}_0 = -y$. □

### F.2 CONTROL OF INITIAL RESIDUAL

We will use some of the notation and operator theory from Section C.1 and Section C.2 in this section, thus it is recommended to have read those sections first. Let $u_1, \ldots, u_n$ denote the eigenvectors of $G^\infty$ normalized to have unit norm in $\|\bullet\|_{\mathbb{R}^n}$, i.e. $\|u_i\|_{\mathbb{R}^n} = 1$. Let $P_k$ be the orthogonal projection onto $\mathrm{span}\{u_1, \ldots, u_k\}$. The goal of this section is to upper bound the extent to which the labels $y$ participate in the bottom eigendirections of $G^\infty$, i.e. to show that $\|(I - P_k)y\|_{\mathbb{R}^n}$ is small. Let $P^{T_{K^\infty}}$ be some projection onto the top eigenspaces of $T_{K^\infty}$. The idea is to show that if $\|(I - P^{T_{K^\infty}})f^*\|_2$ is small then by picking $P_k$ so that $\mathrm{rank}(P_k) = \mathrm{rank}(P^{T_{K^\infty}})$ then $\|(I - P_k)y\|_{\mathbb{R}^n}$ is also small with high probability. The results in this section essentially all appear in the proofs in Su & Yang (2019). We repeat the arguments here for completeness and due to differences in notation and constants.

We use some of the same machinery in (Rosasco et al., 2010). We define operators $L_{\mathcal{H}} : \mathcal{H} \to \mathcal{H}$ and $T_n : \mathcal{H} \to \mathcal{H}$ by

$$T_{\mathcal{H}}f := \int_X \langle f, K_s \rangle_{\mathcal{H}} K_s d\rho(s)$$

$$T_n f := \frac{1}{n}\sum_{i=1}^n \langle f, K_{x_i} \rangle_{\mathcal{H}} K_{x_i}$$

Note that $T_{\mathcal{H}}$ is equal to $T_{K^\infty}$ on $\mathcal{H}$ and $T_n$ is simply the operator you get if you replace $\rho$ in the defintion of $T_{\mathcal{H}}$ with the empirical measure $\frac{1}{n}\sum_{i=1}^n \delta_{x_i}$. We define the "restriction" operator $R_n : \mathcal{H} \to \mathbb{R}^n$ by

$$R_n f = [f(x_1), f(x_2), \ldots, f(x_n)]^T$$

Note here the domain of $R_n$ is $\mathcal{H}$ but in other parts of this paper we will allow $R_n$ to take more general functions as input. Define $R_n^* : \mathbb{R}^n \to \mathcal{H}$ by

$$R_n^*(v_1, \ldots, v_n) = \frac{1}{n}\sum_{i=1}^n v_i K_{x_i}.$$

It can be seen that

$$\langle R_n^* v, f \rangle_{\mathcal{H}} = \langle v, R_n f \rangle_{\mathbb{R}^n}.$$

and thus $R_n^*$ is the adjoint of $R_n$. Using these operators we may write $T_n = R_n^* R_n$ and $G^\infty = R_n R_n^*$. It will follow that $T_n$ and $G^\infty$ have the same eigenvalues (up to some zero eigenvalues) and their eigenvectors are related. We recall the following result from (Rosasco et al., 2010) (Proposition 9)

**Theorem F.1.** *(Rosasco et al., 2010) The following hold*

- *The operator $T_n$ is finite rank, self-adjoint and positive, and the matrix $G^\infty$ is symmetric and semi-positive definite. In particular the spectrum $\sigma(T_n)$ of $T_n$ has finitely many non-zero elements and they are contained in $[0, \kappa]$.*

- *The spectrum of $T_n$ and $G^\infty$ are the same up to zero, specifically $\sigma(G^\infty) \setminus \{0\} = \sigma(T_n) \setminus \{0\}$. Moreover if $\lambda_i$ is a nonzero eigenvalue and $u_i$ and $v_i$ are the corresponding eigenvector and eigenfunction for $G^\infty$ and $T_n$ respectively (normalized to norm 1 in $\|\bullet\|_{\mathbb{R}^n}$ and $\|\bullet\|_{\mathcal{H}}$ respectively), then*

$$u_i = \frac{1}{\lambda_i^{1/2}} R_n v_i$$

$$v_i = \frac{1}{\lambda_i^{1/2}} R_n^* u_i = \frac{1}{\lambda_i^{1/2}} \frac{1}{n} \sum_{j=1}^n K_{x_j}(u_i)_j$$

*where $(u_i)_j$ is the jth component of the vector $u_i$.*

- *The following decompositions hold*

$$G^\infty w = \sum_{j=1}^k \lambda_j \langle w, u_j \rangle_{\mathbb{R}^n} u_j$$

$$T_n f = \sum_{j=1}^k \lambda_j \langle f, v_j \rangle_{\mathcal{H}} v_j$$

*where $k = rank(G^\infty) = rank(T_n)$ and both sums run over the positive eigenvalues. $\{u_i\}_{i=1}^k$ is an orthonormal basis for $ker(G^\infty)^\perp$ and $\{v_i\}_{i=1}^k$ is an orthonormal basis for $ker(T_n)^\perp$.*

We will make use of the following lemma from Rosasco et al. (2010, Proposition 6):

**Lemma F.2.** *(Rosasco et al., 2010) Let $\alpha_1 > \alpha_2 > \ldots > \alpha_N > \alpha_{N+1}$ be the top $N + 1$ distinct eigenvalues of $T_{\mathcal{H}}$. Let $P^{T_{\mathcal{H}}}$ be the orthogonal projection onto the eigenfunctions of $T_{\mathcal{H}}$ corresponding to eigenvalues $\alpha_N$ and above. Let $P^{T_n}$ be the projection onto the top $k$ eigenvectors of $T_n$ so that $k = dim(range(T_n)) = dim(range(T_{\mathcal{H}}))$. Assume further that*

$$\|T_{\mathcal{H}} - T_n\|_{HS} \leq \frac{\alpha_N - \alpha_{N+1}}{4}.$$

*Then*

$$\left\| P^{T_{\mathcal{H}}} - P^{T_n} \right\|_{HS} \leq \frac{2}{\alpha_N - \alpha_{N+1}} \left\| T_{\mathcal{H}} - T_n \right\|_{HS}.$$

The following lemma will be useful.

**Lemma F.3.** *Let $f^* \in L^2$ and let $P^{T_{\mathcal{H}}}$ and $P^{T_n}$ be defined as in Lemma F.2. Then*

$$\sum_{i=k+1}^n |\langle R_n P^{T_{K^\infty}} f^*), u_i \rangle_{\mathbb{R}^n}|^2 \leq \frac{\|f^*\|_2^2 \lambda_{k+1}}{\sigma_k} \left\| P^{T_{\mathcal{H}}} - P^{T_n} \right\|_{HS}^2$$

*Proof.* We repeat the same proof as in (Su & Yang, 2019) for completeness and to remove confusion that may arise from differences in notation. The proof was originally given in (Rosasco et al., 2010) albeit with a minor error involving missing multiplicative factors. Note that

$$P^{T_{K^\infty}} f^* = \sum_{j=1}^k \langle f^*, \phi_j \rangle_2 \phi_j$$

Therefore

$$\langle R_n P^{T_{K^\infty}} f^*, u_i \rangle_{\mathbb{R}^n} = \sum_{j=1}^{k} \langle f^*, \phi_j \rangle_2 \langle R_n \phi_j, u_i \rangle_{\mathbb{R}^n}.$$

Applying Cauchy-Schwarz we get

$$|\langle R_n P^{T_{K^\infty}} f^*, u_i \rangle_{\mathbb{R}^n}|^2 \leq \left[ \sum_{j=1}^{k} |\langle f^*, \phi_j \rangle_2|^2 \right] \left[ \sum_{j=1}^{k} |\langle R_n \phi_j, u_i \rangle_{\mathbb{R}^n}|^2 \right]$$

$$\leq \|f^*\|_2^2 \sum_{j=1}^{k} |\langle R_n \phi_j, u_i \rangle_{\mathbb{R}^n}|^2.$$

Well then note that

$$\sum_{j=1}^{k} |\langle R_n \phi_j, u_i \rangle_{\mathbb{R}^n}|^2 = \sum_{j=1}^{k} |\langle \phi_j, R_n^* u_i \rangle_{\mathcal{H}}|^2 = \sum_{j=1}^{k} \lambda_i |\langle \phi_j, v_i \rangle_{\mathcal{H}}|^2.$$

Therefore

$$\sum_{i=k+1}^{n} |\langle R_n P^{T_{K^\infty}} f^*), u_i \rangle_{\mathbb{R}^n}|^2 \leq \|f^*\|_2^2 \sum_{i=k+1}^{n} \sum_{j=1}^{k} \lambda_i |\langle \phi_j, v_i \rangle_{\mathcal{H}}|^2$$

$$\leq \|f^*\|_2^2 \lambda_{k+1} \sum_{i=k+1}^{n} \sum_{j=1}^{k} |\langle \phi_j, v_i \rangle_{\mathcal{H}}|^2 \tag{12}$$

On the other hand

$$\left\| P^{T_{\mathcal{H}}} - P^{T_n} \right\|_{HS}^2 \geq \sum_{j=1}^{k} \left\| (P^{T_{\mathcal{H}}} - P^{T_n}) \sqrt{\sigma_j} \phi_j \right\|_{\mathcal{H}}^2$$

$$\geq \sum_{j=1}^{k} \sum_{i=k+1}^{n} |\langle (P^{T_{\mathcal{H}}} - P^{T_n}) \sqrt{\sigma_j} \phi_j, v_i \rangle_{\mathcal{H}}|^2.$$

Note that for $1 \leq j \leq k$ and $k+1 \leq i \leq n$ we have

$$\langle (P^{T_{\mathcal{H}}} - P^{T_n}) \sqrt{\sigma_j} \phi_j, v_i \rangle_{\mathcal{H}} = \langle P^{T_{\mathcal{H}}} \sqrt{\sigma_j} \phi_j, v_i \rangle_{\mathcal{H}} - \langle P^{T_n} \sqrt{\sigma_j} \phi_j, v_i \rangle_{\mathcal{H}}$$

$$= \langle \sqrt{\sigma_j} \phi_j, v_i \rangle_{\mathcal{H}} - \langle \sqrt{\sigma_j} \phi_j, P^{T_n} v_i \rangle_{\mathcal{H}} = \langle \sqrt{\sigma_j} \phi_j, v_i \rangle_{\mathcal{H}}.$$

So

$$\sum_{j=1}^{k} \sum_{i=k+1}^{n} |\langle (P^{T_{\mathcal{H}}} - P^{T_n}) \sqrt{\sigma_j} \phi_j, v_i \rangle_{\mathcal{H}}|^2 = \sum_{j=1}^{k} \sum_{i=k+1}^{n} |\langle \sqrt{\sigma_j} \phi_j, v_i \rangle_{\mathcal{H}}|^2$$

$$\geq \sigma_k \sum_{j=1}^{k} \sum_{i=k+1}^{n} |\langle \phi_j, v_i \rangle_{\mathcal{H}}|^2$$

To summarize we have shown

$$\frac{1}{\sigma_k} \left\| P^{T_{\mathcal{H}}} - P^{T_n} \right\|_{HS}^2 \geq \sum_{j=1}^{k} \sum_{i=k+1}^{n} |\langle \phi_j, v_i \rangle_{\mathcal{H}}|^2.$$

Combining this with (12) we get the final result

$$\sum_{i=k+1}^{n} |\langle R_n P^{T_{K^\infty}} f^*), u_i \rangle_{\mathbb{R}^n}|^2 \leq \frac{\|f^*\|_2^2 \lambda_{k+1}}{\sigma_k} \left\| P^{T_{\mathcal{H}}} - P^{T_n} \right\|_{HS}^2.$$

$$\square$$

We can use Lemma F.3 to produce the following bound.

**Lemma F.4.** *Let $f^* \in L^2$ and let $P^{T_\mathcal{H}}$ and $P^{T_n}$ be defined as in Lemma F.2. Then*

$$\sum_{i=k+1}^{n} |\langle R_n f^*, u_i \rangle_{\mathbb{R}^n}|^2 \leq \frac{2}{n} \sum_{i=1}^{n} |(I - P^{T_{K^\infty}}) f^*(x_i)|^2 + 2 \frac{\|f^*\|_2^2 \lambda_{k+1}}{\sigma_k} \left\| P^{T_\mathcal{H}} - P^{T_n} \right\|_{HS}^2.$$

*Proof.* We have that

$$\langle R_n f^*, u_i \rangle_{\mathbb{R}^n} = \langle R_n (I - P^{T_{K^\infty}}) f^*, u_i \rangle_{\mathbb{R}^n} + \langle R_n P^{T_{K^\infty}} f^*, u_i \rangle_{\mathbb{R}^n}.$$

Thus from the inequality $(a+b)^2 \leq 2(a^2 + b^2)$ we get

$$\sum_{i=k+1}^{n} |\langle R_n f^*, u_i \rangle_{\mathbb{R}^n}|^2 \leq 2 \sum_{i=k+1}^{n} |\langle R_n (I - P^{T_{K^\infty}}) f^*, u_i \rangle_{\mathbb{R}^n}|^2 + 2 \sum_{i=k+1}^{n} |\langle R_n P^{T_{K^\infty}} f^*, u_i \rangle_{\mathbb{R}^n}|^2.$$

To control the first term we have

$$\sum_{i=k+1}^{n} |\langle R_n (I - P^{T_{K^\infty}}) f^*, u_i \rangle_{\mathbb{R}^n}|^2 \leq \left\| (I - P^{T_{K^\infty}}) f^* \right\|_{\mathbb{R}^n}^2 = \frac{1}{n} \sum_{i=1}^{n} |(I - P^{T_{K^\infty}}) f^*(x_i)|^2.$$

Then by applying Lemma F.3 to the second term we get the desired result. $\qquad\square$

We recall the following lemma from Rosasco et al. (2010, Theorem 7)

**Lemma F.5.** *(Rosasco et al., 2010) With probability at least $1 - \delta$ over the sampling of $x_1, \ldots, x_n$*

$$\|T_\mathcal{H} - T_n\|_{HS} \leq \frac{2\kappa \sqrt{2 \log(2/\delta)}}{\sqrt{n}}.$$

Now finally we can provide a bound on the labels participation in the bottom eigendirections.

**Theorem F.6.** *Assume $f^* \in L^2(X)$ and let $P^{T_{K^\infty}}$ be the orthogonal projection onto the eigenspaces of $T_{K^\infty}$ corresponding to the eigenvalue $\alpha \in \sigma(T_{K^\infty})$ and higher. Assume that*

$$\left\| (I - P^{T_{K^\infty}}) f^* \right\|_\infty \leq \epsilon'$$

*for some $\epsilon' \geq 0$. Pick $k$ so that $\sigma_k = \alpha$ and $\sigma_{k+1} < \alpha$, i.e. $k$ is the index of the last repeated eigenvalue corresponding to $\alpha$ in the ordered sequence $\{\sigma_i\}_i$. Let $P_k$ denote the orthogonal projection onto $\text{span}\{u_1, \ldots, u_k\}$. Finally assume*

$$n \geq \frac{128 \kappa^2 \log(2/\delta)}{(\sigma_k - \sigma_{k+1})^2}.$$

*Then we have with probability at least $1 - 2\delta$ over the sampling of $x_1, \ldots, x_n$ that*

$$\|(I - P_k) R_n f^*\|_{\mathbb{R}^n} = \|(I - P_k) y\|_{\mathbb{R}^n} \leq 2\epsilon' + \frac{4\kappa \|f^*\|_2 \sqrt{10 \log(2/\delta)}}{(\sigma_k - \sigma_{k+1})\sqrt{n}}.$$

*Proof.* From Lemma F.4 we have

$$\sum_{i=k+1}^{n} |\langle R_n f^*, u_i \rangle_{\mathbb{R}^n}|^2 \leq \frac{2}{n} \sum_{i=1}^{n} |(I - P^{T_{K^\infty}}) f^*(x_i)|^2 + 2 \frac{\|f^*\|_2^2 \lambda_{k+1}}{\sigma_k} \left\| P^{T_\mathcal{H}} - P^{T_n} \right\|_{HS}^2$$

By assumption we have that the first term is bounded by $2(\epsilon')^2$. Now we must control the term

$$2 \frac{\|f^*\|_2^2 \lambda_{k+1}}{\sigma_k} \left\| P^{T_\mathcal{H}} - P^{T_n} \right\|_{HS}^2.$$

By Lemma F.5 we have with probability at least $1 - \delta$

$$\|T_\mathcal{H} - T_n\|_{HS} \leq \frac{2\kappa \sqrt{2 \log(2/\delta)}}{\sqrt{n}}.$$

Then
$$n \geq \frac{128\kappa^2 \log(2/\delta)}{(\sigma_k - \sigma_{k+1})^2}$$
suffices so that the right hand side above is less than or equal to $\frac{\sigma_k - \sigma_{k+1}}{4}$. Thus by Lemma F.2 we have that
$$\left\|P^{T_\mathcal{H}} - P^{T_n}\right\|_{HS} \leq \frac{2}{\sigma_k - \sigma_{k+1}} \|T_\mathcal{H} - T_n\|_{HS} \leq \frac{2}{\sigma_k - \sigma_{k+1}} \frac{2\kappa\sqrt{2\log(2/\delta)}}{\sqrt{n}}.$$
Thus from the above inequality we get that
$$2\frac{\|f^*\|_2^2 \lambda_{k+1}}{\sigma_k} \left\|P^{T_\mathcal{H}} - P^{T_n}\right\|_{HS}^2 \leq \frac{64\kappa^2 \|f^*\|_2^2 \lambda_{k+1} \log(2/\delta)}{\sigma_k(\sigma_k - \sigma_{k+1})^2 \cdot n}.$$
By Proposition 10 in Rosasco et al. (2010) we have separately with probability at least $1 - \delta$
$$\lambda_{k+1} \leq \sigma_{k+1} + \frac{2\kappa\sqrt{2\log(2/\delta)}}{\sqrt{n}}.$$
Note that
$$n \geq \frac{128\kappa^2 \log(2/\delta)}{(\sigma_k - \sigma_{k+1})^2}$$
implies that
$$\frac{1}{\sqrt{n}} \leq \frac{\sigma_k - \sigma_{k+1}}{8\kappa\sqrt{2\log(2/\delta)}},$$
therefore
$$\lambda_{k+1} \leq \sigma_{k+1} + \frac{2\kappa\sqrt{2\log(2/\delta)}}{\sqrt{n}} \leq \sigma_k + \frac{1}{4}(\sigma_k - \sigma_{k+1}) \leq \frac{5}{4}\sigma_k.$$
Thus
$$2\frac{\|f^*\|_2^2 \lambda_{k+1}}{\sigma_k} \left\|P^{T_\mathcal{H}} - P^{T_n}\right\|_{HS}^2 \leq \frac{64\kappa^2 \|f^*\|_2^2 \lambda_{k+1} \log(2/\delta)}{\sigma_k(\sigma_k - \sigma_{k+1})^2 n} \leq \frac{80\kappa^2 \|f^*\|_2^2 \log(2/\delta)}{(\sigma_k - \sigma_{k+1})^2 n}.$$
Thus combined with our previous results we finally get that
$$\|(I - P_k)R_n f^*\|_{\mathbb{R}^n}^2 = \sum_{i=k+1}^n |\langle R_n f^*, u_i \rangle_{\mathbb{R}^n}|^2 \leq 2(\epsilon')^2 + \frac{80\kappa^2 \|f^*\|_2^2 \log(2/\delta)}{(\sigma_k - \sigma_{k+1})^2 n}$$
Thus from the inequality $\sqrt{a+b} \leq \sqrt{2}(\sqrt{a} + \sqrt{b})$ which holds for all $a, b \geq 0$ we have
$$\|(I - P_k)R_n f^*\|_{\mathbb{R}^n} \leq 2\epsilon' + \frac{4\kappa \|f^*\|_2 \sqrt{10\log(2/\delta)}}{(\sigma_k - \sigma_{k+1})\sqrt{n}}.$$
Since $y = R_n f^*$ this provides the desired conclusion. $\qquad\square$

## G  $T_{K^\infty}$ IS STRICTLY POSITIVE

Note that
$$K^\infty(x, x') = \mathbb{E}[\sigma(\langle w, x \rangle_2 + b)\sigma(\langle w, x' \rangle_2 + b)] + \mathbb{E}[a^2\sigma'(\langle w, x \rangle_2 + b)\sigma'(\langle w, x' \rangle_2 + b)][\langle x, x' \rangle_2 + 1] + 1$$
where the expectation is taken with respect to the parameter initialization. It suffices to show that the kernel corresponding to the first term above
$$K_a(x, x') := \mathbb{E}[\sigma(\langle w, x \rangle_2 + b)\sigma(\langle w, x' \rangle_2 + b)]$$
induces a strictly positive operator $T_{K_a}f(x) = \int_X K_a(x, s)f(s)d\rho(s)$. From the discussion in Section C.1 it suffices to show that the RKHS corresponding to $K_a$ is dense in $L^2$. In Proposition 4.1 in Rahimi & Recht (2008a) they showed that the RKHS associated with $K_a$ has dense subset
$$\mathcal{F} := \left\{ x \mapsto \int_\Theta a(w, b)\sigma(\langle w, x \rangle_2 + b)d\mu(w, b) : \int_\Theta |a(w, b)|^2 d\mu(w, b) < \infty \right\}$$
where $\mu$ is the measure for the parameter initialization, i.e. $(w, b) \sim \mu$. Since $C(X)$ is dense in $L^2(X)$ it suffices to show that $\mathcal{F}$ is dense in $C(X)$ which is provided by the following theorem:

**Theorem G.1.** *Let $\sigma$ be L-Lipschitz and not a polynomial. Assume that $\mu$ is a strictly positive measure supported on all of $\mathbb{R}^{d+1}$. Also assume that*

$$\int_{\mathbb{R}^{d+1}} [\|w\|_2^2 + \|b\|_2^2] d\mu(w,b) < \infty.$$

*Then $\mathcal{F}$ is dense in $C(X)$ under the uniform norm.*

*Proof.* We first show that $\mathcal{F} \subset C(X)$. Suppose we have $f \in \mathcal{F}$ and write $f(x) = \int_{\mathbb{R}^{d+1}} a(w,b)\sigma(\langle w,x\rangle_2 + b)d\mu(w,b)$. Well then

$$|f(x) - f(x')| = \left| \int_{\mathbb{R}^{d+1}} a(w,b)[\sigma(\langle w,x\rangle_2 + b) - \sigma(\langle w,x'\rangle_2 + b)]d\mu(w,b) \right|$$

$$\leq \int_{\mathbb{R}^{d+1}} |a(w,b)||\sigma(\langle w,x\rangle_2 + b) - \sigma(\langle w,x'\rangle_2 + b)|d\mu(w,b)$$

$$\leq \int_{\mathbb{R}^{d+1}} |a(w,b)|L|\langle w,x-x'\rangle|d\mu(w,b) \leq \int_{\mathbb{R}^{d+1}} |a(w,b)|L\|w\|_2\|x-x'\|_2 d\mu(w,b)$$

$$\leq L\|x-x'\|_2 \left[ \int_{\mathbb{R}^{d+1}} |a(w,b)|^2 d\mu(w,b) \right]^{1/2} \left[ \int_{\mathbb{R}^{d+1}} \|w\|_2^2 d\mu(w,b) \right]^{1/2}.$$

Thus $f$ is Lipschitz and thus continuous. Now suppose that $\mathcal{F}$ is not dense in $C(X)$. Then by the Riesz representation theorem there exists a nonzero signed measure $\nu(x)$ with finite total variation such that $\int_X f(x)d\nu(x) = 0$ for all $f \in \mathcal{F}$. Well then writing $f(x) = \int_{\mathbb{R}^{d+1}} a(w,b)\sigma(\langle w,x\rangle_2 + b)d\mu(w,b)$ as before we have

$$\int_X \int_{\mathbb{R}^{d+1}} a(w,b)\sigma(\langle w,x\rangle_2 + b)d\mu(w,b)d\nu(x) = 0 \tag{13}$$

Note that

$$\int_{\mathbb{R}^{d+1}} |a(w,b)||\sigma(\langle w,x\rangle_2 + b)|d\mu(w,b)$$

$$\leq \int_{\mathbb{R}^{d+1}} |a(w,b)|[|\sigma(0)| + L|\langle w,x\rangle_2 + b|]d\mu(w,b)$$

$$\leq \int_{\mathbb{R}^{d+1}} |a(w,b)|[|\sigma(0)| + L(\|w\|_2 M + \|b\|_2)]d\mu(w,b) < \infty$$

where we have used Cauchy-Schwarz and the hypothesis on the integrability of $\|w\|_2^2, \|b\|_2^2$ in the last step. Thus the integrand in (13) is $\mu \times \nu$ integrable thus by Fubini's theorem we may interchange the order of integration. To get that

$$\int_{\mathbb{R}^{d+1}} a(w,b) \int_X \sigma(\langle w,x\rangle_2 + b)d\nu(x)d\mu(w,b)$$

and the above holds for any $a \in L^2(\mathbb{R}^{d+1}, \mu)$. Thus $\int_X \sigma(\langle w,x\rangle_2 + b)d\nu(x) = 0$ for $\mu$-almost every $w,b$. However by essentially the same proof as when we showed $\mathcal{F} \subset C(X)$ we may show that $\int_X \sigma(\langle w,x\rangle_2 + b)d\nu(x) = 0$ is a continuous function of $(w,b)$. Thus since $\mu$ is a strictly positive measure on $\mathbb{R}^{d+1}$ this implies that $\int_X \sigma(\langle w,x\rangle_2 + b)d\nu(x) = 0$ for every $(w,b) \in \mathbb{R}^{d+1}$. However by Theorem 1 in Leshno et al. (1993) we have that $\text{span}\{\sigma(\langle w,x\rangle_2 + b) : (w,b) \in \mathbb{R}^{d+1}\}$ is dense in $C(X)$. However by our previous conclusion and linearity we have that $\int g(x)d\nu(x) = 0$ for any $g$ in $\text{span}\{\sigma(\langle w,x\rangle_2 + b) : (w,b) \in \mathbb{R}^{d+1}\}$, which implies then that $\nu$ must equal 0. Thus $\mathcal{F}$ is dense in $C(X)$. $\qquad\square$

Since Gaussians are supported on all of $\mathbb{R}^{d+1}$ we have the following corollary:

**Corollary G.2.** *If $(w,b) \sim N(0, I_{d+1})$ then $K^\infty$ is strictly positive.*

