# OpenReview forum: "Implicit Bias of MSE Gradient Optimization in Underparameterized Neural Networks"
_ICLR.cc/2022/Conference — ICLR 2022 Poster_

### Official Review · Reviewer_VUVP · 2021-10-28

**Correctness:** 4
**Technical Novelty And Significance:** 3
**Empirical Novelty And Significance:** Not applicable
**Recommendation:** 6
**Confidence:** 3

**Main Review:**

This is an interesting theoretical work. The main technique of analysis, if I understood correctly, is to use the infinite width NTK dynamics, which is linear, as an auxiliary term to add and subtract from the actual dynamics, and leverage Duhamel’s principle to study the finite width situations. This is not a completely new trick, as it is often used in the mathematical literature for differential equations; nevertheless, I still think it is remarkable to actually execute the idea (well) for the important problem of neural networks, even if it’s just deterministic gradient flow and 2-layer. I therefore gave a positive rating. Still, there are a few things that I’d like to understand better, hence the followings.

One of my questions is, quantitatively, what regime is exactly the underparameterized regime considered in Theorem 3.5? I’m confused because I didn’t find any assumption on n (number of data). Is it implicitly assumed that Theorem 3.5 applies when $n>m$? Why is the lower bound of $m$ in Theorem 3.5 independent of $n$?

Another question about Theorem 3.5 is, are $\Gamma$ and $T$ really just any numbers that satisfy $\Gamma>1$ and $T>0$? If yes, why not optimize over them to get the best bound?

One more question is, did I understand right about Theorem 3.5 that $exp(-T_K t)r_0$ is decreasing but the bound on its difference from $r_t$ is growing quadratically in t? Wouldn’t this produce a low “signal-to-error” ratio?

I’m also wondering about the gap between the underparameterized regime in Section 3.2 and the overparameterized regime in Section 3.3. If 3.2 really requires $n>m$ (see above), the gap is rather large. What can happen there? Some explanations in the main paper would be very helpful.

I also found the comparison with [Arora et al. 2019] and [Su & Yang 2019] helpful. I understand that this work trains both layers and there are many other important differences (well stated in the paper and thus omitted here), but these are mostly in terms of the results. Could the authors also comment about the differences (and similarities) in terms of the analysis techniques?

A couple of minor observations:

* Section 1.1 Related work, reference Weinan E (2020) should be [E et al. 2020]

* Lemma 2.1 “… semidefinite matrix $G_s$ the time dependent …” probably means “… semidefinite matrix and $G_s$ be the time dependent …”

* No empirical results are provided. This is probably okay, but of course if the tightness of the theoretical results could be supported by empirical evidence, it would be fantastic.


**Summary Of The Paper:**

This paper considers the gradient flow training of 2-layer neural network (both layers are trained) and provides a theoretical tool termed “damped deviations” for improved NTK analysis. One outcome is a characterization of how a neural network deviates, when in an underparameterized regime, from learning the eigenfunctions of the integral operator associated with infinite width NTK at rates proportional to the corresponding eigenvalues.

**Summary Of The Review:**

I found this work very interesting. The proposed theoretical tool of “damped deviations” is useful, and the outcome (the “characterization”, see Summary of The Paper) is a nontrivial step for the important task of going beyond infinite overparameterization.

---

> ### Author Response · Authors · 2021-11-15
> **Author response part 1**
>
> First, thank you for giving the paper a deliberate read and sharing your interpretations and insights.  Indeed we believe the damped deviations equation is the simple and natural thing to do from a differential equations perspective!  One of our big hopes is that the damped deviations equation would make it simple at a high level what is going on, as NTK works have a reputation of being technically complicated and it is easy to get lost in the details.  We think the damped deviations are a very simple but missing interpretation of the dynamics when optimizing the squared error.
>
> The reason why Theorem 3.5 concerns the underparameterized regime is that for the bound to be small, the ratio $p/n$ needs to be small, where $p$ is the number of parameters and $n$ is the number of samples.  Thus in a regime where $p = o(n)$ the second term will go to zero.  Note this is in stark contrast to the typical NTK regime where $p \geq poly(n)$.
>
> The fact that the lower bound on $m$ does not depend on $n$ was essential, in fact this is essentially the key technical difficulty we had to overcome!  In order to be valid in the underparameterized regime you need to be able to send $n \rightarrow \infty$ while keeping $m$ constant.  We had to prove a uniform NTK deviation bound (Theorem 3.4) that does not require the width $m$ to be large relative to the number of samples $n$.  We did this using an epsilon net plus Lipschitzness argument for convergence of the NTK at initialization, combined with a-priori bounds for the time derivatives of the parameter gradients to control deviations throughout time.  In a typical NTK work they require the time dependent NTK matrix to be strictly positive definite, which using the standard techniques would require at least $m \geq n^2$.  All the difficulties of our proof essentially come from having to avoid that sort of argument.
>
> Concerning $\Gamma$ and $T$, we are glad you mentioned this because we have thought about this.  The requirement on width decreases logarithmically with $\Gamma$, but the deviation bound increases polynomially with $\Gamma$.  Thus in the typical case it probably makes most sense to set $\Gamma$ to some moderate constant greater than $1$.  **We have decided for simplicity of presentation it does indeed make sense to set $\Gamma$ to a constant and remove it as a parameter.  We have performed this simplification among others in the statement of Theorem 3.5 in the updated manuscript.**  The constant $T > 0$ represents a stopping time, it is the time up until which the bound holds.  Since we are operating in the setting where the NTK is not positive definite we cannot ensure that the loss is converging to zero, and thus to ensure bounded parameter norms we must stop at some point.
>
> In regard to “One more question is, did I understand right about Theorem 3.5 that $\exp(-T_{K}t) r_0$ is decreasing but the bound on the difference from $r_t$ is growing quadratically in $t$?  Wouldn’t this produce a low signal-to-error ratio?”, this is an excellent question.  We agree that the polynomial dependence on the time $t$ in the bound looks scary at first.  We had anticipated this and this is partly why we introduced Corollary 3.6.  For $\exp(-T_{K} t) r_o$ to learn an eigenfunction $\phi_k$ up to $\epsilon$ accuracy it takes time $t \sim \log(1/\epsilon) / \sigma_k$.  Thus even though the bound depends polynomial with $t$ we have that $t$ only grows logarithmically with the desired accuracy $\epsilon$, thus the bound really translates to polynomial dependency on the inverse of the desired eigenvalue you care about.  Since the NTK has a small number of outlier eigenvalues, this means the bias towards these top eigenvectors is comparatively higher relative to the other directions.  Looking at specific directions rather than all directions at once is another way in which we differ from classical NTK analysis.

---

> ### Author Response · Authors · 2021-11-15
> **Author response part 2**
>
> In response to “I’m also wondering about the gap between the underparameterized regime in Section 3.2 and the overparameterized regime in Section 3.3. If 3.2 really requires $n > m$ (see above) the gap is rather large.  What can happen there?  Some explanations in the main paper would be helpful.”, this is indeed the natural question to ask.  Unfortunately due to the page limit and the number of results we had to introduce we chose to focus on the “what” more than the “how”.  For Theorem 3.5 the bound has two components, the first term that depends on the inverse of the width $1 / \sqrt{m}$ and the second term that depends on the ration $p/n$.  The first term comes from the NTK deviation bound (Theorem 3.4) and the second term comes from having to control the test error rather than just the empirical risk.  If you only care about the empirical risk you really only need to control the first term.   The empirical risk guarantees are of the form:
>
> “Damped deviations equation (simple)" + “NTK Deviation bound (harder)” = ERM guarantee.
>
> Theorem 3.5 can be seen as:
>
> “Damped deviations equation (simple)” + “NTK deviation bound (harder)” + “Empirical process theory arguments (harder)” = control of test error dynamics.
>
> Part of the significance of the damped deviations equation is that results of the form in Arora et al. and Su and Yang follow immediately from the damped deviations equation once one has NTK deviation bounds in hand.  We hoped this fresh perspective would greatly simplify the interpretations of these works.
>
> The differences in techniques between our work and the work of Arora and Su and Yang are interesting to unpack.   We will refer to the second paragraph of our response to reviewer “bkHH”.
>
> Re minor observations:
> * We corrected the citation for E et al..
> * Thank you for pointing the missing phrasing out!  We have corrected it.
> * This is an important point that we thought of.  Ultimately due to the hefty length of this work we ultimately decided to leave such investigations out, however there is still important work that can be done on that front.

---

> > ### Author Response · Authors · 2021-11-29
> > **Following up**
> >
> > Dear Reviewer VUVP, we have made thorough efforts to address all comments and questions from your initial review, in particular regarding the regime that is being addressed in our work.  Most notably, following your suggestion, we have simplified the presentation of Theorem 3.5.  We have removed $\Gamma$ as a parameter from Theorem 3.5, as well as simplifying the requirements on the width $m$ and the final bounds by treating $\kappa$ and $\sigma_1$ as constants.  Since you indicated that you found our work very interesting, useful, and nontrivial, we hope these clarifications and updates will persuade you to consider raising your recommendation.  In any case thank you for your contributions, and please let us know any final comments/questions before the end of the discussion phase.

---

> > > ### Comment · Reviewer_VUVP · 2021-11-29
> > > **increase my rating to 7**
> > >
> > > I thank the authors for their efforts. A good proportion of my concerns are addressed. Although I still have a handful of questions (e.g., see my question on low "signal-to-error" ratio, and I still wonder about what happens in the gap), I'd like to increase my rating to 7. Since the technique of analysis used is not new, I prefer to give a 7 instead of an 8. The system doesn't allow a 7 this time, but I hope the AC could take this into account.

---

> > > > ### Author Response · Authors · 2021-11-30
> > > > **Signal-to-error ratio**
> > > >
> > > > Thank you for your response and consideration of our simplifications.  With regard to the "signal-to-error" ratio we refer back to the final paragraph of part 1 of our initial response to your review:
> > > >
> > > > _"We agree that the polynomial dependence on the time $t$ in the bound looks scary at first.  We had anticipated this and this is partly why we introduced Corollary 3.6.  For $\exp(-T_k t) r_o$ to learn an eigenfunction $\phi_k$ up to $\epsilon$ accuracy it takes time $t \sim \log(1/\epsilon) / \sigma_k$.  Thus even though the bound depends polynomial with $t$ we have that $t$ only grows logarithmically with the desired accuracy $\epsilon$, thus the bound really translates to polynomial dependency on the inverse of the desired eigenvalue you care about."_
> > > >
> > > > We recognize the scaling of the parameters was not sufficiently clear in the initial submission.  **Thus following your guidance we added the following discussion to page 7 of the updated manuscript:**
> > > >
> > > > "To interpret the results, we observe that to track the dynamics for eigenfunctions corresponding to eigenvalue $\sigma_k$ and above, the expression under the $\tilde{O}$ needs to be small relative to $\frac{1}{\sigma_k}$.  Thus the bias towards learning the eigenfunctions corresponding to large eigenvalues appears more pronounced.  When $t = \log(\lVert r_0 \rVert_2 / \epsilon) / \sigma_k$, we have that $\lVert P_k \exp(-T_{K^\infty}t) r_0 \rVert_2 \leq \epsilon$.  Thus by applying this stopping time we get that to learn the eigenfunctions corresponding to eigenvalue $\sigma_k$ and above up to $\epsilon$ accuracy we need $\frac{t^2}{\sqrt{m}} \lesssim \epsilon$ and $\frac{t^2\sqrt{p}}{\sqrt{n}} \lesssim \epsilon$ which translates to $m \gtrsim \sigma_k^{-4} \epsilon^{-2}$ and $n \gtrsim p \sigma_k^{-4} \epsilon^{-2}$.  In typical NTK works the width $m$ needs to be polynomially large relative to the number of samples $n$, where by contrast here the width depends on the inverse of the eigenvalues for the relevant components of the target function."
> > > >
> > > > We hope this clarifies what is going on and suffices to ease your qualms.  Otherwise if you have remaining questions please let us know as we remain attentive.

---

### Official Review · Reviewer_hz1D · 2021-11-02

**Correctness:** 4
**Technical Novelty And Significance:** 3
**Empirical Novelty And Significance:** Not applicable
**Recommendation:** 5
**Confidence:** 3

**Main Review:**

First, I would like to say that I am nor an expert neither a contributor of the abundant NTK literature. Hence, even if I know well the phenomenon as well as kernel methods, I might under-appreciate the novelty of the bounds presented in the paper under review.

That being said, I have to say that the *story of the paper* is clearly written with a nice introduction that makes the reader eager to discover what is really in it. The question: « to what extent is the NTK informative outside the NTK regime ?» could be of primal importance and introduces nicely the paper. *However*, when it comes to the paper itself, I have to say that I was a bit disappointed.

First, the two last paragraphs of the related work are a bit fuzzy comparing bounds with previous papers that are not so much in the exact setting, and furthermore not explaining why it is interesting to have improved these bounds, or considered another setting. That made me think: *what is the actual message of the paper ?* This impression is reinforced by the fact that the title indicates that the focus will be on the implicit bias of neural networks whereas it is really on the NTK regime and hence on the implicit bias of kernel methods.

Going further, my main problem on this paper comes with the second disappointment: in my opinion, the paper does not really study neural nets outside the NTK regime as either it compares to the NTK itself for the test error (Theorem 3.5) or either show convergence to 0 of the train error (Theorems 3.7 and 3.8). In other words, the authors suggest that their damped deviation analysis could be the first stone towards getting a bit out of the NTK regime, but under the current form, I am not very convinced by the argument.  On the one side, I understand that obviously this is not the aim of Theorems 3.7 and 3.8 that could be nice technical contributions of their own (once again I lack some knowledge in this specific domain and do not know how impactful these results are). On the other side, in my opinion, Theorem 3.5 and Corollary 3.6 really describe regimes (underparametrized with large $m$ and $n$) under which the trajectory of the gradient flow remains in the vicinity of the trajectory of the true NTK gradient flow. Then the fact that the estimator given by the flow learns first eigenfunctions of the integral operator related to large eigenvalues in usual in kernel methods. In my opinion getting outside of the NTK regime requires other analysis than the one given in Theorem 3.4: that is I cannot consider new an analysis that presents systematically deviations from the Neural Tangent Kernel.

**In conclusion**, I am not saying that the results are not interesting, but in my opinion they are quite *expected* and are not conceptual novelties but rather technical ones. To what extend these are new, I do not know.




Minor Typos:

- page 2, 3rd paragraph : « generalizeation »
- page 2, 4th paragraph : « is is smaller by… »
- page4, authors may precise that $L^2$ is in fact $L^2_\rho$, precise also that $\kappa$ is finite under mild assumptions on the model but not always the case
- page4, lemma 2.1, put a « and » before $G_s$

**Summary Of The Paper:**

In the paper under review, the authors study the behaviour of gradient flow in the kernel regime of one hidden layer neural networks. More precisely, they show deviations bounds on the test error  w.r.t. the true NTK gradient flow in the underparametrized regime and train error bounds in the overparametrized regime.

**Summary Of The Review:**

As a non-specialist of the NTK regime, I may underestimate the technical contribution of the present paper, but I was not surprised by the results and see the submission as a detailed and nice technical improvement of known results in the NTK regime. Hence, I am between neutral and unsatisfied by the paper.

---

> ### Author Response · Authors · 2021-11-15
> **Author response part 1**
>
> Thank you for taking the time to unpack our paper; you raise some important points.  As to your first point about being “fuzzy comparing bounds with previous papers that are not so much in the exact setting”, could you please specify how we could make the comparison more clear or fair?  As we see it the fundamental differences were stated multiple times, namely: gradient descent v.s. gradient flow, training hidden layer without bias versus both layers with bias, ReLU activation versus smooth activation (see Related Work pg 2 and Main Results pgs 6,7,8).  We also make multiple statements of the form “due to the different settings the results are not directly comparable” (see pg 2), and “due to the different settings we do not claim superiority over this work” (see pg 8).  **To further emphasize this, we have moved these statements to occur before any comparisons of bounds in the related work section in our newest manuscript.**  We take pride in giving proper credit to prior work (we cited over 40 works in his paper), and thus would be genuinely curious to learn any important differences we missed.
>
> Your second concern “not explaining why it is interesting to have improved these bounds, or considered another setting” is a natural question to ask which we can address.  As we see it the main contribution is to introduce the training of both layers and to unify these results under the damped deviations perspective.  Training both layers is what is done in practice, and proving the results for training both layers offers technical difficulties that training only the hidden layer does not have.  As one example, for smooth activations when you fix the outer layer the gradient corresponding to the hidden layer parameters is Lipschitz as a function of the hidden layer parameters (see for example Lemma 6.8 in Oymak et al. [1].).  When you train the outer layer this is not the case as the deviations will depend on the norm of the other layer weights which can increase throughout training.  Thus even for smooth activations the extension going from training only the hidden layer to both layers is nontrivial.  Indeed Oymak et al. [1], Arora et al. [2], Su and Yang et al. [3], all consider training only the hidden layer (note Oymak et al. [1] considers both ReLU and smooth activations).  If training both layers was a mere corollary we believe this would have been shown at some point.  If we want to have confidence that these results can hold for training both layers as in practice, the work has to be done as this issue will not resolve on its own.  We can’t let training only the hidden layer be the final word on these seminal works.
>
> As to your comment “the title indicates that the focus will be on the implicit bias of neural networks whereas it is really on the NTK regime and hence on the implicit bias of kernel methods.”, you are correct that the spectral bias of kernel methods is well understood.  However, what was not well understood to our knowledge before this paper is that the neural network can inherit the bias of the kernel method even outside of the heavily overparameterized regime that is typical of NTK papers.  The point is that the spectral bias observed empirically to be exhibited by neural networks to some extent could simply be from the neural network inheriting kernel-like behavior at the beginning of training as we exhibit in this work.
>
> Your point about how we are comparing to kernel dynamics and thus still are in a type of “NTK regime” is well-taken.  To be precise the key contribution of this paper was to escape the heavily overparameterized regime where the time-dependent NTK Gram matrix is strictly positive definite throughout training, which we had considered synonymous with the “NTK regime”.  **Your point is well-taken and we have updated the wording in the introduction to clarify this important detail.**  To elaborate on the technical significance of this obstacle, in the underparameterized case the NTK matrix cannot be even full rank, thus strict positive definiteness is impossible from the beginning.  When the NTK matrix is not full rank you cannot ensure that the loss is converging to zero, and thus these obstacles must be overcome.  The fact that any sort of NTK analysis could be possible in the underparameterized regime was completely surprising to us and to our knowledge had not been addressed by the community.
>
> As far as the contributions of Theorem 3.7 and Theorems 3.8, we would refer back to the second paragraph of this response.

---

> > ### Comment · Reviewer_hz1D · 2021-11-16
> > **Part 1-2**
> >
> > - *Comparison with other works*: I am sincerely sorry that my comment on fuzziness was... fuzzy. What I wanted to say is the following: I understand that the authors' setting is different than the previous ones (bias, activation, layers), so dropping unreadable bounds that do not compare to previous ones is for me non-relevant here unless there are well commented and explained. I appreciate the fact that the authors cited previous work and refer precisely to their *different* setting.
> >
> > - *Training both layer*: I admit that  do not know well this literature, but the fact that the authors claim to be the first to train both layer is important. It has to be emphasised as one novelty.
> >
> > - *Bias*. If the training is stuck in the NTK regime then the training is guided by a Kernel-like bias. I am just saying that this is kind of expected.
> >
> > - *NTK for underparametrized*: I understand the technical difficulty. Thanks.
> >
> > - *Devil in the details*: I refer to the previous line: I understand that there might be a high technical difficulty in the underparametrized case. And as I said I did not work personally on NTK so that the fact that having a result while the network is underparametrised *was not surprising to me* because $m, n$ are going to infinity and permit a NTK regime. Once again, I may have under-appreciated technical difficulties but if people failed before and that the aim of the authors is to develop the NTK cartography then I understand. The fact is that the title is about spectral bias... so I am a bit confused on the overall message. Does the spectral bias not appear also in the overparametrized regime ?

---

> > > ### Author Response · Authors · 2021-11-19
> > > **Response to reviewer hz1D's first reply**
> > >
> > > Thank you for your prompt response.
> > >
> > > * Thank you for elaborating on this point.  We would like to further emphasize that we believe the way we compared bounds is completely standard for such works (citing difference in activation function, architecture, parameter dependencies, etc.).  Please see paragraphs 2, 3, 4 on page 3 of Du et al. [1] comparing bounds for smooth activations versus ReLU in another work, or paragraph 1 of page 12 in Oymak et al. [2] where they compare bounds for a shallow network to those of a deep network.  Perhaps this practice of comparing bounds and assumptions is confusing at times, but it is certainly conventional.  **In response to your comment about bounds being unreadable, we have simplified the presentation of Theorem 3.5 by suppressing the constants $\kappa$ and $\sigma_1$ under the $\tilde{O}$ notation and setting $\Gamma$ to a fixed value to remove it as a parameter to simplify the presentation.**
> > >
> > > * We are certainly not the first to study convergence in the case of training all parameters, however we are to our knowledge the first to show Theorem 4.1 in Arora et al. holds in any setting where all parameters are trained.  The only other work we are aware of is Basri et al. [3] where they study a deep ReLU network without bias where the first and last layer are fixed.  Since the first and last layer are fixed this cannot apply to a shallow network even for the ReLU activation.  We had cited Basri et al. [3] in the main text in the original draft but unfortunately did not include this notable result in the Related Work section.  **We have updated the draft to mention this result in the related work section.**  As far as the work of Su and Yang [4] we are not aware of any extensions except our own.  Thank you for acknowledging the novelty of the training of all parameters in these results.  All that we ask is that this important point be taken into account in your final decision.
> > >
> > > * From a very high-level viewport, of course conditional on being in a sort of kernel regime it may not be surprising.  However the fact that any kind of kernel regime even exists in this situation was what was highly surprising to us and has hitherto not been demonstrated to our knowledge.
> > >
> > > * Thanks for clarifying.
> > >
> > > * Thank you for bringing this up.  As far as we know in the overparameterized regime no one has been able to demonstrate spectral bias off of the training set.  Cao et al [5] provide an empirical version of our Theorem 3.5 where the eigenfunctions are replaced with the vectors given by their restriction to the training set.  Arora Theorem 4.1 describes the training residual projected along eigenvectors of the NTK, which are similar vectors to those in the projections in Cao et al. [5] but not exactly the same.  In the overparameterized setting the empirical projections being small does not imply the actual projections are small in the same way that small train error does not imply small test error in the overparameterized case.  If someone could demonstrate spectral bias off the training set in the overparameterized regime we would find that highly worthwhile and interesting.
> > >
> > > [1] Simon S. Du, Jason D. Lee, Haochuan Li, Liwei Wang, Xiyu Zhai: Gradient Descent Finds Global Minima of Deep Neural Networks. ICML 2019: 1675-1685
> > > URL: http://proceedings.mlr.press/v97/du19c/du19c.pdf
> > >
> > > [2]  Samet Oymak and Mahdi Soltanolkotabi "Towards moderate overparameterization: global convergence guarantees for training shallow neural networks", 2019 http://arxiv.org/abs/1902.04674
> > >
> > > [3] Ronen Basri, Meirav Galun, Amnon Geifman, David Jacobs, Yoni Kasten, and Shira Kritchman. Frequency bias in neural networks for input of non-uniform density. In Hal Daume III and Aarti ´ Singh (eds.), Proceedings of the 37th International Conference on Machine Learning, volume 119 of Proceedings of Machine Learning Research, pp. 685–694. PMLR, 13–18 Jul 2020. URL https://proceedings.mlr.press/v119/basri20a.html.
> > >
> > > [4] Lili Su and Pengkun Yang. On learning over-parameterized neural networks: A functional approximation perspective. In H. Wallach, H. Larochelle, A. Beygelzimer, F. d'Alche-Buc, E. Fox, and ´ R. Garnett (eds.), Advances in Neural Information Processing Systems, volume 32. Curran Associates, Inc., 2019. URL https://proceedings.neurips.cc/paper/2019/file/ 253f7b5d921338af34da817c00f42753-Paper.pdf.
> > >
> > > [5] Yuan Cao, Zhiying Fang, Yue Wu, Ding-Xuan Zhou, and Quanquan Gu. Towards understanding the spectral bias of deep learning, 2020.

---

> > > > ### Author Response · Authors · 2021-11-22
> > > > **Ongoing feedback**
> > > >
> > > > Thanks again for your initial feedback. We made thorough efforts to address all your comments.  We hope that the technical contributions of tracking the dynamics in the underparameterized case as well as the important introduction of training all parameters to extend works in the overparameterized case will not be overlooked and will motivate you to raise your recommendation.  Kindly do let us know if there is anything else you would like to see clarified or improved before the end of the rebuttal period. We remain attentive to your feedback!

---

> > > > > ### Comment · Reviewer_hz1D · 2021-11-22
> > > > > **Thanks**
> > > > >
> > > > > Thank you for you comments. For now, I think that I have no other questions and, the contribution being merely technical I refer to specialists of the field to judge relevance. I tend to be neutral in the decision process.

---

> > > > > > ### Author Response · Authors · 2021-11-22
> > > > > > **Thank you for response**
> > > > > >
> > > > > > Thank you for being honest. We understand that you want to be neutral and refer to specialists in the field to judge the relevance of our work. However, a recommendation of 5 with a confidence of 3 is not neutral and it also does not reflect your statement about not not feeling confident to judge the relevance of our work. We will defer to Area Chair during the process to decide what is proper protocol in such situations

---

> ### Author Response · Authors · 2021-11-15
> **Author response part 2**
>
> Believe us when we say we are aware of the limitations of the NTK model as well as anyone, and thus your criticism in this regard is well taken.  Nevertheless we will emphasize that the damped deviations perspective does allow for comparing against other kernels besides $K^\infty$.  **We have updated the draft to include a discussion about comparing against other kernels in the last paragraph of Section 3.1.  As a proof-of-concept we have included a proof of a version of Theorem 3.5 where the kernel is chosen to be $K_0$ and the parameter initialization is arbitrary and deterministic in Appendix C.8 Theorem C.15.**  Furthermore, it is largely because of the limitations of the NTK model that it is so important to form a catalogue of which regimes the NTK dynamics are present, and adding the underparameterized setting to that catalogue is a nontrivial but important step.  To say “I cannot consider new an analysis that presents systematically deviations from the Neural Tangent Kernel” is a bit like saying I cannot consider new a generalization bound that uses a metric entropy argument; using a fundamental tool in new and unexpected ways is not trivial.  NTK works tend to be conceptually simple but technically difficult, and the devil is in the details.  Yes the NTK has limits, but the first step to moving beyond a tool is to sharpen a tool to its limits.  We appreciate your feedback about communicating the message of the paper better.  We hope that the technical innovations will not be overlooked and will be taken into account in the final review.
>
> Re “Minor Typos”: Thank you for pointing these out so that we can clarify the exposition!  We have fixed these typos.  Also note that $\kappa$ in our setting is always finite since the domain is bounded and the Kernel is continuous.  We have added a sentence to clarify this.
>
> References:
>
> [1] Samet Oymak and Mahdi Soltanolkotabi. Toward moderate overparameterization: Global convergence guarantees for training shallow neural networks. IEEE Journal on Selected Areas in Information Theory, 1(1):84–105, 2020.
>
> [2] Sanjeev Arora, Simon Du, Wei Hu, Zhiyuan Li, and Ruosong Wang. Fine-grained analysis of optimization and generalization for overparameterized two-layer neural networks. In Kamalika Chaudhuri and Ruslan Salakhutdinov (eds.), Proceedings of the 36th International Conference on Machine Learning, volume 97 of Proceedings of Machine Learning Research, pp. 322–332. PMLR, 09–15 Jun 2019. URL https://proceedings.mlr.press/v97/arora19a. html.
>
> [3] Lili Su and Pengkun Yang. On learning over-parameterized neural networks: A functional approximation perspective. In H. Wallach, H. Larochelle, A. Beygelzimer, F. d'Alche-Buc, E. Fox, and ´ R. Garnett (eds.), Advances in Neural Information Processing Systems, volume 32. Curran Associates, Inc., 2019. URL https://proceedings.neurips.cc/paper/2019/file/253f7b5d921338af34da817c00f42753-Paper.pdf.

---

### Official Review · Reviewer_bKHH · 2021-11-03

**Correctness:** 4
**Technical Novelty And Significance:** 3
**Empirical Novelty And Significance:** 4
**Recommendation:** 8
**Confidence:** 3

**Main Review:**

Strengths
----
- The work provides an original treatment of neural networks in under-parametrized regime. A number of new results are established.
- The paper is well-written and easy to follow.
- The paper appears to be technically correct. The theoretical setting of the paper is clearly stated.
- The tecniques developed in this paper are powerful enough to recover and extend previous works even in the over-parametrized regime.

Limitations
---
- My main critic is that the ssumptions of the paper exclude nonsmooth activation functions,  for example (perhaps) the most important activation function known to both practitioners and theoreticians alike: the ReLU. Which part aspect of the proofs needs smoothness of the activation function ?
- The analysis in this paper is for continuous time gradient-descent (i.e gradient-flow). Which aspects of the results would continue to hold after time discretization ?
- There have been other contributions on the function-space view of neural gradient descent on neural networks (for example Ongie et al. "A FUNCTION SPACE VIEW OF BOUNDED NORM INFINITE WIDTH RELU NETS: THE MULTIVARIATE CASE", ICLR 2020). Is there any comparison to be made here ?

Questions
---
- All the results in this paper appear to be about training error. Can anything be said about generalization error, i.e $\mathbb E_{x \sim \rho}[r_t(x)^2]$ ?


**Summary Of The Paper:**

This work studies the implicit bias of optimizing underparametrized neural networks with MSE loss function and establishes upper-bounds on the training error. One of the takeaways of the paper is that under-parametrized neural networks first learn the most important eigenfunctions associated with the NTK integral operator. This is what the work posits as the implicit bias of gradient-descent in this setting. A key tool developed in the paper is the concept of damped deviations, where in the evolution of MSe w.r.t to one kernel is related to that of another kernel and the damped "distance" between these kernels. As a byproduct, the paper uses this tool to extend certain results in the literature in the over-parametrized setting.

**Summary Of The Review:**

A new function-space view of gradient-flow on neural networks. Establishes that in function space, dynamics of training (when projected along the most important eigendirections of the associated kernel integral operator, is very similary to NTK dynamics, even in under-parametrized regime.


After authors' response
---

My concerns have been fully addressed by the reviewers. I'm raising my score from 6 to 8, and therefore strongly recommending the paper for acceptance.

---

> ### Author Response · Authors · 2021-11-15
> **Author response**
>
> First, thank you for reading the paper and providing a clear, helpful review.  Your first point about ReLU is a natural and important question.  The area of the proof where we use the boundedness of the second derivative is when we are proving the NTK deviations bounds which requires getting a bound on the time derivative of the gradients of the network with respect to the parameters.  To prove the same deviation bound for ReLU would take additional care, for example controlling how many of the activation patterns change is the route that was taken in the works of Arora et al. [1] and Su and Yang [2] that we cite.  The closest we could get to ReLU without modifying the proofs would be to use a softmax approximation to ReLU, say $\ln(1 + \exp(\alpha x)) / \alpha$.  The width will need to grow with the norm of the second derivative, thus this is where you will pay a penalty as $\alpha \rightarrow \infty$. The constants that depend on the norm of the second derivative are tracked in the appendix if one wanted to unpack this. We added a remark in Appendix B.7, where we discuss the specific scaling for the case of a Sofplus approximation.
>
> The question about gradient descent versus gradient flow is also an important question.  Mathematically as far as we know there is no general theorem that allows one to port results between the two.  However gradient flow is typically viewed as a fair approximation to gradient descent when the step sizes are small.  In this particular work we think the gradient flow greatly simplifies the presentation and clarifies what’s going on, as the damped deviation equation is the natural equation to write from an ODE perspective.  Without this high-level view it is easy to get overwhelmed in the technicalities when working solely from a gradient descent perspective.  For the discrete-time version the term corresponding to $\exp(-T_{K^\infty}t)$ is the matrix $(I - \eta G^\infty)^k$ if we think of $t = k \eta$ where $\eta$ is the step size.  Thus the damping operator corresponds to a damping matrix.  The damping matrices appear in the analysis of Arora and Su and Yang, however looking at the damping along specific components is absent from the analysis in those works as far as we can tell.  Part of our goal with introducing the damped deviations equation was to show how the works of Arora et al. [1] and Su and Yang [2] when viewed from a continuous time ODE perspective can become much clearer and intuitive.  Nevertheless, gradient descent proofs introduce technical difficulties which is why we made sure to emphasize in the related works section this important difference.
>
> As for your third point, indeed the function-space view has become a common angle of analysis in modern machine learning.  The paper by Ongie et al. is an excellent work, but as we see it it is addressing a much different question.  We view their work as addressing what functions can certain neural network models represent (in particular infinite width networks with bounded parameter norms), rather we seek to characterize the functions imparted from gradient optimization.  Thus we view their work as lying more in the realm of expressivity or representation capacity of neural networks rather than optimization.
>
> As for your question: “All the results in this paper appear to be about training error.  Can anything be said about generalization error?”, **the answer is yes!**  The main result, Theorem 3.5, is talking about the dynamics of the actual test error along different components, not the empirical risk.  This can be used directly to get a test error bound as in Corollary 3.6.  We did believe that some readers could be confused by this, which is why we introduced Corollary 3.6 to help clarify this point.  Indeed **the fact that Theorem 3.5 is describing the actual risk is one of our essential contributions in this paper**, as previous works have addressed the dynamics of the empirical risk in the overparameterized setting fairly well such as Arora et al. [1].  However, you are correct that Theorem 3.7, Theorem 3.8, and Corollary 3.9 are concerning the empirical risk.  We hope the important point about the implications of Theorem 3.5 for the test error will be taken into account in the final review.
>
> Thank you for your helpful review and discussion.
>
> [1] Sanjeev Arora, Simon Du, Wei Hu, Zhiyuan Li, and Ruosong Wang. Fine-grained analysis of optimization and generalization for overparameterized two-layer neural networks. In Proceedings of the 36th International Conference on Machine Learning, volume 97 of Proceedings of Machine Learning Research, pp. 322–332. PMLR, 09–15 2019. URL https://proceedings.mlr.press/v97/arora19a.html.
>
> [2] Lili Su and Pengkun Yang. On learning over-parameterized neural networks: A functional approximation perspective. In Advances in Neural Information Processing Systems, volume 32. 2019. URL https://proceedings.neurips.cc/paper/2019/file/253f7b5d921338af34da817c00f42753-Paper.pdf.

---

> > ### Author Response · Authors · 2021-11-23
> > **Comment before end of rebuttal phase**
> >
> > Thank you for your feedback which has helped us improve our manuscript.  We hope that the important point about Theorem 3.5 (and Corollary 3.6) tracking the test/generalization error dynamics is taken into account in the final review as we view this as our most fundamental contribution of the manuscript.  Since your original comment asked if we can say anything about the generalization error, we hope this may persuade you to improve your score.  In any case thank you for your contributions.

---

> > > ### Comment · Reviewer_bKHH · 2021-12-03
> > > **Wrapping up**
> > >
> > > Thanks for your detailed answers to my questions and concerns. I've raised my score from 6 to 8. Nice work!

---

### Official Review · Reviewer_Q2fJ · 2021-11-04

**Correctness:** 4
**Technical Novelty And Significance:** 2
**Empirical Novelty And Significance:** Not applicable
**Recommendation:** 6
**Confidence:** 3

**Main Review:**

The paper is well-written and somehow clear and easy to follow. As far as I checked, it is technically sound (I did not go through all the appendices). The authors added plenty of discussions and comparison with previous work. I particularly appreciated the residual evolution equation with the damped deviations, which is quite simple, giving good intuition on the dynamics, and is new as far as I can tell. However I have a few concerns:

I am not convinced about the novelty of the contributions in this paper. The uniform Lipschitzness of the Kernel function + a bound on the change of the weights were obtained in several papers (e.g., 'On the linearity of large non-linear models: when and why the tangent kernel is constant', Liu et al.), which are the main ingredients of the uniform NTK concentration. Considering the change of the residuals (admittedly, only on the empirical data) along the eigenvectors of the NTK was already considered in 'Towards Understanding the Spectral Bias of Deep Learning’, Cao et al. While there is value in working out the technical details, I think the results in this paper are quite expected and not much surprising. At the start of the dynamics, the weights are close enough to initialization for the evolution to follow the NTK and be biased toward the top eigenvectors.

There has been a lot of works that are studying the NTK regime. Here, the authors consider the change of the function along the data-independent eigendirections of the NTK (the fact that we are using early stopping is another clue that we are not escaping the NTK regime). Several works have showed convincingly that the interesting question is to study how GD escape the RKHS and learn functions that are not aligned with NTK, which I am not convinced can be done with these techniques. Furthermore, the spectral bias is well understood  for linear models and kernel methods, which is approximately the behavior of models at the beginning of GD so that we don’t need much overparametrization in order to still be able to get NTK-like behavior.

If the point is to study the underparametrized regime, other work like 'Generalization error of random features and kernel methods: hypercontractivity and kernel matrix concentration’ (Mei et al.) showed that training only the second layer (random features model), $\min(n,m) \geq d^{\delta} / \sigma_k$ is enough to learn perfectly the top $k$ eigenvectors, which is much tighter than the results in Corollary 3.6. I am not sure that developing more techniques to study the NTK regime is interesting or helpful, especially if they are not tight with respect to the dependency on the different parameters.

**Summary Of The Paper:**

In this paper, the authors study gradient flow on the empirical squared loss. More specifically, they consider the evolution in function space of the residuals and show that it can be written as the sum of 1) the residual obtained by training with the NTK operator (infinite width and samples, no training of the kernel), and 2) damped deviations between time dependent kernel matrix and the NTK operator. Based on this decomposition and a uniform deviation bound on the NTK, they show that 1) in the underparametrized regime, GD at the beginning of the dynamics learn eigenfunctions of NTK at rates proportional to the corresponding eigenvalues; 2) in the overparametrized regime, they derive new bounds for the `kernel regime’ to appear; 3) in the overparametrized regime, if we further assume that the target function is aligned with the top eigenvectors of the NTK, then a milder overparametrization is required.

**Summary Of The Review:**

I am not convinced that the current paper is bringing anything new to what is already known about the `spectral bias’ in the NTK regime, even though I believe the technical contribution to be non-trivial. For that reason, my recommendation is marginally below the acceptance threshold. If the authors can convince me that this damped equation can be used in more interesting ways (take target function in a different RKHS or use $K^\infty$ that is not the NTK etc.) or that I misunderstood or missed some important conceptual/technical contributions, I will raise my recommendation!

---

> ### Author Response · Authors · 2021-11-15
> **Author response part 1**
>
> First, thank you for giving the paper an honest read and a thoughtful review.  In response to “The uniform Lipschitzness of the Kernel function + a bound on the change of the weights were obtained in several papers…”, we were not aware of the paper by Liu et al., thank you for sharing the reference. Indeed their work, as we understand it, does offer an alternative uniform NTK deviation bound, and using their bounds could allow us to potentially extend the results of this paper to deep networks and other architectures, which is an exciting avenue to pursue!  So thank you again for bringing this to our attention.  **We removed the comment about our NTK deviation bound being of independent interest from the main text in light of their work.  We have also cited their paper and mention their bound as an alternative to our proof of NTK deviations at the beginning of Appendix B.**  Nevertheless, we would like to emphasize that the uniform NTK deviation bound was ultimately a means to an end, namely proving Theorem 3.5.  **As far as we know tracking the test error dynamics in the underparameterized setting in this fashion is new, as well as the insight that the damped deviations equation provides a simple unifying perspective for NTK convergence results when dealing with the squared error.**
>
> The work “Towards Understanding the Spectral Bias of Deep Learning” by Cao et al. is indeed notable which is why we cited it on the top of page 7.  **Nevertheless, thanks to your feedback after taking a second look we have decided that the work by Cao et al. definitely deserves a more prominent discussion in the related work section and we have updated the draft to give it a proper overview.**  Nevertheless, the differences between our work and theirs are significant.  First, the networks they consider are significantly wider, in their proof of Theorem 4.2 they explicitly state that the width $m \geq \tilde{O}(\max\{\lambda_{r_k}^{-14}, \epsilon^{-6}\})$ where $\lambda_{r_k}$ is the cutoff eigenvalue.  Versus by contrast in our work to have the projection onto the top eigenvectors be bounded by epsilon in L2 norm (the analog of their empirical quantity) this requires setting $t = \tilde{\Omega}(1/\sigma_k)$ which translates to $m =\tilde{\Omega}(\sigma_k^{-4} \epsilon^{-2})$.  **More importantly, we view the difference between tracking the empirical quantities versus the actual quantities to be highly nontrivial.**
>
> It is true that the neural network adapting to learn functions that are not aligned with the NTK is an important phenomenon that has not been adequately addressed by the theory; and indeed this is an important open problem for theorists.  However, demonstrating that NTK-like behavior is not peculiar to the heavily overparameterized case and in fact can be inherited in the total opposite regime, the underparameterized setting, is important to catalogue.  Furthermore, the damped deviations equation allows one to compare the dynamics against other kernels aside from $K^\infty$.  You could compare against $K_0$ or the kernel corresponding to some subset of parameters such as the random feature kernel corresponding to the outer layer.  **We have updated the manuscript to include a discussion about comparing against other kernels in the last paragraph of Section 3.1.  As a proof-of-concept we have included a proof of a version of Theorem 3.5 where the kernel is chosen to be $K_0$ and the parameter initialization is arbitrary and deterministic in Appendix C.8 Theorem C.15.**
>
> It is not clear to us what was meant by “we don’t need much overparametrization in order to still be able to get NTK-like behavior”.  The main result Theorem 3.5 is applicable in the underparameterized case, not the overparameterized case.  If you’re talking about solving the ERM problem in the overparameterized case as far as we know quadratic overparameterization $m \geq n^2$ (see for example “Towards moderate overparameterization: global convergence guarantees for training shallow neural networks’’ By Oymak et al.) is state of the art for solving the ERM problem for general labels.  We would be happy to see any references that improve on this, or perhaps a reference could clarify what you mean.

---

> > ### Author Response · Authors · 2021-11-22
> > **Ongoing feedback**
> >
> > Thank you for your helpful feedback.  We have made a systematic effort to address your concerns in the updated manuscript.  We hope that the flexibility of the damped deviations point-of-view as well as our improvements to existing results in the overparameterized case by introducing the training of all parameters will be taken into account and will motivate you to consider raising your recommendation.  Kindly do let us know if there is anything else you would like to see clarified or improved before the end of the rebuttal period. We remain attentive to your feedback!

---

> > > ### Comment · Reviewer_Q2fJ · 2021-11-23
> > > **Response to rebuttal**
> > >
> > > Thank you to the authors for the detailed answer and considering my comments.
> > >
> > > I agree with all the technical remarks made in the rebuttal to the different reviewers. Sorry fo the confusion, when I wrote "we don’t need much overparametrization in order to still be able to get NTK-like behavior”, I meant "does not need many neurons": as when we remain at the beginning of GD, the first order Taylor expansion remains correct because the weights did not move too much from initialization, and the empirical kernel matrix has enough samples (data points) for its first large eigenvectors to coincide with the eigenvector of the infinite-width kernel operator. This does not mean that it is an easy result to obtain.
> > >
> > > To the best of my knowledge, this paper indeed provides some technical contribution in a regime that was previously not considered, with a residual evolution equation in terms of damped deviations that is quite nice. I am raising my recommendation to 6, because my concerns are mostly a matter of opinion, rather than novelty or technical soundness.

---

> ### Author Response · Authors · 2021-11-15
> **Author response part 2**
>
> In regard to the work ``Generalization error of random features and kernel methods: hypercontractivity and kernel matrix concentration’’ (Mei et al.), please note that the random feature model is a different setting than what we consider.  Wide shallow networks under the standard parameterization *roughly* correspond to the random feature model; however, networks under the NTK parameterization, which is what we consider, do not.  In the NTK parameterization, changes to the inner layer weights are on par with the changes in the outer layer weights, and hence the kernel corresponding to the inner layer has an effect on par with the random feature kernel corresponding to the outer layer.  Furthermore, even in the standard parameterization, the connection between the network trained under the standard parameterization and the true random feature model is nonexact; see e.g. (A Comparative Analysis of Optimization and Generalization Properties of Two-layer Neural Network and Random Feature Models Under Gradient Descent Dynamics by Weinan E et al.) for a thorough discussion of this.  Also please note that the purpose of Corollary 3.6 was to clarify Theorem 3.5 and to demonstrate the dependencies of the theorem on the different parameters.  The goal was never to get a full learning guarantee, rather to demonstrate that certain components are learned faster due to the network inheriting the bias of the kernel.  Corollary 3.6 is a minor part of this work; we hope the other elements will be emphasized in your final decision.
>
> Also please note that in Theorem 3.5 **the target function does not need to be in the RKHS** associated with $K^\infty$.  The theorem holds for any target function in $L^2$.  The purpose of Theorem 3.5 is not to provide a learning guarantee but to show that in the underparameterized-setting the neural network inherits the bias from the associated kernel method along the top eigendirections.  If we were comparing against the NTK model to provide a learning guarantee you are correct there would be difficulties, as in the exact NTK regime changes to the residual are in the RKHS associated with $K^\infty$.
>
> You raise an interesting point about comparing to kernels other than $K^\infty$.  **The damped deviations equation allows you to compare the dynamics of the network to the dynamics of an arbitrary kernel** regression.  In the initial manuscript we only compared against $K^\infty$, however there are other natural choices.  The second most obvious choice is to compare against $K_0$ which we demonstrate in Appendix C.8 in the updated version of the manuscript in which we provide a version of Theorem 3.5 where $K^\infty$ is replaced with $K_0$ and the initialization is deterministic and arbitrary.  Comparing against other kernels such as $K_{t_0}$ for $t_0 > 0$ to account for adaptations to the kernel is an interesting avenue that we do not address in this work but is worthwhile to ultimately pursue.
>
> We hope that our technical contributions and the flexibility of the damped deviations point-of-view will factor into the final review.  In any case, thank you for being a conscientious reviewer and your helpful contributions.

---

### Decision · Program_Chairs · 2022-01-20

**Decision:**

Accept (Poster)

**Comment:**

There was a healthy discussion with all the reviewers with a consensus that the results are somewhat expected and unlikely to shed light beyond the ntk  regime, yet within the confine of ntk there is a solid and nicely written technical contribution.